# Twenty-first-century glacio-hydrological changes in the Himalayan headwater Beas river basin

Lu Li[1*], Mingxi Shen[3], Yukun Hou[3], Chong-Yu Xu[2,3], Arthur F. Lutz[5], Jie Chen[3], Sharad K Jain[4], Jingjing Li[3], Hua Chen[3]

[1]NORCE Norwegian Research Centre, Bjerknes Centre for Climate Research, Jahnebakken 5, 5007 Bergen

[2]Department of Geosciences, University of Oslo, Norway

[3]State Key Laboratory of Water Resources and Hydropower Engineering Science, Wuhan University, Wuhan, China

[4]National Institute of Hydrology, Roorkee, India

[5]FutureWater, Costerweg 1V, 6702 AA, Wageningen, Netherlands

*Correspondence to: Lu Li (luli@norceresearch.no)

**Abstract**. The Himalayan Mountains are the source region of one of the world's largest supplies of freshwater. The changes in glacier melt may lead to droughts as well as floods in the Himalaya basins, which are vulnerable to hydrological impacts. This study used an integrated glacio-hydrological model: Glacier and Snow Melt - WASMOD model (GSM-WASMOD) for hydrological projections under 21st century climate change by two bias correction methods under two Representative Concentration Pathways (RCP4.5 and RCP 8.5) in order to assess the future water (i.e., water availability and hydrological regime) change at the Himalayan Beas basin. Besides, the glacier extent loss of the 21st Century from eight GCMs was also investigated as part of the glacio-hydrological modelling as an ensemble simulation.

A high-resolution WRF precipitation suggested much heavier winter precipitation over high altitude ungauged area in the Himalaya Beas river basin, which was used for precipitation correction in the study for both the historical period and future scenarios. The glacio-hydrological modeling shows that at present, the glacier ablation accounts for about 5% of the annual total runoff during 1986-2004 in this area. Under climate change, the temperature will increase by 1.8 °C (RCP4.5) and 2.8 °C (RCP8.5) for the early future (2046-2065), and by 2.3 °C (RCP4.5) and 5.4 °C (RCP8.5) for the late future (2080-2099). In general, the uncertainty of projection from RCP8.5 is much larger than that from RCP4.5. Comparing two bias correction methods, i.e., the daily bias correction (DBC) and the local intensity scaling (LOCI), there is a wider spread of precipitation and temperature increase from DBC than that from LOCI. It is very likely that the Beas river basin will get warmer and wetter compared to the historical period. In this study, the glacier extent in the Beas river basin is projected to decrease over the range of 63 - 81 % (RCP4.5) and 76 - 87 % (RCP8.5) by the middle of the century (2050) and 89-99 % (RCP4.5) and 93-100 % (RCP8.5) at the end of the century (2100) compared to the glacier extent in 2005. This loss in glacier area will, in general, result in a reduction in glacier discharge in the future, while the future runoff is most likely to have a slight increase because of the increase from both precipitation and temperature under all the scenarios. However, there is widespread uncertainty regarding the changes of total discharge in the future, including the seasonality and magnitude. In general, the largest increase of river total discharge also has the largest spread. The uncertainty of future hydrological change is not only from GCMs but also comes from the bias correction methods. A decrease of discharge is found in July from DBC, while it is opposite from LOCI. Besides, there is a drop in evaporation in September from DBC, which cannot be seen from LOCI. The study helps to understand the hydrological impacts of climate change in North India and contributes to stakeholders and policymakers' engagement in the management of future water resources in North India.

## 1 INTRODUCTION

Outside the polar regions, the Himalayas store more snow and ice than any other place in the world. Hence, Himalayas are

also called the 'Third Pole" and are one of the world's largest suppliers of freshwater. Similar to the glaciers all over the world, the Himalayan glaciers are also changing as a result of global warming. Changes in glacier mass, ice thickness, and melt will impart major changes in flow regime of Himalayan basins. Among other things, it may lead to increased prevalence of droughts and floods in the basins of Himalayan rivers. Hydrological models have been developed and are being used as the main assessment tool to estimate the impacts of climate change for future water resources. However, most hydrological models either do not have a representation of glaciers (Ali et al., 2015; Horton et al., 2006; Stahl et al., 2008) or do not have proper glacier representation with limited glacier cover assumption (i.e., assumptions with intact glacier cover, 50% or none glacier cover) (Akhtar et al., 2008; Hasson 2016; Aggarwal et al. 2016). A glacio-hydrolgical model which includes a comprehensive parameterization of glaciers is highly required for the water resources assessment of high mountainous basins over the Himalayan region. Recently, Lutz et al. (2016) investigated the future hydrology by a glacio-hydrological model with a proper representative glacier module over the whole mountainous Upper Indus Basin (UIB) with an ensemble of statistically downscaled CMIP5 GCMs. Results obtained by them indicated a shift from summer peak flow towards the other seasons for the most ensemble members. An increase in intense and frequent extreme discharges is likely to occur for the UIB in the future of the 21st century according to their study. Besides, Li et al. (2016) applied a hydro-glacial model in two basins in the Himalayan region and assessed the future water resources under climate change scenarios, which were generated by two bias corrected COordinated Regional climate Downscaling EXperiment (CORDEX, Jacob et al. 2014) data from the World Climate Research Program (WCRP). However, their results showed a conflicting future glacier cover at the end of the century under different scenarios. Especially in Beas river basin, the result indicated that the glaciers are predicted to gain mass under Representative Concentration Pathways (RCP) 2.6 and RCP 4.5 while losing mass under RCP 8.5 for the late future after 2060. This conflicting future is not only seen for the glacier projections but also for the river flow. The impact of glacier melt on river flow is noteworthy in the future in the Himalaya region. On the one hand, some studies suggested an increase of future water availability in Upper Indus Basin over Himalayan region for the 21st century (Ali et al., 2015; Lutz et al., 2014; Khan et al., 2015). On the other hand, a substantial drop in the glacier melt and subsequent reduction in water availability are suggested for the near future by the other studies (e.g., Hasson, 2016). Furthermore, a few recent studies suggested highly uncertain water availability in the late/long-term future and no consistent conclusion can be seen in the UIB over Himalaya region (e.g., Lutz et al., 2016; Li et al., 2016; Hasson et al., 2016). As of now, there is a lack of in-depth understanding of the water resources in the future, which will be highly affected by glacier melting in the mountainous basin over Himalayan region (Hasson et al., 2014; Li et al., 2016; Lutz et al., 2016; Ali et al., 2015).

To investigate the climate change impact on the future water availability, the variables produced by GCMs are downscaled by an appropriate regional climate model (RCM) for use as inputs to hydrological models. This approach is adopted because the outputs of GCMs are too coarse to directly drive hydrological models at regional or basin scale (Akhtar et al., 2008). However, the RCM simulations have systematic biases resulting from an imperfect representation of physical processes, numerical approximations and other assumptions (Eden et al., 2014; Fujihara et al., 2008; Anand et al. 2017). Furthermore, some recent studies have evaluated CORDEX data and have highlighted the need for proper evaluation before use of RCMs for impact assessment for sustainable climate change adaptation. For instance, Mishra (2015) analyzed the uncertainty of CORDEX, and the results showed that the RCMs exhibit large uncertainties in temperature and precipitation in South Asia regional model and are unable to reproduce observed warming trends. Singh et al. (2017) compared CORDEX with GCMs and found that no consistent added value is observed in the RCM simulations of changes in Indian summer monsoon rainfall over the recent periods in general. In this case, concerning the large bias from GCMs and RCMs, the statistical downscaling is still the most popular and widely used approach for providing input in hydrological models in quantifying the impact of climate change on hydrology (e.g., Fang et al., 2015; Fiseha et al., 2015; Smitha et al., 2018). Previous studies have applied statistical downscaling methods based on GCM or RCM, as input for hydrological models over different basins in the world, including two widely used methods, i.e., regression-based downscaling methods (Chen et al., 2010, 2012) and bias

correction methods (Troin et al., 2015; Johnson and Sharma, 2015; Li et al., 2016; Ali et al., 2014; Teutschbein and Seibert, 2012). The regression-based downscaling methods, e.g., Statistical Downscaling Model (SDSM) (Wilby et al. 2002; Chu et al. 2010; Tatsumi et al. 2014) and support vector machine (SVM) (Chen et al. 2013), which involve estimating the statistical relationship (e.g., linear relationship for SDSM and nonlinear relationship for SVM) between large scale predictors (e.g., vorticity, mean sea-level pressure, geopotential height and relative humidity) and local or site-specific predictands (e.g., precipitation and temperature) using observed climate data. The reliability of a regression-based method relies on relationships between observed daily climate predictors and predictands. However, these relationships are usually weak, especially for daily precipitation (Chen et al. 2011). Besides, the regression-based method is usually incapable of downscaling precipitation occurrence and generating proper temporal structure of daily precipitation, which is critical for hydrological simulations. Moreover, this perfect prognosis (PP) downscaling method establishes a relationship between predictors and predictands for the historical period and then applies it to future periods. However, this relationship may not hold for the future in a changing climate. In particular, the relationship between predictors and predictands established using reanalysis predictors are applied to GCM predictors based on the assumption that reanalysis predictors and GCM predictors are both "perfectly" simulated at the grid scale (Wilby et al., 2002; Dibike and Coulibaly, 2005; Chen et al., 2011).

Another widely used statistical downscaling method, i.e., bias correction method, which involves estimating a statistical relationship between a climate model variable (e.g., precipitation) and the same variable of the observations to correct the climate model outputs. The use of bias correction methods is usually considered as reasonable way to achieve physically plausible results for impact studies. Some articles found that bias correction results in physical inconsistencies since the corrected variables are not independent of each other (e.g., Immerzeel et al., 2013; Cannon et al., 2015). For instance, although bias corrected RCM precipitation data are expected to improve the hydrological calibration results, they will no longer be consistent with modeled other variables, e.g., temperature, radiation. However, compared to PP methods, bias correction methods are relatively simple to use and negate the prerequisite of a strong relationship between local-scale variables and large-scale climate model variables. In this case, we chose bias correction method for downscaling in the study over Himalayan Beas river basin with very complex topography.

There are wide uncertainty resources in hydrological impacts under climate change and a number of articles have studied them (i.e., Chen et al., 2011, 2013; Pechlivanidis et al., 2017; Samaniego et al., 2017; Vetter et al., 2017; Shen et al., 2018). Chen et al. (2011) investigated the uncertainty of six dynamical and statistical downscaling methods in quantifying the hydrological impacts under climate change in a Canadian river basin. A significant uncertainty was found to be associated with the choice of downscaling methods, which is comparable to uncertainty from GCM. Chen et al. (2013) stated that the importance of uncertainty is geography dependent. The uncertainty of future extreme events is typically larger compared to that of the mean discharges. They noted that climate models usually underestimate the inter-annual variance of precipitation compared to the observations. Further, uncertainty associated with the choice of empirical downscale methods is similar to that related to RCM simulations. The study by Chen et al. (2013) also emphasized the importance of using several climate projections to delineate uncertainty when attempting a climate change impact study over a new region. Furthermore, a project called "Inter-Sectoral Impact Model Intercomparison Project phase 2" (ISI-MIP2) provides a excellent opportunity to investigate the propagation of forcing and model uncertainties impact on the century-long timer series of hydrological variables using an ensemble of hydrological model projections across a broad range of climate scenarios and regions in the world (Pechlivanidis et al., 2017; Samaniego et al., 2017). For example, in the study of Samaniego et al. (2017), six hydrological models were set up in seven large river basins over the world, which were forced by bias-corrected outputs from five GCMs under RCP2.6 and RCP8.5 for the period 1971-2099. They found that GCM uncertainty mostly dominated over Hydrological model uncertainty for the projections of runoff drought characteristics in general and emphasized the need for multi-model ensembles for the assessment of future drought projections. Pechlivanidis et al. (2017) investigate the future hydrological projections based on five regional-scale hydrological models driven by five

GCMs and four RCPs for five large basins over the world. They found that the high flows are sensitive to changes in precipitation, while the sensitivity varies between the basins. The results from their study also indicated that climate change impact studies can be highly influenced by uncertainty both in the climate and impact models; however, in the dry regions, the sensitivity to climate modelling uncertainty becoming greater than hydrological model uncertainty. More evaluation of uncertainty sources in projected hydrological changes under climate change was made by Vetter et al. (2017) over 12 large-scale river basins. The results showed that in general, the most significant uncertainty is related to GCMs, followed by RCPs and hydrological models, which are the lowest contributors of uncertainty for $Q_{10}$ and mean flow, but the hydrological models contribute more significant for $Q_{90}$.

However, the previous climate change impact studies have presented conflicting results regarding the largest source of uncertainty in essential hydrological variables, especially the evolution of streamflow and derived characteristics over glacier feed river basin over high mountainous ungauged or poor-gauged area, e.g., Himalayan region (Hasson et al., 2014; Li et al., 2016; Lutz et al., 2016; Ali et al., 2015). At present, a complete understanding of the hydroclimate variability is also a challenge in the Himalayan basins due to poor in-situ coverage (Maussion et al., 2011) and incomplete or unreliable records (Hewitt 2005; Bolch et al. 2012; Hartmann and Andresky 2013). An overview of the variation in precipitation estimates of gridded products was provided by Palazzi et al. (2013), in which six gridded products are compared with simulation results from a global climate model EC-Earth despite having different resolutions. In the Himalayan region, precipitation is strongly influenced by terrain. The regional patterns and amounts of the precipitation are not always captured by global gridded precipitation datasets, e.g., Tropical Rainfall Measuring Mission (TRMM) (Huffman et al. 2007), ERA-Interim (ECWMF, Dee et al., 2011), Climate Research Unit (CRU) (Mitchell and Jones, 2005), and the Asian Precipitation - Highly-Resolved Observational Data Integration Towards Evaluation (APHRODITE) (Yatagai et al. 2012) (see also Biskop et al. 2012; Dimri et al. 2013; Ménégoz et al. 2013; Ji and Kang 2013). Previous studies showed that the high-resolution (<4km grid spacing) RCMs had demonstrated reasonable skill in reproducing precipitation distribution and intensity patterns over complex terrain (e.g., Rasmussen et al. 2011, 2014; Collier et al. 2013). A high-resolution Weather Research and Forecasting (WRF) dynamical simulation has been applied in the Beas basin in Himalaya showing promising potential in addressing the issue of high spatial variability in the complex terrain and high elevation precipitation (Li et al., 2017). This high-resolution WRF simulation from Li et al. (2017) provides an estimation of liquid and solid precipitation in high altitude areas, where satellite and rain gauge networks are not reliable.

The following questions are studied in this paper: (1) How much uncertainty is in the precipitation over the ungauged high-altitude in Beas river basin? (2) How will the future water availability change due to higher glacier melt under warmer future in Beas river basin over the Himalayan region? (3) How are the uncertainties of the future water from GCMs or statistical downscaling methods?

To answer these questions, precipitations from a high-resolution WRF simulation and Gauge are investigated, and corrected precipitation is used for the hydrological simulation for the historical baseline, as well as in the future scenarios. In the study, we use a glacio-hydrological model together with two ensembles of four GCMs under two generation of scenarios, i.e., RCP 4.5 and RCP 8.5, and two bias correction methods. We firstly focus on the simulation of the present day water cycle, calibration and validation of the glacier mass balance and discharge by observations. The uncertainties of the precipitation over the high-altitude area and hydrological simulation are further discussed. Besides, the future climate change, glacier extent change and hydrological change are investigated. At last, the uncertainty from GCMs and statistical downscaling methods is analyzed and discussed before presenting the main conclusions.

## 2 STUDY AREA AND DATA

### 2.1 Study area

The study area is Beas river basin upstream of the Pandoh Dam with a drainage area of 5406 km$^2$, out of which only 780 km$^2$ (14%) is under permanent snow and ice. It is one of the important rivers of the Indus River system. The length of the Beas River up to Pandoh is 116 km; among its tributaries, Parbati and Sainj Khad Rivers are glaciers fed. The altitude of the study area varies from about 600 m to above 5400 m above mean sea level (a.m.s.l.). The study area falls in a lower Himalayan zone and varies in climate due to elevation difference. The mean annual precipitation is 1217 mm, of which 70% occurs in the monsoon season from July to September. The mean annual runoff is 200 m$^3$/s, of which 55% occurs in the monsoon season and only 7.2% occurs in winter from January to March (Kumar et al., 2007). The mean temperature rises above 20ºC in summer and falls below 2ºC in January. The topography and drainage map of the river system along with rain gauge stations is shown in Fig. 1.

### 2.2 Data

The basin boundary in the study is delineated based on HYDRO1k (USGS, 1996a), which is derived from the GTOPO30 30-arc-second global-elevation dataset (USGS, 1996b) and has a spatial resolution of 1 km. HYDRO1k is hydrographically corrected such that local depressions are removed, and basin boundaries are consistent with topographic maps. Daily precipitation of 7 gauge stations, daily minimum and maximum temperature and relative humidity of 4 meteorological stations obtained from Bhakra Beas Management Board (BBMB) in India were used for GSM-WASMOD modelling. The discharge of Thalout station was used for GSM-WASMOD model calibration and validation, which was also obtained from the BBMB. The hydrological and meteorological data from 1990 to 2005 were used, which have undergone quality control in the previous studies (Kumar et al., 2007, Li et al., 2013a, 2015a). Glacier outlines were taken from the recently published Randolph Glacier Inventory (RGI 6.0) (2017) (https://doi.org/10.7265/N5-RGI-60). The annual glacier mass balance data of Chhota Shigri Glacier that are used in the model calibration are taken from the previous studies of Berthier et al. (2007), Wagnon et al. (2007); Vincent et al. (2013) and Azam et al. (2014). Two ensembles of four GCM models under RCP4.5 and RCP 8.5, including CamESM2, CSIRO_Mk3_6_0, Inmcm4, IPSL_CM5A_LR, MIROC5, MRI_CGCM3 and MRI-ESM1 (Taylor et al., 2012) are chosen for driving the empirical statistical downscaling future simulations (see in Table 2). Furthermore, the daily precipitation from a horizontal high-resolution of 3 km WRF simulation by Li et al. (2017) is also used in the study for further bias correction of high mountainous winter precipitation in all the simulations.

## 3 METHODOLOGY

### 3.1 Glacier- and snow- melt module (GSM)

A conceptual glacier- and snow-melt module (GSM) (Li et al., 2013a; Engelhardt et al., 2012) was used to compute glacier mass balances and melt-water runoff from the glacier in the study basin, which was only applied to the grid cells of the glacier-covered area. Those glacier grid cells were defined by ESRI ArcGIS system v. 9.0 (or higher) and set up before modeling based on the glacier outlines from the RGI (6.0) (2017) (Berthier, 2006; Raup et al., 2007). The gridded temperature and precipitation are interpolated based on the station data by Inverse Distance Weighted (IDW) method, in which the vertical temperature lapse rate of $-6\,°C\,km^{-1}$ is used to convert station temperature to the elevations of the grid cells (Kattel et al., 2013). The daily gridded temperature and precipitation were input data for the GSM module, which calculated both snow accumulation and melt-water runoff. A temperature-index approach (Hock, 2003; Engelhardt et al., 2012, 2017) was used in the study for the calculation of the conceptual GSM module. In the GSM module simulation, the precipitation shifted from rain to snow linearly within a temperature interval of ΔT (Table 1). Additionally, the liquid water from rain or melt infiltrated and refrozen in the snowpack, which filled the available storage. Runoff occurred when the storage was filled, which depended on the snow depth.

The snow melting started firstly, followed by the melting of the refrozen water and firn. At last, the ice started to melt when the firn has all melted away. We used different degree-day factors of firn ($DDF_f$) and ice ($DDF_i$), which are 15 % and 30 % larger than that of snow ($DDF_s$), respectively (Singh et al., 2000; Hock, 2003). The debris cover is not yet considered in the modeling right now. The related equations can be found in Table 1.

## 3.2 GSM-WASMOD model

A integrated glacio-hydrological model: Glacier and Snow Melt - WASMOD model (GSM-WASMOD) was developed by coupling the water and snow balance modeling system (WASMOD-D) (Xu, 2002; Widen-Nilsson et al., 2009; Gong et al., 2009; Li et al., 2013b, 2015b) with the GSM module. The spatial resolution of the GSM-WASMOD modeling is 3 km in the study. The daily precipitation, temperature and relative humidity from the observed stations were interpolated by the IDW method to 3 km resolution gridded data, which were used as input for the GSM-WASMOD model. For the temperature, the vertical temperature lapse rate of $-6\,°\mathrm{C\,km^{-1}}$ is used in the interpolation. GSM-WASMOD calculates snow accumulation, snowmelt, actual evapotranspiration (ET), soil moisture, fast flow and slow flow at the non-glacier area. The routing process of GSM-WASMOD model in the study is the aggregated network-response-function (NRF) routing algorithm, which was developed by Gong et al. (2009). The spatially distributed time-delay was calculated and preserved by the NRF method based on the 1 km HYDRO1k flow network, which is from the U.S. Geological Survey (USGS). The runoff generated in the resolution of 3 km grid was transferred by the NRF method based on the simple cell-response function. More details can be found in Gong et al. (2009). The equations of GSM-WASMOD model are shown in Table 1.

## 3.3 Glacier evolution parameterization

GSM-WASMOD is a conceptual glacio-hydrological model, which means that the glacier extent is not changing in the historical simulation. This assumption cannot be applied in future simulations under climate change since the future of the glacier extent is a crucial factor for the future hydrology in the Beas river basin. In this case, we used a basin-scale regionalized glacier mass balance model with parameterization of glacier area changes and subsequent aggregation of regional glacier characteristics (Lutz et al., 2013), to estimate future changes in glacier extent. It estimates changes in the glacier extent as a function of the glacier size distribution and distribution over altitude and temperature and precipitation. The model is calibrated to the observed glacier mass balance (e.g., Azam et al., 2014), and subsequently forced with an ensemble of statistically downscaled climate scenarios (section 3.4, Table 2). The model runs at a monthly time step to ensure that seasonal differences in the climate change signal are taken into account. A detailed description the glacier evolution parameterization is described in Lutz et al., (2013).

## 3.4 Bias correction methods

Since GCM outputs are spatially too coarse and too biased to be used as direct inputs to glacio-hydrological model for impact studies, downscaling or bias correction techniques must be applied for generating site-specific climate change scenarios (Rudd and Kay 2016). In this study, two bias correction methods, i.e., Daily bias correction (DBC) (Schmidli et al., 2006; Mpelasoka and Chiew, 2009; Chen et al. 2013) and Local intensity scaling (LOCI) (Schmidli et al., 2006; Chen et al., 2011), with different levels of complexity were applied for correcting GCM-simulated daily precipitation, temperature and relative humidity in the Himalayan Beas river basin under climate change of the 21st Century (i.e., 2046-2065 and 2080-2099).

### 3.4.1 Local intensity scaling (LOCI)

LOCI is a mean-based bias correction method, which corrects the precipitation frequency and quantity at monthly basis with the following three steps: (1) a wet-day threshold is determined from the GCM-simulated daily precipitation series for each

calendar month to ensure that the threshold exceedance for the reference period equals the observed precipitation frequency in that month; (2) a scaling factor is calculated to ensure that the mean of GCM precipitation for the reference period is equal to that of the observed precipitation for each month; (3) the monthly thresholds and scaling factors determined in the reference period are further used to correct GCM precipitation in the future period. Since there is no occurrence problem for humidity, LOCI only corrects the mean value of GCM-simulated humidity for each month. In addition, the mean and variance of temperature are corrected using the variance scaling approach of Chen et al. (2011).

### 3.4.2 Daily bias correction (DBC)

DBC is a distribution-based bias correction method. Instead of correcting the mean value, the DBC method corrects the distribution shape of GCM-simulated climate variable. Specifically, the ratio (for precipitation and humidity) or difference (for temperature) between observed and GCM-simulated data in 100 percentiles (from 1th percentile to 100th percentile) at the reference period multiplied or added to the future time series for each percentile. The wet-day frequency of precipitation occurrence is corrected using the same procedure of LOCI. The DBC method is also carried out on a monthly basis.

Both bias correction methods are calibrated in the historical period of 1986-2005 from the observations. The calibration of downscaling models used the station-scale meteorological data and GCM historical variables to construct the relationship. The calibrated bias correction models are then utilized to predict the future climate change for the meteorological variables including precipitation, temperature and relative humidity in two periods, i.e., early future of 2046-2065 and the late future of 2080-2099, under both the RCP4.5 and RCP8.5 scenarios.

### 3.5 Precipitation correction

According to the previous studies over Himalaya and surrounding area (Winiger et al., 2005; Immerzeel et al., 2015; Ji et al., 2015; Shrestha et al., 2012), specifically in Beas river basin up to Pandoh, there are quite large uncertainties in precipitation over high altitude area. Li et al. (2017) applied the Weather Research and Forecasting model (WRF) over Beas river basin at high-resolution of 3 km in 1996-2005. The seasonal WRF precipitation compared with gauge rainfall data is shown in Fig. 2, which indicates that the WRF model predicts more winter precipitation at high altitude area in Beas river basin. Currently, we have no rainfall and snowfall observation data at the high mountainous area. The highest gauge station is Manali (see Fig. 1), whose altitude is 1926 m a.m.s.l.

In this study, we have compared the data from the high-resolution 3 km WRF simulation with gauge precipitation during the overlapping period of 1996-2005. The winter precipitations from gauge and WRF over different altitudes are listed in Table 3, from which we can see that the winter precipitations from WRF at mountainous over 4000 m and 4800 m a.m.s.l. are almost triple times as that from Gauge. This is comparable with the results from previous studies (Immerzeel et al., 2015; Dahri et al., 2016). For example, Immerzeel et al. (2015) estimated annual precipitation of altitude over upper Indus Basin and found that an increase of over 300% over the uncorrected high mountainous precipitation between 3751 m and 4250 m a.m.s.l. It was also suggested in their study that APHRODITE underestimates annual precipitation by as much as 200% over the upper Indus Basin (Immerzeel et al., 2015). In the study of Dahri et al. (2016), a basin-wide, seasonal and annual correction factor for each gridded precipitation product was provided based on a geo-statistical analysis of precipitation observations which revealed substantially higher precipitation in most of the sub-basins compared to earlier studies. For the high-altitude western and northern Himalayan basins, including Indus, the correction factor for winter precipitation varies from 1.93 to 2.47 and from 1.82 to 4.44 comparing with APHRODITE and TRMM, respectively. Considering that we lacked observed precipitation over the high mountainous area in Beas river basin, especially in the winter period, we bias corrected the winter precipitation (December - March) from gauge station with the WRF precipitation to provide more reliable precipitation for the Glacier-hydrological model calibration and validation. However, we cannot evaluate the correction factors of WRF/Gauge for winter precipitation, although WRF shows reasonable performances on winter precipitation over complex terrain in previous studies (Rasmussen et al., 2011; Li et al.,

2017).  In this case, we chose an average value of 2.7 in the study for the winter precipitation (DJFM) correction in Beas river basin for all the grids whose altitude is over 4800 m a.m.s.l.. The same bias correction is also applied for the winter precipitation for all the future scenarios.

## 3.6 GSM-WASMOD Model calibration

There are six parameters to be calibrated in GSM-WASMOD by searching for an optimal parameter set for the discharge at the Thalout station, including the snowfall temperature $a_1$, snowmelt temperature $a_2$, actual evapotranspiration parameter $a_4$, the fast-runoff parameter $c_1$, the slow-runoff parameter $c_2$ and the degree-day factor of snow $DDF_s$. The observed average annual glacier mass balance and discharge in Beas River basin are both used for the calibration in the study. There is an intra-regional variability of individual glacier mass balance in High Mountain Asia (HMA) in the recent study of Brun et al. (2017). From their study, the glacier mass balance is -0.49+/-0.2 annual meter water equivalent (m w.e. $a^{-1}$) in Spiti-Lahaul region (where Chhota Shigri glacier locates) during 2000-2008 based on ASTER and 0.37+/-0.09 m w.e. $a^{-1}$ in Western Himalaya region from RGI Inventory during 2000-2016 based on ASTER. Besides, a detailed map of elevation changes during 2000-2011 in Spiti-Lahaul region based on SPOT5 DEM is given in the study of Gardelle et al. (2013), which showed that the changes of the glaciers in the Beas river basin are quite similar to the changes in Chhota Shigri glacier during 2000-2011 in general, although there is variability both in independent glacier and over the region. Furthermore, in our study basin, the glaciological mass balance series published in Spiti-Lahaul region (of HMA) available for comparison, are the Chhota Shigri glacier and Bara Shigri glacier (Berithier et al. 2007). In which, the only one is long enough to be comparable to our simulation period is the Chhota Shigri glacier (2002-2014), which also has geodetic mass balance for validation (Azam et al. 2016). So we used the mass balance data of Chhota Shigri glacier as a representation for the glaciers in our small basin (see Fig. 1 and Table 4). In the calibration, we firstly 'pre-calibrate' all parameters according to the observed discharge data of Thalout station. Secondly, we manually adjusted the parameters of glacier module according to the observed annual glacier mass balance data in Table 4, which is from previous studies (Berthier et al. 2007; Wagnon et al., 2007; Vincent et al. 2013; Azam et al. 2014, 2016). Then, all parameters except the glacier module parameters were re-calibrated according to discharge data at the very last time. The calibration and validation period in this study were 1986-2000 and 2001-2004, respectively. We used the data of 1986 for three preceding spin-up years. All the calibration and validation results of glacier mass balance in the study are listed in Table 4.  In the study, we used 1986-2004 period (2005 was included in the calibration and simulation of bias correction) for glacier and hydrological calibration and validation, because those are the periods fit to the available glacier mass balance data from previous studies. In the calibration, GSM-WASMOD run with the 5000 parameter sets, which were obtained by the Latin-Hypercube sampling method (Gong et al., 2009, 2011; Li et al., 2015a). The best parameter set was then chosen based on three indices, including Nash-Sutcliffe coefficient (NSC), relative volume error (VE) and root-mean-square error (RMSE). For the best model performance, the NSC is to be 1 and the other two indices, i.e., VE and RMSE, are to be 0.

## 4 RESULTS

### 4.1 Corrected Precipitation

The uncorrected and corrected mean annual precipitation (1986-2004) are 1213 mm/yr and 1374 mm/yr, respectively. The calibration results (1986-2000) show that the daily NSC driving by uncorrected and corrected precipitation is 0.64 and 0.65, respectively (Table 5). The RMSE, VE and monthly NSC from the calibration of GSM-WASMOD driving by the corrected precipitation are 2.01, 7% and 0.75, respectively, while those by uncorrected precipitation are 2.03, 8% and 0.70, respectively. It shows an improvement of all indices in both calibration and validation from the corrected precipitation comparing with that from uncorrected precipitation. The results confirmed that there is much heavier precipitation at high altitude in Himalaya regions than what we knew from the gauge data and other gridded data set. The high-resolution precipitation of RCM, i.e.,

WRF, has the potential to provide more information and knowledge for the high altitude precipitation in Himalaya region, although it still has challenges in capturing the precipitation variability accurately at high-resolution spatial scale (i.e., complex topography) and temporal scale (i.e., daily or hourly).

**4.2 GSM-WASMOD model calibration and validation**

The calibration (1986-2000) and validation (2001-2004) results from WASMOD and GSM-WASMOD are given in Table 5, which shows that GSM-WASMOD has improved the performance of WASMOD in reproducing historical discharge in Beas river basin. For example, for the GSM-WASMOD modeling, the daily NSC and monthly NSC in the calibration are 0.65 and 0.75 respectively, which are 0.61 and 0.66 respectively in the validation. While for the WASMOD model, the daily NSC and monthly NSC in calibration are 0.50 and 0.65 respectively, which are only 0.31 and 0.36 in the validation. It shows that the GSM-WASMOD performs more reliably than WASMOD comparing the results from both calibration and validation. Furthermore, the precipitation correction has improved the modeling performance in Beas river basin, especially regarding the results of model validation. For the Beas river basin, located to North mountainous India, the model underestimates the flow during June-August, which leads to a large negative bias (Fig. 3). The mean annual un-corrected precipitation and corrected precipitation is 1213 mm/yr and 1374 mm/yr of 1986-2004, while the observed discharge of 1284 mm/yr is even larger than the uncorrected precipitation. The bias is most likely related to an underestimation of precipitation due to limited rain gauge stations, although we did precipitation correction over high mountain area in winter period. In Fig. 4, the total discharge includes fast-flow and slow-flow from the non-glacier area and discharge from the glacier area, which includes rainfall discharge, snow-melt and ice-melt discharge. The fast-flow is generally considered to be the surface runoff and the slow-flow refers to base-flow.

The Chhota Shigri glacier is the only one which has been well studied and has detailed and longer period of glacier mass balance data in the Spiti-Lahaul region where Beas river basin locates. The Chhota Shigri Glacier is a part of the Chandra Basin, which is a sub-basin of the Chenab river basin (Ramanathan, 2011), but it is attached to northeast boundary of Beas river basin, which is close to Manali and Bhunter stations (Fig.1). In this case, the glacier mass balance of Chhota Shigri Glacier is used for glacier module calibration in the study, which is to be comparable to the simulation. The total runoff (including rainfall discharge, ice-melt and snow-melt discharge) from glacier cover area contribute about 19 % of total runoff and the glacier imbalance is about 5 % of total runoff in Beas River basin up to Thalout station during 1986-2004. The monthly hydrography of ice and snow melt discharge, total glacier area discharge, and simulated and observed discharges during the calibration and validation period are shown in Fig. 5. For validation of the model results on glacier mass balance, we compared our results to the previous studies (Table 4 and Fig. 6). For example, the simulated annual glacier mass balance of Beas river is -0.22 m w.e. a$^{-1}$ of 1986-2000 in our simulation, which is comparable to the results of the modelled annual glacier mass balance of Chhota Shigri glacier (1986-2000), which is -0.01 (+/-0.36) m w.e. a$^{-1}$ by Azam et al. (2014) and -0.29 (+/-0.33) m w.e. a$^{-1}$ by Engelhardt et al.(2017). Besides, the annual glacier mass balance is -1.09 m w.e. a$^{-1}$ of 1999-2004 from our study, which is also similar with the results from the other two previous studies, i.e., the measured annual glacier mass balance (1999-2004) of Chhota Shigri glacier is -1.02 or -1.12 m w.e. a$^{-1}$ from geodetic measured by Berthier et al. (2007) and -1.03(+/- 0.44) m w.e. a$^{-1}$ by Vincent et al. (2013). Considering the uncertainties in the meteorological forcing data and high complexity in the hydrological cycle over high altitude Himalaya mountainous area, the model is considered to be satisfactory for estimating the impacts of climate change for the future Beas's water.

**4.3 Evaluation of LOCI and DBC**

The performance of LOCI and DBC in correcting precipitation and temperature is evaluated using two common statistics over the historical period (1986-2005): mean and standard deviation. Fig. 7 shows an example of evaluation results of corrected precipitation and temperature at the Pandoh station. The figure shows that GCM-simulated precipitation and temperature are

considerably biased concerning reproducing the mean and standard deviation. Both LOCI and DBC are capable of reducing the bias of mean and standard deviation of precipitation and temperature at the reference period, even though there are some uncertainties related to GCMs. However, DBC performs much better than LOCI at reproducing the standard deviation of precipitation, which is expected, because the standard deviation of precipitation was not specifically considered in LOCI. In other words, LOCI only corrected the mean of monthly precipitation. However, this is not the case of DBC, as it corrected the distribution shape of precipitation. The standard deviation was corrected along with the mean. For temperature, both LOCI and DBC can remove biases of mean and standard deviations for the reference period. Above evaluation results indicate the reasonable performance of both bias correction methods. The precipitation in Fig. 7 is un-corrected precipitation from DBC and LOCI, which are different from the precipitation in Fig. 8 that shows the corrected precipitation (based on the precipitation correction method in section 3.5).

## 4.4 Future climate change

The climate change scenarios for GSM-WASMOD simulation are illustrated in Table 2. The changes of mean monthly precipitation and temperature of the Beas river basin in the early future (2046-2065) and the late future (2080-2099) compared with the baseline period (1986-2005) are shown in Fig. 8 and Fig. 9. In general, the temperatures from DBC and LOCI are all shown increasing for all scenarios for the both early and later future; while there is more uncertainty in precipitation change in the future. It is consistent with the annual precipitation and temperature changes of the Beas river basin, which are shown in Fig. 10. From the figure, we can see that under Climate change impact, the study area will be getting warmer. The uncertainty of temperature increase in the late future is much larger than that from early future, while for the future change of precipitation, both early and late future have a widespread uncertainty, especially by LOCI method. It is worth to point out that the winter precipitation (December -March) in Fig. 8 is much higher than that from Fig. 7. This is because the precipitation correction has made in Fig. 8. A more detailed statistical analysis result is shown in Table 6, which is based on the corrected precipitation. The annual mean temperature of Beas river basin is approximately warm up to ~1.8°C (RCP4.5) and ~2.8 °C (RCP8.5) in the middle of the century (2046-2065) comparing with baseline period (1986-2005), and up to ~2.3 °C (RCP4.5) and ~5.4 °C (RCP8.5) at the end of the century (2080-2099) comparing with the same baseline period. For the annual mean precipitation, the change will be +9.8 % (RCP4.5) and +33.3 % (RCP8.5) in the middle of the century (2046-2065) comparing with the baseline period (1986-2005), and +17.7 % (RCP4.5) and +39.7 % (RCP8.5) in Beas river basin at the end of the century (2080-2099). However, there is a similar widespread of uncertainty in precipitation increase from LOCI as DBC. While for the temperature increase, the uncertainty spread of temperature increase from DBC is much wider than that from LOCI, especially under RCP85 for late future (2080-2099). It is very likely that the Beas river basin will get warmer and wetter compared to the historical period, which are also confirmed by other studies (e.g., Aggarwal et al., 2016; Ali et al., 2015). Under DBC RCP8.5, the temperature increases the most, while for precipitation, the LOCI RCP8.5 increases most.

## 4.5 Future glacier extent change

The projected changes in glacier extent in the Beas river basin under eight climate change scenarios are shown in Fig. 11. Unsurprisingly, the glacier extent will keep retreating in the future at Beas river basin. There are large uncertainties in the changes of the glacier extent from different projections (Fig. 11), which are confirmed by other studies (e.g., Kraaijenbrink et al., 2017, Lutz et al., 2016; Li et al. 2016). In this study, the glacier extent in the Beas river basin is projected to decrease over the range of 63 - 81 % (RCP4.5) and 76 - 87 % (RCP8.5) by the middle of the century (2050) and 89 - 99 % (RCP4.5) and 93 - 100 % (RCP8.5) at the end of the century (2100) compared to the glacier extent in 2005. The range in the projections is comparable for both statistical downscaling methods. The rapid decrease in glacier extent is mainly driven by strong temperature increase, which cannot be compensated by an increase in precipitation. In the Beas river basin, approximately 90%

of the glacier surfaces is located between 4500 and 5500 m a.m.s.l. This relatively small altitudinal range may be another reason for the rapid retreat.

## 4.6 Future Hydrological changes

There is a consistent trend of projected hydrological changes over all the scenarios, although there are large uncertainties. The glacier discharge is projected to decrease over the century across all the scenarios led by the glacier extent decrease (Fig. 12), while the future change of total discharge over Beas river basin is not that clear in Fig. 13. This is most likely because of the increase in both precipitation and temperature throughout the whole 21st century. There is a wide spreading of glacier ablation near the middle of the century, which indicates a larger uncertainty in the prediction discharge over this period. Table 6 provides more details of the change of glacier extent, precipitation, temperature, discharge and evaporation (ET) in Beas river basin in the middle of the century (2046-2065) and at the end of the century (2080-2099) comparing with the historical baseline period (1986-2005). There are large ranges in different climate change scenarios. The future delta change of (future minus baseline) and future predicted mean monthly evaporation and discharge over Beas river basin up to Pandoh are shown in Fig. 14 and Fig. 15. According to those two figures, we can see that (1) the projected discharge will increase in general especially in winter and pre-monsoon under both RCP4.5 and RCP8.5 for near future (2045-2055) and far future (2080-2099); (2) under RCP8.5, there is a slight decrease in discharge can be seen from the mean results of DBC during monsoon season, especially in July, also with the largest uncertainty comparing with other seasons. One of the main reasons for this decrease of summer discharge is probably the significant glacier retreating under the future climate; (3) the largest change of discharge can be observed in July for near future (2046-2065), which also has the widest range, i.e., from -99 mm to over 265 mm by LOCI and from -120 mm to 108 mm by DBC; (4) for the late future (2080-2099), the widest discharge change can be observed in August, which is from -117 mm to 309 mm by LOCI method and from around -145 mm to over 228 mm by DBC method. This is probably due to both the glacier extent decrease and the temperature increase. The uncertainty of projected discharge under RCP8.5 is much larger than that under RCP4.5; (5) for the evaporation, a general increase can be seen all over the year from both LOCI and DBC; (6) the largest increase of evaporation will be in April, with also the largest spread, i.e., around 5 ~ 26 mm and 1 ~ 26 mm by LOCI and DBC, respectively. This large evaporation increase most likely is driven by the increase of both precipitation and glacier melting regarding increased temperature, which will provide a much wetter environment in the future than the historical periods.

## 5 DISCUSSIONS

### 5.1 Uncertainty of high mountain precipitation

There are many uncertainties and challenges for the future hydrological projection under climate change in the Beas river basin. The dedication of snow and glacier melting is significant for the total runoff, which varies from 27.5 % ~ 40% by previous studies (e.g., Kumar et al. 2007; Li et al. 2013a, 2015a). In our study, the total snow and glacier melting from the glacier-covered area is 19% of the total runoff, and the glacier retreat is accounting for round 5% during 1986-2004, which is comparable with the same value of 5% during 2003-2008 by Kääb et al. (2015), who used ICESat satellite altimetry data. There are several reasons for this large spread of percentage of snow and glacier melting in the Beas river basin. Most common knowledge of one of the challenges in high mountain area is the data issue. A large disagreement between precipitation from dynamical RCM simulations (WRF) and other data sources (i.e., TRMM 3B42 V7, APHRODITE and gauge data) were found over high altitude in the Beas river basin by the previous study of Li et al. (2017). There are no gauge stations over 2000 m a.m.s.l. in our study, and neither of the gauge stations includes appropriate snowfall measurement. Lacking of reliable snowfall measurement over the Himalaya regions is one of the reasons for a poor understanding and a large uncertain of high altitude precipitation over this area (Mair et al., 2013; Ragettli and Pellicciotti, 2012; Immerzeel et al., 2013, 2015; Viste and Sorteberg, 2015; Ji et al., 2015; Dahri et al., 2016). Some previous studies showed that the high altitude precipitation is much larger than

previously thought and other datasets (Immerzeel et al., 2015; Li et al., 2017; Dahri et al., 2016). Dahri et al. (2016) applied a geo-statistical analysis of precipitation observations revealed substantially higher precipitation in most of the sub-basins compared to earlier studies and they pointed out that the uncorrected gridded precipitation products are highly unsuitable to estimate precipitation distribution and to drive glacio-hydrological models in water balance studies in the high-altitude areas of Indus basin. Comparison of the high-resolution WRF precipitation with gauge rainfall showed an underestimation of WRF at Manali station in the summer period (July-September). The Manali precipitation is more heavily influenced by the complex topography than other stations because it locates at a bit deeper valley in the mountains. This is probably the main reason that WRF underestimates the rainfall in summer period comparing with gauge rainfall. While for winter period (December-March), the WRF results showed much larger precipitation over high altitude in Beas river basin comparing with gauge rainfall. Although we did precipitation correction based on this high-resolution WRF precipitation, which improved results for both calibration and validation in the study, the real amount of precipitation over Himalayan region, like Beas river basin, is still uncertain.

## 5.2 Uncertain future of glacio-hydrological changes in Beas river basin

In our study, the results show a large uncertainty in the future river flow changes over the Beas river basin up to Pandoh among all the future scenarios, although the glacier is retreating from all the scenarios. From the results, we can see that there are differences (i.e., seasonal change and hydrological element's variability) from those two BC methods, i.e., LOCI and DBC, although in general, the annual changes of the main variables in hydrological cycle are similar from those two BC methods. For example, the discharge during the monsoon period (June-August) is likely to decrease, although it varies a lot within the impact of all the GCMs, RCP and BC methods. The main decrease is found in July from DBC, while a slight increase can be seen from the mean of LOCI. Besides, the peak flow in the middle of the century is slightly shifted to be early in July from the LOCI, which confirmed the study result from Lutz et al. (2016), while this change cannot be seen in the results from DBC. In general, the future runoff over Beas river basin is likely to increase slightly, especially in the winter and pre-monsoon period, with large uncertainty in the summer period. The results are consistent with some previous studies. For instance, the future river flow in the Beas river basin was projected to be increasing for the future periods (during 2006 ~ 2100) compared with the baseline period of 1976-2005 by Ali et al. (2015). In their study, however, the future hydrological simulation was lacking glacier component, which did not account for glacier retreat under future climate change impact. In the other study of Li et al. (2016), a large spread of river flow changes from different scenarios can be seen, and no uniform conclusion can be conducted from their projections. Furthermore, there is an obvious evaporation decrease in September from DBC method, which cannot be seen from the LOCI method. From our study, we can see that the uncertainty of future hydrological change comes not only from GCMs but also from the two bias correction methods.

There are several limitations of this study that need to be addressed. Firstly, only two bias correction methods were used in the study. According to the previous studies, bias correction results in physical inconsistencies since the corrected variables are not independent of each other (Ehret et al., 2012; Immerzeel et al., 2013). For instance, although bias corrected precipitation data will improve the hydrological calibration results, it will no longer be consistent with modeled other variables, e.g., temperature, radiation. It is generally based on the assumption of stationary climate distribution regarding the variance and skewness of the distribution, which however is crucial for assessing the impact of climate change on seasonality and extremes of the hydrological cycle. More ensemble statistical downscaling methods are needed for predicting future river flows to include enough uncertainties and to have a better picture of the robust future hydrological impact assessment. Secondly, the simplification of glacier module, especially without considering the effect of debris, will also result in uncertainty in the results (Scherler et al., 2011; Azam et al., 2018). Furthermore, the limitations of data, e.g., sparsely rainfall stations and no snowfall measurement, in such high-mountain drainage basin also lead to considerable uncertainty in hydrological simulation, and this is a common challenge for modeling study in this region.

**6 CONCLUSIONS**

An integrated glacio-hydrological model: Glacier and Snow Melt - WASMOD model (GSM-WASMOD) was applied for investigating the hydrological projection under climate change during the 21st century in the Beas basin. The river flow is impacted by the glacier melt. The glacier extent evolutions under climate change were estimated by a basin-scale regionalized glacier mass balance model with parameterization of glacier area changes, which were used in the study for constructing the future glacier extent scenarios in the GSM-WASMOD model for investigating the hydrological response of Beas river basin up to Pandoh. The changes of precipitation, temperature, runoff and evaporation in Beas river basin in the early future (2046-2065) and the late future (2080- 2099) were investigated in the study.

A high-resolution WRF precipitation suggested much higher winter precipitation over high altitude area in the Beas river basin than we knew from the gauge data and other available gridded datasets, which was used for precipitation correction in our study. The results indicate that the corrected precipitation is more reliable and performs better in both the calibration and validation of GSM-WASMOD in the Beas river basin, compared with the uncorrected precipitation. Besides, the calibration and validation based on both glacier mass balance and discharge show that GSM-WASMOD, which although has only a conceptual glacier module, performs much better than the early version of WASMOD. Furthermore, the results reveal that the glacier imbalance of -0.4 (-1.8 ~ +0.6) m w.e. $a^{-1}$ is about 5 % of total runoff during 1986-2004 in Beas River basin up to Thalout station at present (1990-2004).

Under Climate change impact, the temperature will increase by 1.8 °C (RCP4.5) and 2.8 °C (RCP8.5) for the early future (2046-2065), and increase by 2.3 °C (RCP4.5) and 5.4°C (RCP8.5) at the late future (2080-2099), while the precipitation will increase by 9.8 % (RCP4.5) and 33.3 % (RCP4.5) for the early future, and increase by 17.7 % (RCP4.5) and 39.7 % (RCP8.5) for the late future over the Beas river basin. However, there is a large uncertainty spread during different future scenarios based on the impact of GCMs and RCPs. The glacier extent loss is about 73 % under RCP4.5 scenario and 81 % under RCP8.5 scenario at the early future and 94 % under RCP4.5 scenario and 99 % under RCP8.5 scenario at the late future, which results in a loss of discharge in monsoon period. There was a broad spread of evaporation and discharge change in the Beas river basin in the future scenarios. The runoff was projected to have a slight increase from the mean of all the future scenarios, although the changes vary with seasons and have a large uncertainty. The precipitation increase and glacier retreat make a complex future of total discharge with a general increase in winter and pre-monsoon period, while considerable uncertainty can be seen in monsoon period, i.e., a discharge decrease in July from DBC and discharge increase from LOCI. Besides, there is a drop in evaporation in September from DBC, which cannot be seen from LOCI. The peak flow in the middle of the century is slightly shifted to be early in July from LOCI, while this change cannot be seen in the results from DBC. It indicates that the uncertainty of future hydrological change comes not only from GCMs but also from the two bias correction methods. Furthermore, the Beas river basin is very likely to become warmer and wetter in both the early and late future, although large uncertainties in the study of future water under climate change can be seen.

**ACKNOWLEDGMENTS**

This study was jointly funded by the Research Council of Norway (RCN) project 216576 (NORINDIA), project-JOINTINDNOR 203867, project GLACINDIA 033L164 and project EVOGLAC 255049.

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

Table 1. Daily GSM-WASMOD equations and parameters

| Variable controlled | Parameter (units) | Equation | |
|---|---|---|---|
| **WASMOD-D module** | | | |
| Snow fall | $a_1, a_2$ ($^{\circ}C$) | $s_t = p_t \left\{1 - \exp\left(-\left(\{(T_a - a_1)/(a_1 - a_2)\}^{-}\right)^2\right)\right\}^{+}$ | (1) |
| Rainfall | | $r_t = p_t - s_t$ | (2) |
| Snow storage | | $sp_t = sp_{t-1} + s_t - m_t$ | (3) |
| Snow melt | | $m_t = sp_t \cdot \left\{1 - \exp\left(-\left(((a_2 - T_a)/(a_1 - a_2))^{-}\right)^2\right)\right\}^{+}$ | (4) |
| Actual evapotranspiration | $a_4(-)$ | $e_t = \min[ep_t(1 - a_4^{w_t/ep_t}), w_t]$ | (5) |
| Available water | | $w_t = r_t + sm_{t-1}^{+}$ | (6) |
| Saturated percentage area | $c_1(-)$ | $sp_t = 1 - e^{-c_1 w_t}$ | (7) |
| Fast flow | | $s_t = (r_t + m_t) \cdot sp_t$ | (8) |
| Slow flow | $c_2$(mm$^{-1}$day) | $f_t = w_t\left(1 - e^{-c_2 w_t}\right)$ | (9) |
| Total flow | | $d_t = s_t + f_t$ | (10) |
| Land moisture | | $sm_t = sm_{t-1} + r_t + m_t - e_t - d_t$ | (11) |
| **Glacier and snow (GSM) module** | | | |
| Glacier and snow mass gain | $T_s(^{\circ}C)$, $\Delta T(K)$ | $G_t = \begin{cases} p_t & \forall T_a \le T_s - \Delta T/2 \\ p_t \cdot [(T_s - T_a)/\Delta T + 0.5] & \forall T_s - \Delta T/2 < T_a < T_s + \Delta T/2 \\ 0 & \forall T_a \ge T_s + \Delta T/2 \end{cases}$ | (12) |
| Glacier and snow mass melt | $DDF$ | $M_{s/f/i} = \max\left(DDF_{s/f/i}\left(T_a - T_0\right), 0\right)$ | (13) |

*where {x}+ means max(x,0) and {x}- means min(x,0); $ep_t$ is the daily potential evapotranspiration; $a_1$ is the snowfall temperature and $a_2$ is the snow melt temperature; $T_a$ is air temperature($^{\circ}C$); $p_t$ is the precipitation in a given day; $sm_{t-1}$ is the land moisture (a available storage; $T_s$ is a threshold temperature for snow distinguishes between rain and snow $T_s = 1$ $^{\circ}C$; $\Delta T$ is a temperature interval, $\Delta T = 2$ K; $DDF_s$, $DDF_f$ and $DDF_i$ are the degree day factor for snow, firn and ice, and $T_0$ is the melt threshold factor in GSM module.*

770

Table 2. Climate change scenarios for Beas river basin at the 21st Century (2046-2065 and 2080-2099)

| Statistical Downscaling | RCP | GCMs | Abbreviation | Description |
|---|---|---|---|---|
| DBC | 4.5 | CamESM2 | CA2 | |
| DBC | 8.5 | CSIRO_Mk3_6_0 | CS0 | Wet&Cold |
| LOCI | 4.5 | CamESM2 | CA2 | |
| LOCI | 8.5 | CSIRO_Mk3_6_0 | CS0 | |
| DBC | 4.5 | Inmcm4 | IN4 | |
| DBC | 8.5 | MRI-ESM1 | MR1 | Dry&Cold |
| LOCI | 4.5 | Inmcm4 | IN4 | |
| LOCI | 8.5 | MRI-ESM1 | MR1 | |
| DBC | 4.5 | IPSL-CM5A-LR | IPR | |
| DBC | 8.5 | IPSL_CM5A_LR | IPR | Dry&Warm |
| LOCI | 4.5 | IPSL-CM5A-LR | IPR | |
| LOCI | 8.5 | IPSL_CM5A_LR | IPR | |
| DBC | 4.5 | MRI_CGCM3 | MR3 | |
| DBC | 8.5 | MIROC5 | MI5 | Wet&Warm |
| LOCI | 4.5 | MRI_CGCM3 | MR3 | |
| LOCI | 8.5 | MIROC5 | MI5 | |

Table 3. The average winter precipitation (DJFM) of WRF and Gauge at different altitudes

| Altitude (m a.m.s.l.) | >2000 | >3000 | >4000 | >4800 | >6000 |
|---|---|---|---|---|---|
| Area (%) | 88% | 62% | 41% | 21% | 1% |
| Gauge | 279.3 | 279.7 | 278.7 | 279.0 | 278.9 |
| WRF | 629.2 | 725.9 | 762.3 | 746.4 | 628.7 |
| WRF/Gauge | 2.25 | 2.59 | 2.74 | 2.67 | 2.25 |

Table 4. Calibration (1986-2000) and validation (1999-2004) of simulated glacier mass balance in Beas river basin comparing with the data from previous studies

| Unit: m w.e. a$^{-1}$ | 1986-2000 | 1999-2004 | Methods |
|---|---|---|---|
| GSM-WASMOD | -0.22 | -1.09 | model |
| Azam et al. (2014) | -0.01(-/+0.36) | / | model |
| Engelhardt et al. (2017) | -0.29 (-/+033) | -0.8(-/+0.33) | model |
| Berthier et al. (2007) | / | -1.02 /-1.12* | Geodetic measurement |
| Vincent et al. (2013) | / | -1.03 (-/+0.44) | Geodetic measurement |

*: from different assumptions*

Table 5. Calibration (1986-2000) and validation (2001-2004) of WASMOD and GSM-WASMOD based on uncorrected and corrected precipitation.

| Model | Precipitation | Calibration (1986-2000) | | | | Validation (2001-2004) | | | |
|---|---|---|---|---|---|---|---|---|---|
| | | NSC_d | NSC_m | VE | RMSE | NSC_d | NSC_m | VE | RMSE |
| WASMOD | Corrected | 0.50 | 0.65 | 5% | 2.40 | 0.31 | 0.36 | 28% | 2.62 |
| GSM-WASMOD | Uncorrected | 0.64 | 0.70 | 8% | 2.03 | 0.49 | 0.52 | 28% | 1.94 |
| GSM-WASMOD | Corrected | 0.65 | 0.75 | 7% | 2.01 | 0.61 | 0.66 | 15% | 1.71 |

*NSC_d: daily Nash-Sutcliffe coefficient; NSC_m: monthly Nash-Sutcliffe coefficient*

Table 6. Annual mean change (including the mean, minimum and maximum values) of main hydrological variables over Beas river basin under future climate comparing with the historical periods.

| Period | RCP | Glacier loss (%)* | dP (%) | dT (·C) | dET (%) | dQ (%) |
|---|---|---|---|---|---|---|
| 2046-2065 | RCP4.5 | 73(63/81) | 9.8(-11.5/29.9) | 1.8(0.8/2.7) | 72.4(36.5/116.6) | 2.6(-19.9/23.9) |
| | RCP8.5 | 81(76/87) | 33.3 (5.3/68.1) | 2.8(2.3/3.8) | 86.7(13.4/161) | 25.3(-6.5/58) |
| 2080-2099 | RCP4.5 | 94(89/99) | 17.7(6.4/39.4) | 2.3(1.2/3.3) | 82(18.7/139.1) | 8.9(-2.2/32.2) |
| | RCP8.5 | 99(93/100) | 39.7(-18.5/89.1) | 5.4(4.2/7.2) | 145(50.9/274.4) | 27(-40.6/84.9) |

*dP: the changes of precipitation; dT: the changes of temperature; dET: the changes of ET; dQ: the changes of runoff*

*\*: Comparing with baseline glacier extent, the future glacier cover loss at the end of 2050 and 2099 in the table, which is respect to 2046-2065 and 2080-2099, respectively.*

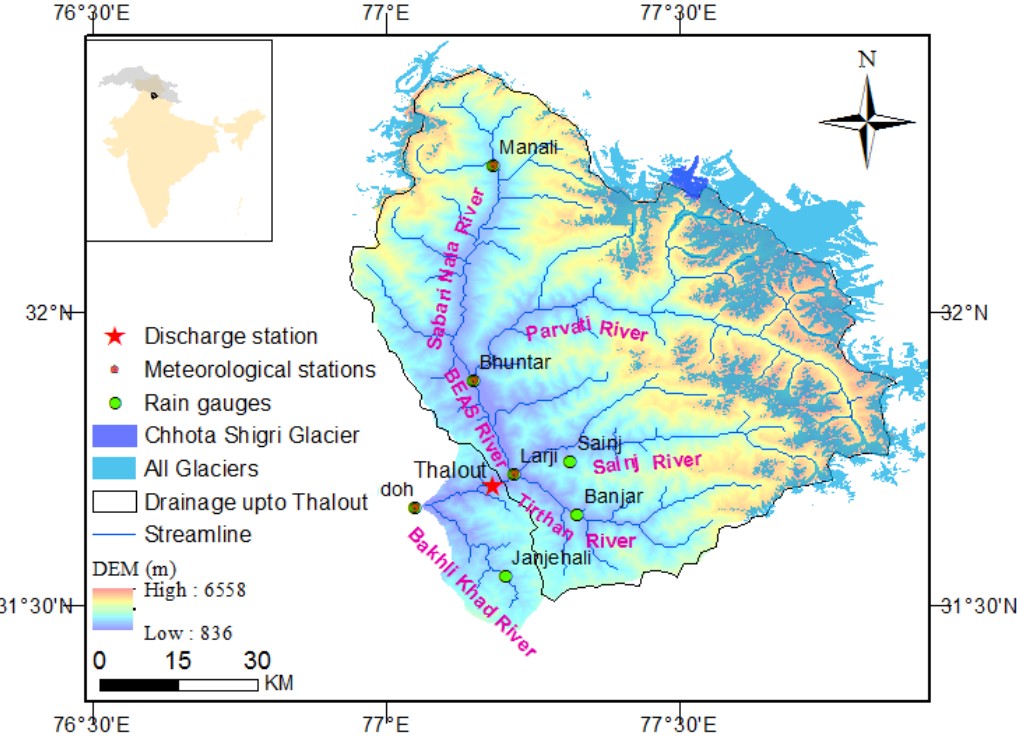

**Fig. 1 The topography, stream network and glacier cover of Beas river basin up to Pandoh dam with seven rain gauges and Thalout discharge station (The small figure on the upper right corner shows the location of Beas river basin up to Pandoh within Upper Indus Basin (UIB) region and India).**

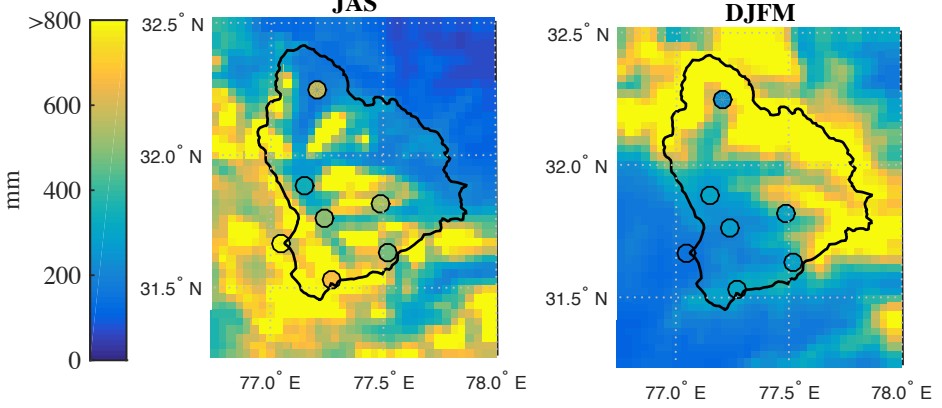

**Fig. 2 Seasonal precipitation (1998-2005) from 3km WRF (from Li et al., 2017) and Gauge (dot) in Beas River basin.**

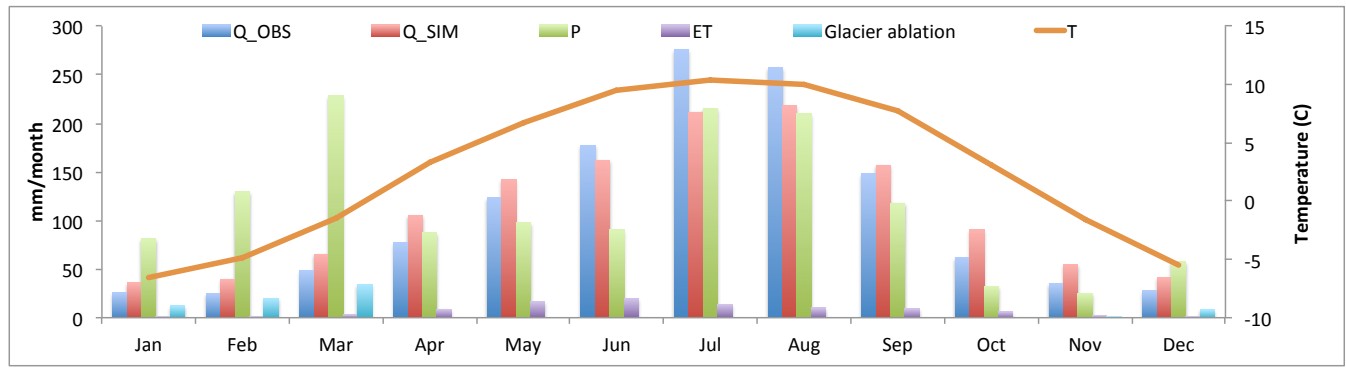

**Fig. 3 Monthly mean of the water balance terms and temperature for the Beas river basin (1986-2004), which shows observed**
**discharge (Q_OBS), simulated discharge (Q_SIM), precipitation (P), evaporation (ET), glacier ablation (in the primary axis on the left side) and temperature (T) (in the secondary axis on right side).**

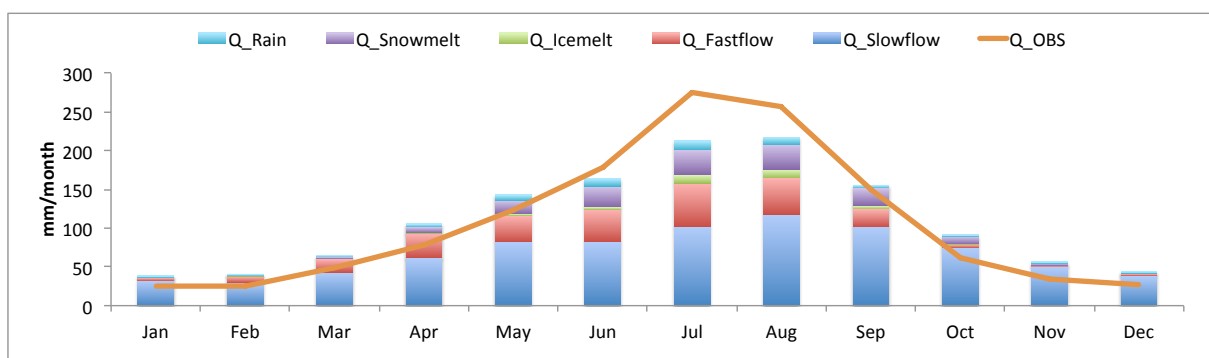

**Fig. 4 the Mean monthly observed discharge (Q_OBS) and the components of simulated discharge in Beas river basin (1990-2004), including fast flow (Q_fastflow), slow flow (Q_slowflow) from non-glacier area and discharges from glacier area, which includes rainfall discharge (Q_Rain), snow-melt (Q_Snowmelt) and ice-melt (Q_icemelt) discharge.**

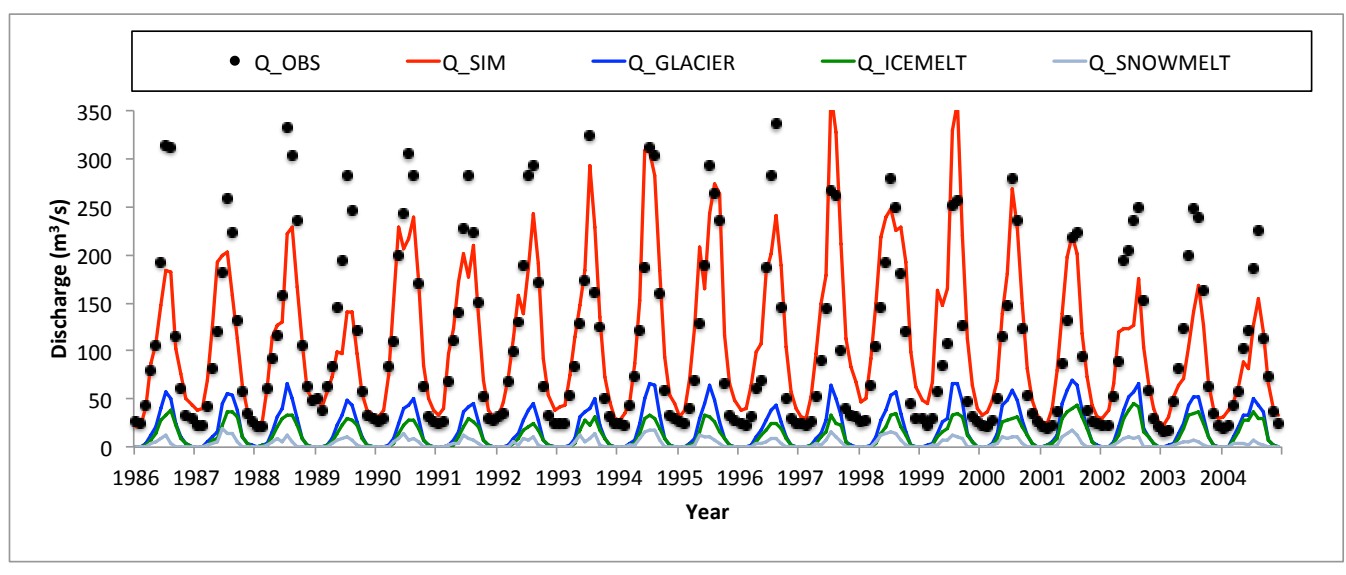

**Fig. 5 Monthly hydrograph of the observed (Q_OBS) and simulated discharge (Q_SIM), total discharge from glacier (Q_GLACIER), ice melting (Q_ICEMELT) and snow melting discharge (Q_SNOWMELT) in Beas river basin during 1986-2004.**

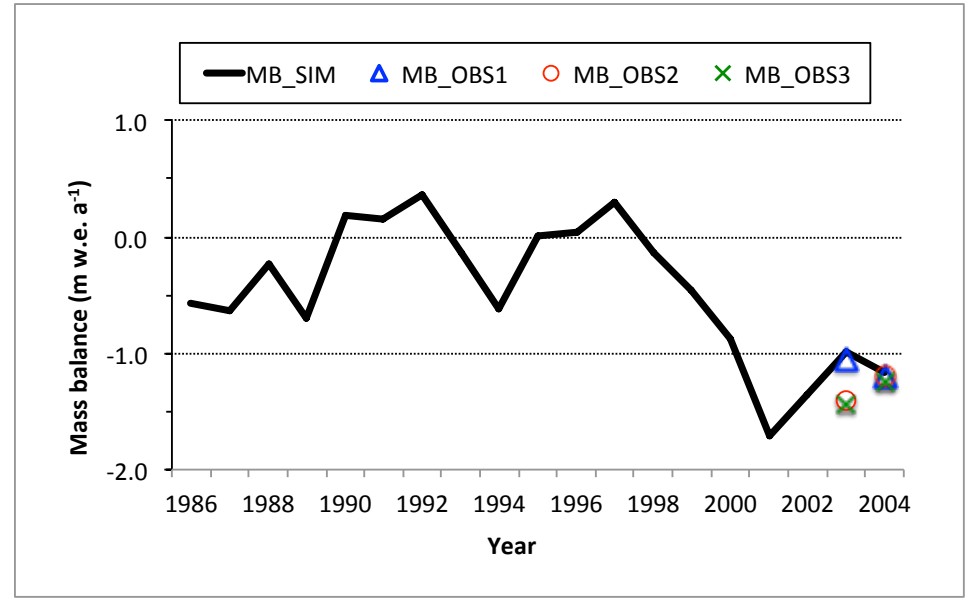

**Fig. 6 The simulated glacier mass balance (MB_OBS) in Beas River basin during 1986-2004 and observed Chhota Shigri glacier mass balance, i.e., MB_OBS1 (Berthier et al., 2007), MB_OBS2 (Wagnon et al., 2007), and MB_OBS3 (Azam et al., 2016).**

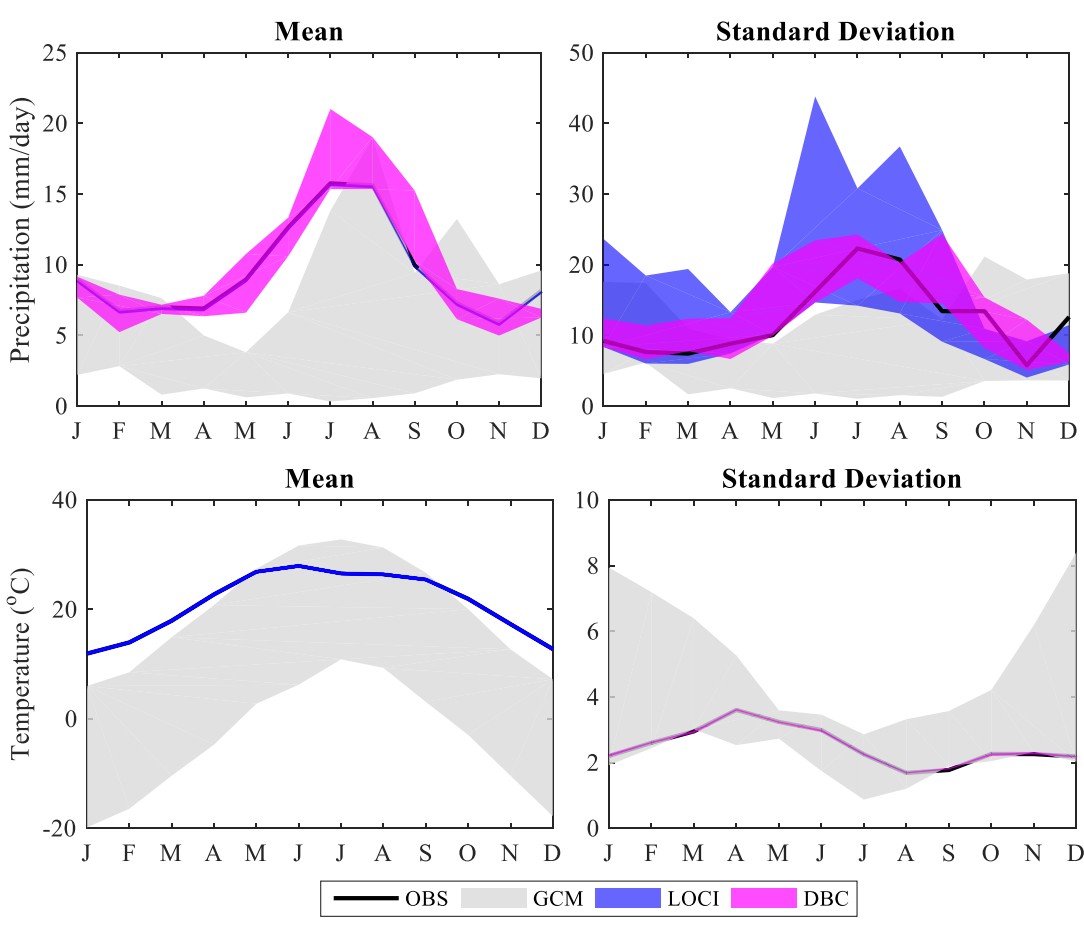

**Fig. 7 Monthly means and standard deviations of daily precipitation (upper panel) and temperature (down panel) from observation (OBS), climate models (GCM) and downscaling of climate models (LOCI and DBC) at Pandoh station in 1986-2005. The envelope represents the results of multiple model ensemble.**

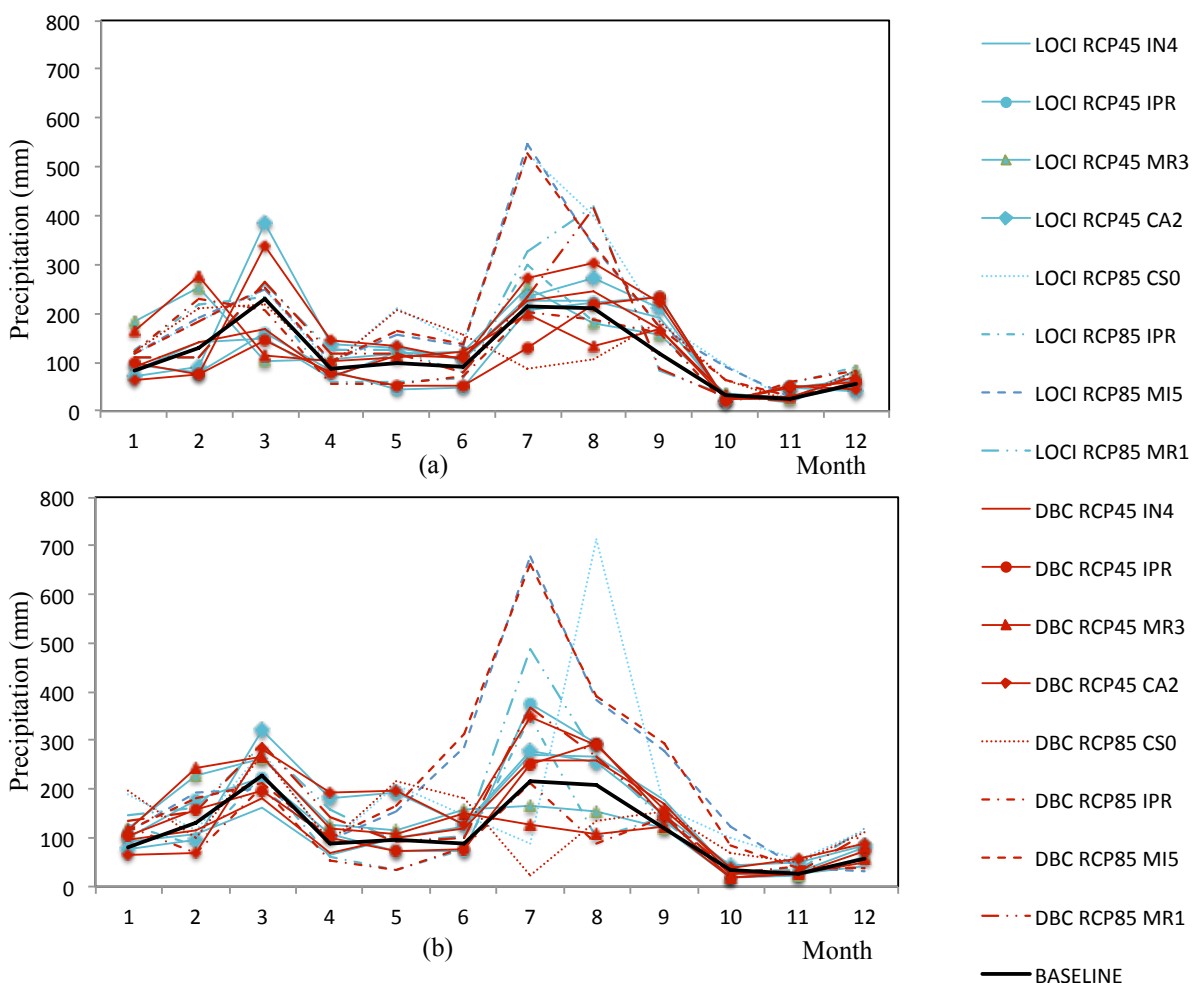

**Fig. 8** Average monthly mean observed (Baseline of 1986-2005) and simulated precipitation based on two bias correction methods under climate change scenarios from two ensembles of four GCMs over Beas river basin during (a) 2046-2065, (b) 2080-2099.

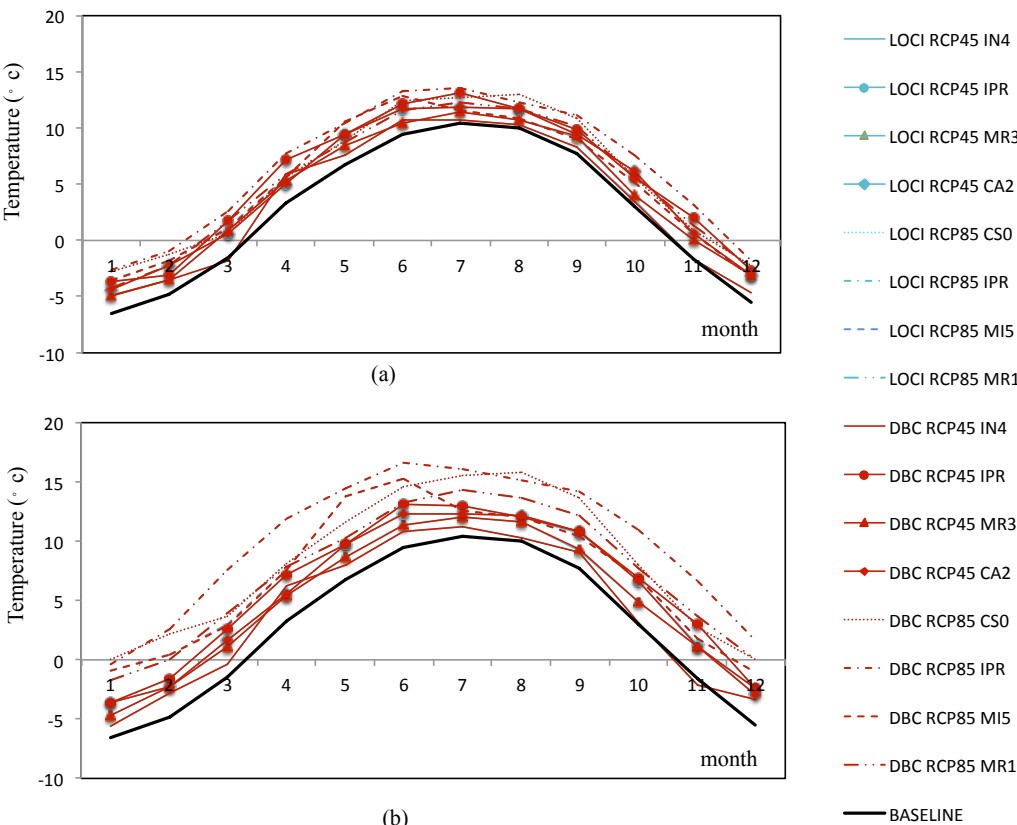

**Fig. 9 Average monthly mean observed (Baseline of 1986-2005) and simulated temperature based on two bias correction methods under climate change scenarios from two ensembles of four GCMs over Beas river basin during (a) 2046-2065, (b) 2080-2099.**

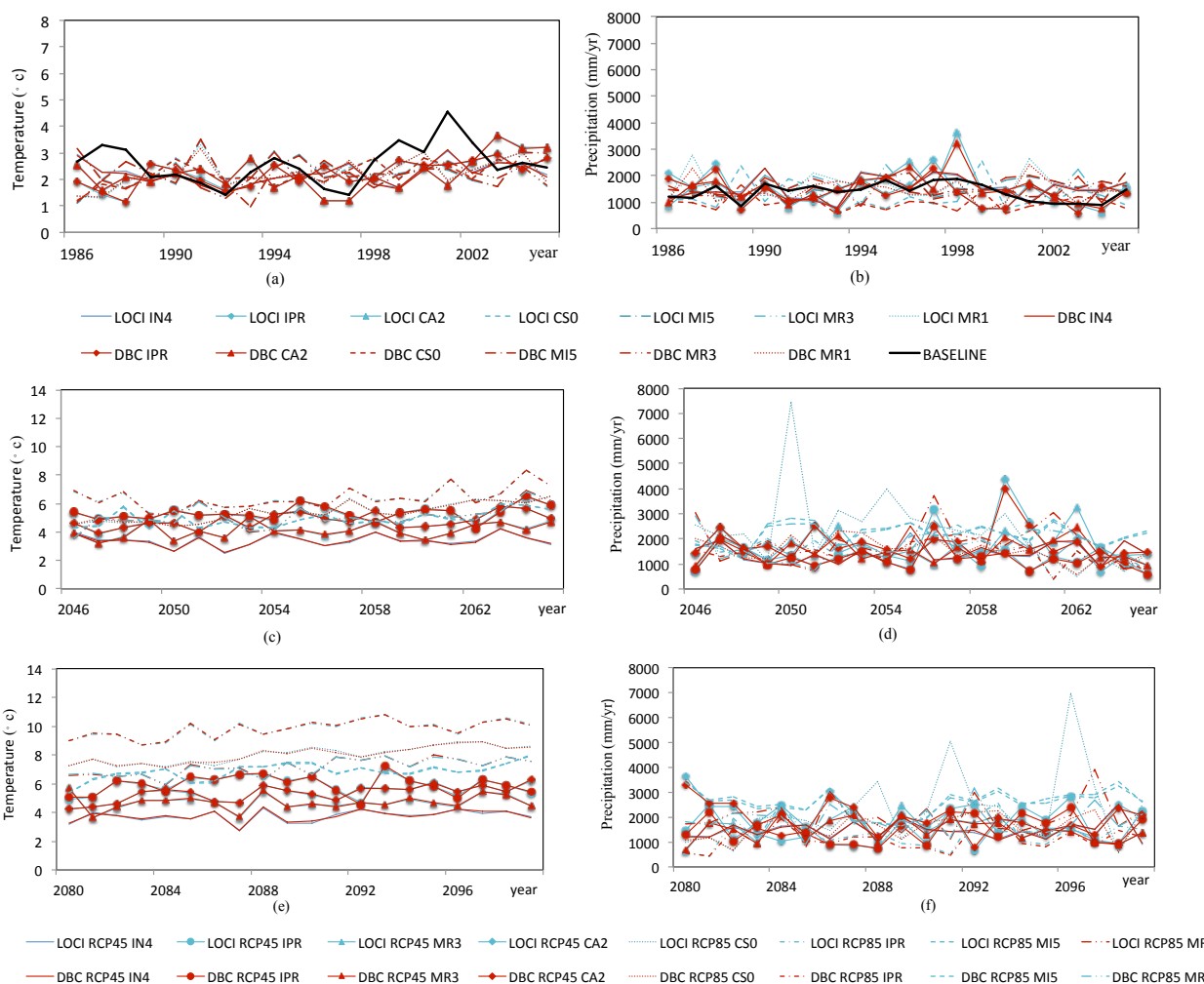

**Fig. 10 The average annual temperature and precipitation based on two bias correction methods under climate change scenarios from two ensembles of four GCMs, including RCP45 and RCP84, over the Beas river basin during 1986-2005, 2046-2065 and 2080-2099.**

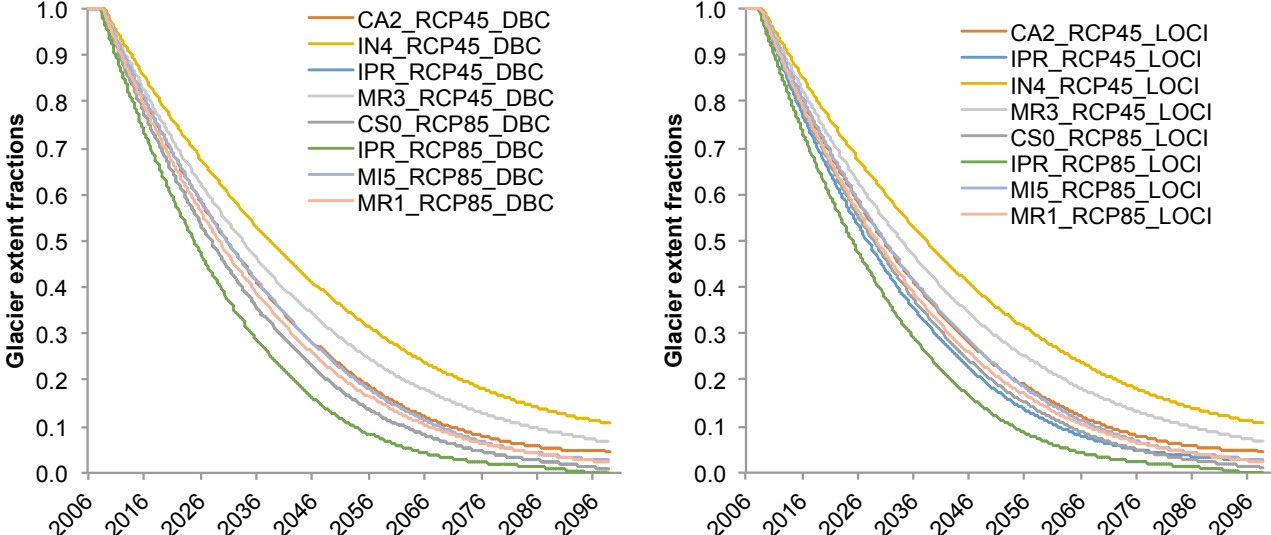

**Fig. 11 Projected changes of glacier extent for Beas river basin during 21st century.**

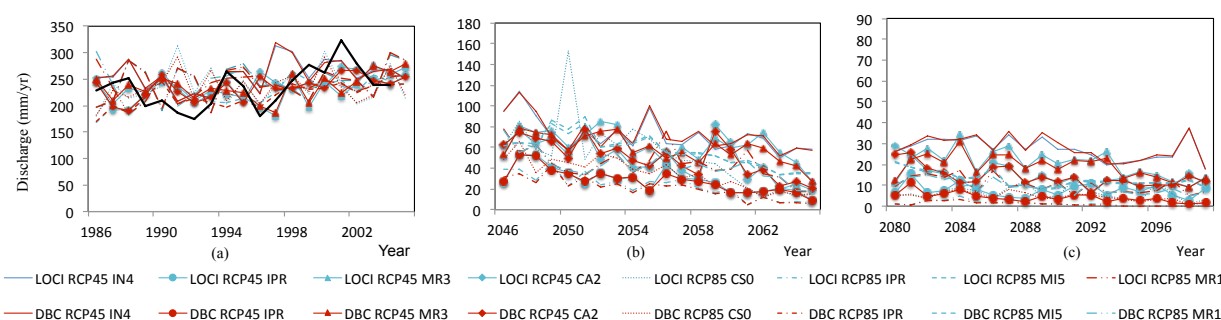

**Fig. 12 Glacier discharge from LOCI and DBC under scenarios from two ensembles of four GCMs including RCP4.5 and RCP8.5 over Beas river basin during (a) 1986-2005, (b) 2046-2065 and (c) 2080-2099. The back line in sub-figure (a) represents the baseline glacier discharge in historical period with the corrected precipitation (please note the scales change of Y-axis in three sub-figures).**

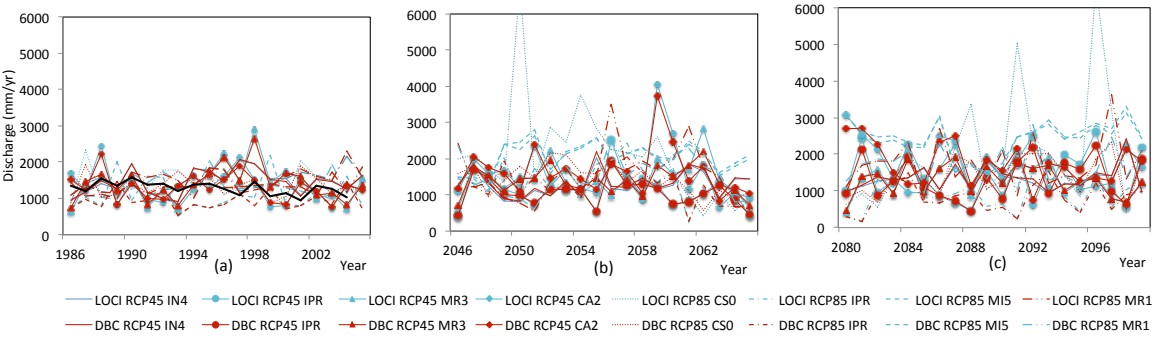

**Fig. 13 Total discharge from LOCI and DBC under scenarios from two ensembles of four GCMs including RCP4.5 and RCP8.5 over Beas river basin during (a) 1986-2005, (b) 2046-2065 and (c) 2080-2099. The back line in sub-figure (a) represents the baseline discharge in historical period with the corrected precipitation.**

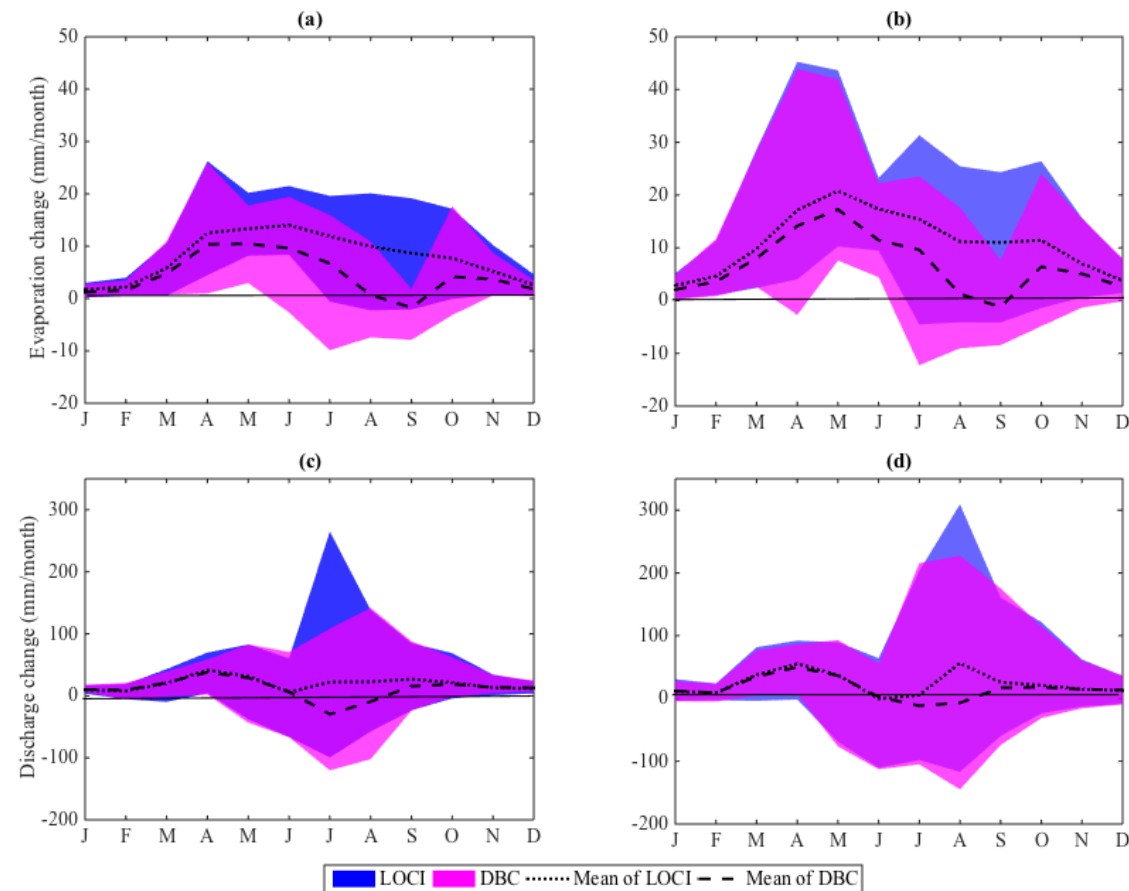

**Fig. 14 Changes of mean monthly evaporation (upper panel) and discharge (down panel) over Beas river basin for the middle of the century (2045-2055) ( left panel) and the end of the century (2080-2099) (right panel) comparing with the baseline period (1986-2005). The envelope represents the results of multiple model ensemble.**

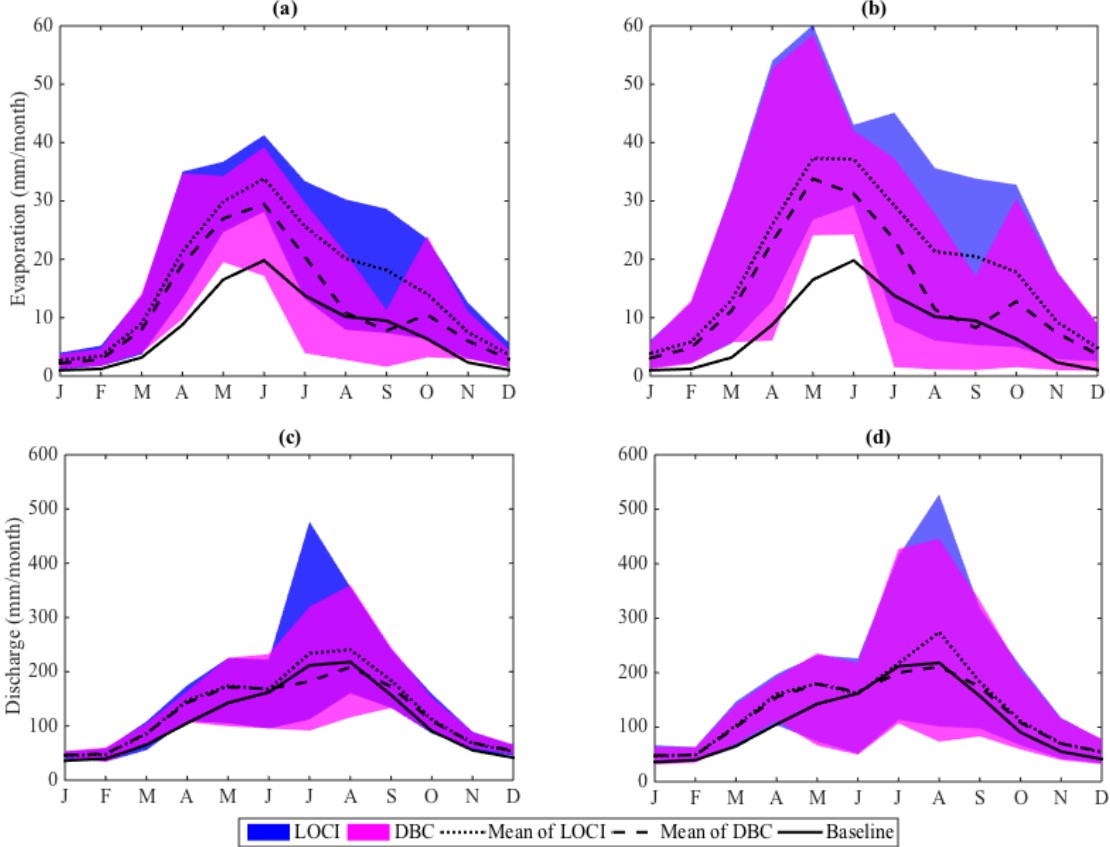

**Fig. 15 The mean monthly evaporation (upper panel) and discharge (down panel) over Beas river basin for the middle of the century (2045-2055) (left panel) and the end of the century (2080-2099) (right panel) comparing with the baseline (1986-2005). The envelope represents the results of multiple model ensemble.**
