# Peer review of "Twenty-first-century glacio-hydrological changes in the Himalayan headwater Beas river basin"

_Hydrology and Earth System Sciences, 2017_

## Short Comment (SC1) · 14 Nov 2017

Dear authors,

This is a very useful study that has been conducted for the data-scarce Himalayan Basin. I have gone meticulously through the paper and I have the following queries:

1) Line 24. The study helps to understand the hydrological impacts of climate change in North India and make a contribution to stakeholders and policymakers with respect to the future of water resources in North India. -However, since only one GCM (BCC_CSM 1.1) is used for the study, how accurate would be the predictions to be

able to be referred by the policymakers? -How is the use of this particular GCM, 'Beijing Climate Center Climate System Model' (BCC_CSM 1.1), justified for use over the Himalayan basin? Please elaborate on this issue.

2) Line 237. Authors should present a figure showing the location of Chhota Shigri glacier in the Beas Basin. Because according to SERB report (Ramanathan, 2011), Chhota Shigri glacier is a part of the Chandra Basin. Chandra basin is a sub-basin of the Chenab river basin according to IndiaWRIS basin maps and the SERB report by Ramanathan (2011).

3) Line 150. Chhota Shigri glacier Area is about 16 Km2 (Ramanathan, 2011), the resolution of the hydrological model GSM-WASMOD is 10*10 Km2.The limitation measured on line 306 also mentions the same thing. However, I feel that the model in the study is too coarse to be able to accurately represent the outflow from the glacier melt. How is such a coarse model justified to be used for representing glacier melt from such small area glaciers and the glacier evolution?

4) Line115. Since the outlet station is Thalout station used for calibration of discharge, I would like to know what is the area of the Beas basin upto Thalout?

Reference: Ramanathan, AL. (2011). Status Report on Chhota Shigri Glacier (Himachal Pradesh), Department of Science and Technology, Ministry of Science and Technology, New Delhi. Himalayan Glaciology Technical Report No.1,pp-88p.
* * *

---

## Referee Comment (RC1) · Anonymous Referee #1 · 21 Nov 2017

**Review of "Projection of future glacier and runoff change in Himalayan headwater Beas basin by using a coupled glacier and hydrological model", by Li et al., HESS**

**General comments**

The manuscript addresses a relevant topic: the impact of climate change for hydrology in Asian high mountain catchments, which supply water to the irrigated areas downstream. Although the assessment is relevant and provides new insights on climate change impacts for water resources in Himalayan catchment, I believe that some major revisions are required to make the work publishable. These are specified in the comments below.

- Section 3.4 on statistical downscaling requires more elaboration. See also the specific comments below.
- It is unclear which precipitation input has been used for the historical period. The paper mention gauge based precipitation, which I assume has been used. In the latter part of the paper the improvement of precipitation forcing using a combination of WRF modelling and gauge data is introduced. It is however unclear if this is used in the study. To me it seems that is was not used although the authors indicate that this method yields better precipitation data. If it was not used, I suggest to redo the modelling using this improved precipitation fields.
- The model resolution (10x10 km) seems to coarse to me for hydrological modelling of mountainous areas with large variability over short horizontal distances. I think this coarse resolution is problematic for proper simulation of melt process, which are very much dictated by elevation and lapsing of temperature fields. Besides I believe that routing will be problematic at this resolution. Since the authors mention that they have higher resolution precipitation forcing data (3x3 km), my suggestion would be to setup the whole model at that resolution.
- The two used statistical downscaling techniques yield very different climate projections although they were used to downscale the same GCM. This implies that the quality of the downscaling for at least one of the methods is questionable. Also the sudden jumps in forcing when going from the historical period to the future period are unnatural and would be smooth if the downscaling performed well. My advice is to validate both statistical downscaling methods for a historical period, then use the one that performs best for the remainder of the study, if the performance is sufficient. If the performance of the downscaling method is insufficient, another method should be used. See also the specific comments below
- The paper now has a 'Results and Discussion' section and a 'Discussion' section. This is double and should be restructured as a 'Result and Discussion' section or a separate 'Results' and 'Discussion' section.
- There are many textual errors. Please have the whole text reviewed by a native English speaker before submitting the revised manuscript.

**Specific comments**

L14: Be more specific about 'future water change'. Do you mean changes in water availability, water resources, hydrological regimes, or a combination?

L45-46: What is the message of this sentence? I do not understand the relation here between the importance of glacier melt and the increase in precipitation.

L50: You mention 'few studies' but you cite only one. If there are a few, please cite them all.

L79: You could add citation to (Palazzi et al., 2013), providing an overview of the variation in precipitation estimates in gridded products.

L88-90: Is this referring to the current study or to the cited Li et al., 2017 study?

L95-98: I would expect that you would answer question 3 first, because it also affects the answers to questions 1 and 2.

L99-105: I would not sum the sections but just describe your approach in 2 or 3 sentences: To answer these questions we use … and … etc.

L107: Mention the percentage of the basin area covered by glaciers.

L119: Include also the glacier outline data and glacier mass balance data you used in this section. Also mention the future climate data (the one downscaled GCM).

L120: Add a citation for the Hydro1k dataset

L123: Also show the locations of the stations where temperature and potential evapotranspiration is measured in Figure 1. Are you sure that potential evapotranspiration is measured there, or should this be actual evapotranspiration?

L133: Is there a specific reason you used the GLIMS dataset and not the more recent Randolph Glacier Inventory (Arendt and 87 others, 2015)? Did you do any quality control of the GLIMS data over your basin? Given that your basin is not that large, it may be worthwhile to do your own mapping of glacier outlines using remote sensing data, if the quality of GLIMS or RGI are insufficient over your basin.

L137: What is the assumption of 20% based on?

L142: How where these percentages for adjustments of the DDFs obtained? Has debris cover been considered in your modelling? If debris cover is present on glaciers in your basin, this will have very different melt properties (e.g. Vincent et al. 2016).

L130-144: This section requires more elaboration of the description of the GSM. Did you calculate for different elevation zones and use vertical temperature lapse rate? Or is the same elevation, temperature and precipitation used for all glaciers? If this module was used before, please provide a reference. Otherwise it will be better to write out the equations listed in Table 1 in the main text and complete describe the model.

L148: A spatial resolution of 10 x 10 km sounds very coarse to me considering the size of the basin. I don't think you can get a sufficient representation of the changes in meteorological variables, which vary strongly over short distances in the mountains. Besides, this resolution is probably problematic to do a proper routing.

L150: Simply interpolation air temperature horizontally will not be sufficient for terrain with strong vertical differences. I advise to use a vertical temperature lapse rate to downscale the temperature field to the elevations of your grid cells.

L164: Include reference to the paper that describes the glacier change parameterization (Lutz et al., 2013).

L181: I do not understand the acronym MLR for linear regression. Where does the 'M' stand for?

L185: the variables 'u', 'w' and 'W' need to be described. Otherwise this part is completely unclear. A few lines of description in addition to the two equations would also be useful.

L188: I would not state 'superior ability of simulation' for a method that does well in transforming changes in the mean, but not the standard deviation and extremes, as stated in L181-182.

L195 'l' and 'i' need to be explained.

L198: 'T' needs to be explained

L199: What is meant by 'the parameters'. What kind of parameters?

L201: subscript 'j' needs to be explained

L213: Should 'station-scale hydrological data' be 'station-scale meteorological data'? I cannot imagine that the downscaling is done with hydrological data.

L213-215: Which GCMs are used?

L221-222: Do I understand correctly that the observed glacier mass balance at one glacier was used for calibration? Can you elaborate on the assumption that this one glacier is representative for all glaciers in the catchment? You could also compare with remote sensing data (Brun et al., 2017; Gardelle et al., 2013) to see how large the spatial variation in glacier mass balance is in your catchment, to say something about the representativeness of Chhota Shigri.

L233: The biases are seem to be large for June-August because of the common problem of underestimated high-altitude precipitation in gauge-based data. If you did not use the improved precipitation fields based on WRF which you discuss in section 5.1, I believe you should include a correction for that in your model. It could be an additional parameter that you calibrate in advance, to make sure that the precipitation input is at least higher than the observed discharge. Have a look at for example (Dahri et al., 2016; Immerzeel et al., 2015). I am not saying you should use an approach as in the cited studies, but at least do a correction on the precipitation input to make it more realistic.

L236: Fig 2 and 3 are much repetition and can be combined in one figure. How do slow flow and fast flow relate to rain-runoff? I rain-runoff surface runoff and are slow flow and fast flow both groundwater flow and flow through the soil layer?

L241-243: I think it is a bit misleading to show one of the years where the model has best performance in figure 5 and then conclude that the model 'worked fine' in the study basin. You clearly indicated that there are quite large biases, especially during the high flow season, which is understandable when simulating high mountain hydrology. I would remove figure 5.

L243-246: Similar comment as for L221-222. How representative is the glacier mass balance at Chhota Shigri for your entire catchment? Here you compare the simulated glacier mass balance for the entire catchment to the observed mass balance at one glacier.

Fig.6: Move the legend outside the plot or draw a clear boundary around it. Now it seems that de symbols in the legend are actual plotted values.

Table 4: I do not understand the line below the headers (0, mean, mean, mean). I also do not understand why the column indicating the statistical downscaling method is headed 'RCM'. I also do not understand the meaning of the header 'Glacier – GCM'. I also do not understand what the line at the bottom of the table 'CMIP5: Bcc-csm' should indicate.

Table 4: You used different GCMs to generate future meteorological forcing for the hydrological model than were used for the future glacier projections. Ideally these should be the same since they are part of the same system. However, I understand that you took glacier projections from another study and included them in yours. I agree that it is better to use some glacier projection is stead of none, but you need to describe the disadvantages of the mismatch in future meteorological forcing used for the glacier evolution model and the hydrological model. This should be more elaborated than in line 300-305.

Fig 7: In the caption you mention 'RCMs'. I do not think you can do that because you did not use RCMs in your study, but statistically downscaled GCMs. I suggest to replace 'RCMs' by 'downscaled GCMs'.

L250-254, Fig 7: The two different downscaling methods lead to very different changes. This needs explanation of the underlying reasons. I wonder why the two methods were used. I suggest to validate both downscaling methods for the historical period, select the method that performs best, then use that method for the future projections.

Fig9: For precipitation, there is a large mismatch between the two downscaling methods, even at the start of the future simulation. For SVM they start around 1000 mm/yr whereas for SDSM they start at around 1500 mm/yr. This means that at least one of them shows a large sudden jump going from the historical period to the future period. If the downscaling was done properly, the transition from the end of the historical period to the start of the future period should be smooth.This needs to be addressed.

The difference in temperature projections for the SDSM and SVM method are enormous towards the end of the century. Describe the reasons for this large difference in the manuscript. Since these methods where used to downscale 1 GCM, the quality of the downscaled forcing for at least one of the downscaling approaches is questionable to me.

Table 5: The change in discharge is very negative, although you have positive changes in precipitation. It seems that it can partly be explained by the increase in evapotranspiration and by losing the additional water from the negative glacier mass balance in the future. Nevertheless, it feels to me that the decrease in total runoff cannot be that large when precipitation amounts are increasing. Please provide a check of the simulated water balance components (precipitation, evapotranspiration, ice melt, snow melt, rainfall discharge, fast flow, slow flow, and changes in storages) of the catchment for your reference and future runs in the revised manuscript.

Fig 12: The different lines of the ensemble member are indistinguishable. I suggest to show the ranges as a shaded area with a line for the mean. Since the figure shows all the members from Table 4, I do not understand why all precipitation projections here start around 1000 mm/yr, whereas in Figure 9 the precipitation projections show large difference for the two downscaling methods.

In the caption you mention that the plot shows glacier melt discharge for CanESM2. In Table 4 you indicate that 2 out of the 16 members use glacier projections for CanESM2. How come all the 16 members are shown in Figure 12? This is very unclear. It is also unclear how one could derive the in the caption mentioned tipping point years (2026 and 2052) from the plot.

Glacier melt contributions around 500 mm/yr seem rather high compared to the plot you showed in Figure 2. There the annual sum of the 'Glacier ablation' seems to be much lower (around 250 mm/yr as far as I can estimate). This would also imply a sudden 'jump' going from the historical period to the future period, which is unnatural. This makes the whole story somewhat questionable. To gain confidence about the projections please provide a check of water balance components as indicated in the comment to Table 5.

L306: See comment on the use of 'RCM' at Figure 7

L308: I think you refer here to the 'jump' I point out in my comment about about the glacier melt in Figure 12. I think this is something that needs to be addressed before the projections have sufficient reliability to be published in HESS.

L313-337: This part comes out of the blue. It is unclear if you used the combined precipitation and WRF forcing in this study. If you did not, I suggest you redo the study with this precipitation dataset if it has a better representation of precipitation. If you did, integrate this part then in the manuscript (i.e. the methodology to the Methods section, and the results to the Results and Discussion section).

**Technical comments**

L11: Remove 'the' at the end of the sentence

L13: remove 'the'. 'Climate' shoud be with lower case 'c'

L18: remove 'impact'

L21: Better to reword to: 'This will result in a general decrease in river runoff for all the scenarios.'

L23: I guess you mean 'WRF precipitation projection'

L28: Remove the first 'The'. From here I will stop correcting the redundant use or absence of 'the'. Please have the manuscript checked by a native English speaker. It is advisable to do this before submission for any future manuscripts.

L29: Reword to: 'Hydrological models have been developed and used as a main assessment tool in the Himalayan region to estimate the impacts of climate change for future water resources.'

L35: 'the' should be 'an'. 'GCM' should be 'GCMs'.

L38: Change 'More' to 'An increase in'

L39: Change 'by' to 'according to'

L54: Introduce Regional Climate Models before using the acronym RCM

L83: 'simulations' should be 'simulation'

L95: Reword to 'The main questions we try to answer in this study are:'

L110: add m asl (metres above sea level)

L115: correct 'meteorological'

L137: No parentheses needed here

L141: No parentheses needed here

L167: 'totally' should be 'in total'

L178: Remove 'was' and 'which'

L243: Included 'simulated' between 'The' and 'annual'

General: There are many textual errors. Please have the whole text reviewed by a native English speaker before submitting the revised manuscript. Please do this for future submissions before the initial submission of the manuscript.

**References**

Arendt, A. and 87 others: Randolph Glacier Inventory [5.0]: A Dataset of Global Glacier Outlines, Version 5.0, Global Land Ice Measurements from Space (GLIMS), Boulder, Colorado, USA., 2015.

Brun, F., Berthier, E., Wagnon, P., Kääb, A. and Treichler, D.: A spatially resolved estimate of High Mountain Asia glacier mass balances, 2000-2016, Nat. Geosci., (August), doi:10.1038/ngeo2999, 2017.

Dahri, Z. H., Ludwig, F., Moors, E., Ahmad, B., Khan, A. and Kabat, P.: An appraisal of precipitation distribution in the high-altitude catchments of the Indus basin, Sci. Total Environ., 548–549, 289–306, doi:10.1016/j.scitotenv.2016.01.001, 2016.

Gardelle, J., Berthier, E., Arnaud, Y. and Kääb, a.: Region-wide glacier mass balances over the Pamir-Karakoram-Himalaya during 1999–2011, Cryosph., 7(4), 1263–1286, doi:10.5194/tc-7-1263-2013, 2013.

Immerzeel, W. W., Wanders, N., Lutz, A. F., Shea, J. M. and Bierkens, M. F. P.: Reconciling high-altitude precipitation in the upper Indus basin with glacier mass balances and runoff, Hydrol. Earth Syst. Sci., 19(11), 4673–4687, doi:10.5194/hess-19-4673-2015, 2015.

Lutz, A. F., Immerzeel, W. W., Gobiet, A., Pellicciotti, F. and Bierkens, M. F. P.: Comparison of climate change signals in CMIP3 and CMIP5 multi-model ensembles and implications for Central Asian glaciers, Hydrol. Earth Syst. Sci., 17(9), 3661–3677, doi:10.5194/hess-17-3661-2013, 2013.

Palazzi, E., Hardenberg, J. Von and Provenzale, A.: Precipitation in the Hindu-Kush

Karakoram Himalaya : Observations and future scenarios, J. Geophys. Res. Atmos., 118, 85–100, doi:10.1029/2012JD018697, 2013.

Vincent, C., Wagnon, P., Shea, J. M., Immerzeel, W. W., Kraaijenbrink, P. D. A., Shrestha, D., Soruco, A., Arnaud, Y., Brun, F., Berthier, E. and Sherpa, S.: Reduced melt on debris-covered glaciers: investigations from Changri Nup Glacier, Nepal, Cryosph. Discuss., 1–28, 2016.

---

## Referee Comment (RC2) · Anonymous Referee #2 · 4 Jan 2018

The manuscript by Li et al. investigates the impact of climate change on glacier melt contribution to discharge in a medium-sized catchment in the Indus basin. To this end, a calibrated glacio-hydrological model was driven by statistically downscaled climate projections from one GCM under two GHG concentration scenarios. The simulations build on ensemble projections of glacier extent derived from a previous study by Lutz et al. (2016) who have already provided a more comprehensive assessment for the entire Indus basin. The manuscript mainly reports on model application in a particular basin and generally lacks novelty.

Major comments

**HESSD**

The glacio-hydrological modeling capitalizes on projections of future glacier extent from Lutz et al. (2016). Data derived from the Lutz et al. study should be moved to Materials and Methods and should be separated more clearly from the GSM-WASMOD modeling results obtained in the current study. This concerns section 4.3 including figures 10 and 11.

The manuscript has a poor structure and is more often than not hard to follow. For example, modeling results are presented and superficially discussed in "Results and discussion" which is however followed by a "Discussions" section that in fact introduces a completely new modeling experiment including data, methods, results and discussion. The additional material addresses the issue of uncertainty in precipitation data in high altitudes. This topic is without question relevant for hydrological modeling in the study region, however falls largely out of the scope of manuscript. In the remainder part (section 5.2) this topic is further discussed while a critical discussion of the main results presented in sections 4.2 - 4.4 is largely missing.

It is only mentioned by the end of the results section that only one GCM was downscaled to drive the glacio-hydrological simulations while all previous sections give the impression that a GCM ensemble was used. A plethora of previous studies has shown that GCMs contribute a large share to total uncertainty in simulated hydrological impact and it is consequently common practice to drive (an ensemble of) impact models with a GCM ensemble. In this regard, the study clearly falls behind the state of the art and the material does not support significant conclusions.

The manuscript contains a large amount of figures and tables, 21 in total, of which some seem redundant and the authors should make an effort to streamline the material. For example, Table 4 listing all possible combinations of GCM, RCP and method of bias correction is largely identical in content to Table 2.

The standard of English needs to be improved throughout the manuscript. While the meaning is usually (but not always) clear, there are a lot of grammatical errors (far too

many to list) and diction is often poor.

Specific comments

L. 11: Why would the glacier melt lead to extreme rainfall?

L. 13: I strongly disagree with the use of the term RCM when referring to the two methods of GCM bias correction/downscaling applied in this study. The term RCM describes numerical prediction models.

L. 30-32: Colloquial, please rephrase.

L. 36: Please correct to "CMIP5"

L. 67: Correct to "Mishra 2015"

L. 88: Unclear, please rephrase.

L. 115-117 : This section describes the study basin/region; information on the model and data used should be moved to the corresponding sections.

L. 115: Please correct to "meteorological"

L. 130: Was the GSM module developed in the scope of this study? If not, please add the reference to the original publication.

L. 148: What was the reason for choosing a modeling resolution of 10 km? Most of the input data sets do seem to support a higher modeling resolution; please clarify.

L. 149: It was mentioned earlier that potential evaporation was only available from one station. Were these station values used for the entire basin? Please clarify.

L. 155-156: Unclear, please rephrase.

Section 3.4: 1) The authors miss to describe and reference the 21st century GCM ensemble data used in the study. Please add a section or paragraph. 2) Lutz et al. (2016) applied the same GCM ensemble but a different downscaling approach

to simulate the future glacier extent used in this study. Why did the authors choose a different downscaling technique? Given that the downscaling technique is found to have a profound effect on projected precipitation and temperature (which drive both the simulated glacier extent and melt), how does this inconsistency affect the results for the Beas river basin and the conclusions drawn? 4) Sections 3.4.1 and 3.4.2 need to be rewritten to enhance comprehensibility. In the current version, it is impossible to understand how both downscaling approaches work.

L. 209-215: "SSVM is directly used to construct the relationship between hydrological data and atmospheric variables" and "The calibration of downscaling models used the station-scale hydrological data and GCM historical atmospheric variables to construct the relationship": I understood from the earlier text that both techniques were used the downscale GCM simulated atmospheric variables to station-scale meteorological data which subsequently were used to drive the glacio-hydrological simulation. Did the authors establish a direct statistical relationship between atmospheric variables and hydrological fluxes? Please clarify.

Section 3.5: 1) In L. 220, Li et al. 2013a or Li et al. 2013b? 2) Glacier mass balance data were apparently used for calibration, but this data-set has not been described or mentioned yet. Please add a description to the data section. 3) The efficiency criteria listed seem to refer to simulated discharge only. How was model efficiency evaluated with respect to glacier mass balance? 4) Were discharge and glacier mass balance calibrated simultaneously?

L. 242 "worked fine": Colloquial, please rephrase. Further, I cannot see how Fig. 5 adds important new information. If its only purpose was to show that the model "worked fine", the figure can be removed.

L. 245: It was mentioned earlier that glacier mass balance data were used to calibrate GSM-WASMOD; are those the same data as used here for validation?

L. 250: Table 4 formally belongs to the methods section and should be referenced

there.

L. 255-265: The two downscaling methods seem to introduce a large uncertainty with respect to future climate in the region. How does this uncertainty compare to the spread between the different GCMs?

L. 294 "It shows that the summer peak of runoff sifts to the other seasons in Beas river basin": Cannot be inferred from the figure.

L. 300 and following: It is mentioned here for the first time that only GCM was down-scaled to drive the glacio-hydrological model. This should have been made clear in the methods section.

Tab. 2: Please rephrase the caption and correct to "glacier evolution"; "Selected model" in the table heading is rather ambiguous and could be replaced by "GCMs"

Table 3: Please correct to "validation", "Nash-Sutcliffe coefficient" and "NS_d" (row 6); typographical error in the last row; missing space before table number.

Table 5: Please provide a more informative caption. I assume ensemble median and range are show. "Change" should be spelled lower case. Does the table show changes over the glacierized area or for the entire river basin?

Fig. 2: In the legend, please correct to "Simulated dis"

Fig. 3: Please add the observed discharge for reference

Fig. 4: Please correct to "Monthly hydrographs". The quality of the Figure should be improved.

Fig. 6: The observed data shown seem to be mean values over certain time periods rather than estimates for a single year (e.g. 1999–2004 in Vincent et al. 2013), but are depicted as points in the figure which is misleading. Please correct. Further, please add a table listing all external glacier MB data including reference period and estimation method.

Fig. 7+8+9: I strongly disagree with the use of the term "RCM" when the authors actually refer to bias correction methods, please correct. Please revise the captions. Do the figures show the ensemble mean? If yes, please add the ensemble range.

Fig. 10: Y-axis label should read "Glacier"

Fig. 11: Is this the ensemble mean?

Fig. 12: The figure needs profound revision. 1) I can only guess that the numbers in the legend refer to the index given in Table 4. Listing all ensemble members in the legend is somewhat obsolete since they are not distinguishable in the plot. 2) The caption claims that results for only one GCM are shown (CANESM2) while the figure apparently shows the whole ensemble. 3) Are both RCPs shown? If yes, please color-code accordingly. 4) In all simulations glacier melt discharge approaches 0 by the end of the century while according to Table 5 glacier cover remains larger than 0. Please explain. 4) Why is glacier-melt discharge given in negative numbers?

Fig. 14: The two subfigures seem to show exactly the same data with respect to the single ensemble members. Please double-check.

---

## Author Comment (AC1) · 9 Mar 2018

Dear authors,

This is a very useful study that has been conducted for the data-scarce Himalayan Basin. I have gone meticulously through the paper and I have the following queries:

1) Line 24. The study helps to understand the hydrological impacts of climate change in North India and make a contribution to stakeholders and policymakers with re- spect to the future of water resources in North India. -However, since only one GCM (BCC_CSM 1.1) is used for the study, how accurate would be the predictions to be able to be referred by the policymakers? -How is the use of this particular GCM, 'Bei- jing Climate Center Climate System Model' (BCC_CSM 1.1), justified for use over the Himalayan basin? Please elaborate on this issue.

- Thanks for your positive evaluation in general and for your professional comment. We agree with it and we will add two ensembles of four GCMs the same as in glacier projections (lutz et al. 2016) for a more comprehensive comparison and uncertainty investigation for the future water cycle and availability in this Himalaya headwater Beas river basin.

2) Line 237. Authors should present a figure showing the location of Chhota Shigri glacier in the Beas Basin. Because according to SERB report (Ramanathan, 2011), Chhota Shigri glacier is a part of the Chandra Basin. Chandra basin is a sub-basin of the Chenab river basin according to IndiaWRIS basin maps and the SERB report by Ramanathan (2011).

- Thanks for the comment and reference. We have corrected it in the manuscript: "The Beas river basin is located in Spiti-Lahaul region, where the available glaciological mass balance series published for comparison are the Chhota Shigri glacier and Bara Shigri glacier (Berithier et al. 2007). The Chhota Shigri glacier is the only one which has been well studied and has detailed and longer period of glacier mass balance data, which also has geodetic mass balance data for validation (Azam et al. 2016). The Chhota Shigri Glacier is a part of the Chandra Basin, which is a sub-basin of the Chenab river basin (Ramanathan, 2011), but it is attached to northeast boundary of Beas river basin, which is close to Manali and Bhunter stations (Fig 1.). In this case, it is used for glacier module calibration in the study, which is to be comparable to the simulation."

  We have also added the location of Chhota Shigri glacier in the Fig. 1 of the revised manuscript. Please see the Fig 2 in the "reply to the RC1".

3) Line 150. Chhota Shigri glacier Area is about 16 Km2 (Ramanathan, 2011), the resolution

of the hydrological model GSM-WASMOD is 10*10 Km2.The limitation mea- sured on line 306 also mentions the same thing. However, I feel that the model in the study is too coarse to be able to accurately represent the outflow from the glacier melt. How is such a coarse model justified to be used for representing glacier melt from such small area glaciers and the glacier evolution?

- Reply: Thanks for the comment. We understand the concern from you. We used mass balance data of Chhota Shigri glacier for comparison with the simulation of the study, because it is the well monitored and studied glacier whose data are available for using. From the revised new Figure 1, we can see that the Chhota Shigri glacier is very small glacier compared with the whole glacier cover in the Beas river basin. In the study, we are looking at the whole glacier extent of Beas river basin and its impact to the total basin runoff, instead of a single Chhota Shigri glacier, which has been done by several previous papers (i.e. Berithier et al. 2007, Azam et al.2016).

  Furthermore, in the replies to the comments of previous two reviewers, we have explained the reason for choosing 10*10km resolution in our original version of the manuscript. Following reviewers' suggestions, we are now re-running the model on 3*3 km resolution and if the results are found to be better, we will replace the old results.

4) Line115. Since the outlet station is Thalout station used for calibration of discharge, I would like to know what is the area of the Beas basin upto Thalout?

- Thanks for the comment and reference. We added the mark of watershed area up to Thalout in the new Fig 1. (please see Fig. 2 in the "reply to the RC1").

Reference: Ramanathan, AL. (2011). Status Report on Chhota Shigri Glacier (Hi- machal Pradesh), Department of Science and Technology, Ministry of Science and Technology, New Delhi. Himalayan Glaciology Technical Report No.1,pp-88p.

---

## Author Comment (AC2) · 9 Mar 2018

**View Letter**

Dear Editor and reviewers:
Many thanks for the review comments that we received with respect to our paper. They have contributed to considerably improving the quality of the manuscript. We have carefully addressed the reviewers' comments and suggestions. Typos will be corrected. Below are our point-by-point responses to each of the reviewer's comments in blue.

**COMMENTS FROM EDITORS AND REVIEWERS**

**Review of "Projection of future glacier and runoff change in Himalayan headwater Beas basin by using a coupled glacier and hydrological model", by Li et al., HESS**

**General comments**

Comment #1:

The manuscript addresses a relevant topic: the impact of climate change for hydrology in Asian high mountain catchments, which supply water to the irrigated areas downstream. Although the assessment is relevant and provides new insights on climate change impacts for water resources in Himalayan catchment, I believe that some major revisions are required to make the work publishable. These are specified in the comments below.

- Reply: We thank to the reviewer for your positive evaluation on our work in general and for your professional and constructive comments detailed below.

Comment #2:

Section 3.4 on statistical downscaling requires more elaboration. See also the specific comments below.

It is unclear which precipitation input has been used for the historical period. The paper mention gauge based precipitation, which I assume has been used. In the latter part of the paper the improvement of precipitation forcing using a combination of WRF modelling and gauge data is introduced. It is however unclear if this is used in the study. To me it seems that is was not used although the authors indicate that this method yields better precipitation data. If it was not used, I suggest to redo the modelling using this improved precipitation fields.

- Reply: thank you for the comment and sorry that we failed to describe this part clear enough in the original version. Yes, the gauge rainfall was used as input for the historical period. The combined WRF and gauge precipitation was only evaluated in the experiment analysis part. In this revised manuscript, we will redo the modeling using the combined precipitation for the

historical period as baseline. In this case, we will split section 5.1 and fill it into three parts: 1) section 2.2 Data, 2) section 3.1 precipitation data correction by WRF and 3) section 4.1 results of model calibration and validation.

Comment #3:

The model resolution (10x10 km) seems to coarse to me for hydrological modelling of mountainous areas with large variability over short horizontal distances. I think this coarse resolution is problematic for proper simulation of melt process, which are very much dictated by elevation and lapsing of temperature fields. Besides I believe that routing will be problematic at this resolution. Since the authors mention that they have higher resolution precipitation forcing data (3x3 km), my suggestion would be to setup the whole model at that resolution.

- Reply: Thanks for the comments. The reason for using 10*10 km resolution in our original version of the manuscript is the following. In our study, we have only seven rain gauge stations and four meteorological stations (i.e. temperature and relative humidity). According to the previous studies and analysis of the influence of interpolation and station density on gridded daily data (i.e. Dirksa et al. 1998; Hofstra et al., 2010; Xu et al., 2013), the results showed that the network density could introduce biases in the mean and variance of the grid values (i.e. precipitation and temperature) compared to those expected for the true area-averages. Especially, it was found by Hiofstra et al. (2010) that both the mean and variance of daily precipitation and temperature are reduced through interpolation unless the network density is extremely high. In this case, we chose 10 km in the study in order to balance the low gauge density of measurements and the resolution of hydrological modeling, which can be fairly reasonable for the simulation. Considering the routing, the routing method in GSM-WASMOD is called NFR routing algorithm (Gong et al. 2009, 2010), which was developed to adapt to the coarse resolution hydrological modeling. This is a scale-independent routing method for network-response function using high-resolution aggregated hydrographgy HYDRO1k. The algorithm preserves the spatially distributed time-delay information in the form of simple network-response functions for any low-resolution grid cell in a large-scale hydrological model. We will add those clarification and citations in the methodology part related to the model resolution chosen and routing method.

  Following reviewer's suggestion, we are now test running the 3km resolution simulation and checking the results. We will replace with the simulation in higher resolution of 3*3 km in the revised manuscript, if it turns out that 3km is truly necessary for getting better result although with limited gauge data.

- Dirks, K.N., Hay, J.E., Stow, C.D. and Harris, D., 1998. High-resolution studies of rainfall on Norfolk Island: Part II: Interpolation of rainfall data. *Journal of Hydrology*, *208*(3-4), pp.187-193.
- Hofstra, N., New, M. & McSweeney, C. Clim Dyn (2010) 35: 841. https://doi.org/10.1007/s00382-009-0698-1
- Xu, H., Xu, C.Y., Chen, H., Zhang, Z. and Li, L., 2013. Assessing the influence of rain gauge density and distribution on hydrological model performance in a humid region of China. *Journal of Hydrology*, *505*, pp.1-12.
- Gong L., E. Widén-Nilsson, S. Halldin, C.-Y. Xu, 2009. Large-scale runoff

routing with an aggregated network-response function. Journal of Hydrology, Volume 368, Issues 1-4, Pages 237-250, doi: 10.1016/j.jhydrol.2009.02.007. Copyright 2009 by Elsevier, reprinted with permission.

Comment #4:

The two used statistical downscaling techniques yield very different climate projections although they were used to downscale the same GCM. This implies that the quality of the downscaling for at least one of the methods is questionable. Also the sudden jumps in forcing when going from the historical period to the future period are unnatural and would be smooth if the downscaling performed well. My advice is to validate both statistical downscaling methods for a historical period, then use the one that performs best for the remainder of the study, if the performance is sufficient. If the performance of the downscaling method is insufficient, another method should be used. See also the specific comments below

- Reply: Thank you for your advice. We are now doing more comprehensive validation, and if one method is found to be significantly better than another, we will use the better one.

Comment #5:

The paper now has a 'Results and Discussion' section and a 'Discussion' section. This is double and should be restructured as a 'Result and Discussion' section or a separate 'Results' and 'Discussion' section.

- Reply: Thanks for the careful review. We have corrected it and there are two parts, i.e. "Results" and "Discussion".

Comment #6:

There are many textual errors. Please have the whole text reviewed by a native English speaker before submitting the revised manuscript.

- Reply: Thank you for the comments.  We will do that.

**Specific comments**

L14: Be more specific about 'future water change'. Do you mean changes in water availability, water resources, hydrological regimes, or a combination?

- Reply: We meant both water availability and hydrological regimes. We will clarify it in the revision.

L45-46: What is the message of this sentence? I do not understand the relation here between the importance of glacier melt and the increase in precipitation.

- Reply: Thanks. We have corrected the sentence to be "The impact of glacier

melt on river flow is noteworthy in the future in the Himalaya region."

L50: You mention 'few studies' but you cite only one. If there are a few, please cite them all.

- Reply: Thank you. Yes, we will add two more citations here (Li et al. 2016; Hasson et al. 2016).
- Li H, Xu C-Y, Beldring S, Tallaksen TM, Jain SK, 2016. Water Resources under Climate Change in Himalayan basins. Water Resources Management 30:843–859. DOI:10.1007/s11269-015-1194-5.
- Hasson, S.U., 2016. Future Water Availability from Hindukush-Karakoram-Himalaya upper Indus Basin under Conflicting Climate Change Scenarios. Climate 2016, Vol. 4, Page 40 4, 40. doi:10.3390/cli4030040

L79: You could add citation to (Palazzi et al., 2013), providing an overview of the variation in precipitation estimates in gridded products.

- Reply: Thanks. We will add the citation in here as: "An overview of the variation in precipitation estimates of gridded products was provided by Palazzi et al. (2013), in which six gridded products are compared with simulation results from a global climate model EC-Earth despite having different resolutions."

L88-90: Is this referring to the current study or to the cited Li et al., 2017 study?

- Reply: This is referring to the current study. We will correct it like this: "… This high-resolution WRF model from Li et al. (2017) provides a first estimation of liquid and solid precipitation in high altitude areas, where satellite and rain gauge networks are not reliable.
  Due to the underestimation of summer precipitation in WRF at the foothill of the basin in its western part, we combined these high-resolution WRF data with gauge data in our study. A solid understanding of the water cycle in the Himalaya headwater basin is interpreted. The comparison of the simulations from gauge precipitation, WRF precipitation and the combined precipitation is further assessed. There is a large uncertainty of winter precipitation over high altitude area in Himalayan river basin. In the study, a glacio-hydrological model has been applied in the Beas river basin in Himalaya for assessing the current and future water resources by statistical downscaling method under climate change scenarios.  …"

L95-98: I would expect that you would answer question 3 first, because it also affects the answers to questions 1 and 2.

- Reply: Thanks for the comment. We agree with reviewer's suggestion. We will change the order as suggested and rewrite: "(1) How much uncertainty is in the precipitation over the ungauged high-altitude in Beas river basin? (2) How will the future water availability change due to higher glacier melt under warmer future in Beas river basin over the Himalaya region? (3) What are the uncertainties of the future water from GCMs or Statistical downscaling methods? "

In this revised manuscript, we will redo the modeling using the combined precipitation for the historical period as baseline.

L99-105: I would not sum the sections but just describe your approach in 2 or 3 sentences: To answer these questions we use ... and ... etc.

- Reply: We will re-write this part as this: "To answer these questions, a combined precipitation from a high-resolution regional climate model, i.e. WRF and gauge data is investigated and used for the hydrological simulation as the historical baseline. In the study, we use a glacio-hydrological model together with two ensembles of four GCMs, i.e. under two generation of scenarios of RCP 4.5 and RCP 8.5, and two statistical downscaling (SD) methods. We firstly focus on the simulation of the present day water cycle and validation of the simulated discharge by using the observed discharge. The uncertainties of the precipitation over high-altitude area and hydrological simulation are discussed. Besides, the future climate change, glacier extent change and hydrological changes will be investigated. At last, the uncertainty from GCMs and statistical downscaling methods will be analyzed and discussed before presenting the main conclusions."

L107: Mention the percentage of the basin area covered by glaciers.

- Reply: Yes, we have added it: "The study area is Beas river basin, upstream of the Pandoh Dam with a drainage area of 5406 $km^2$, out of which 780 $km^2$ (14.4%) is under permanent snow and ice".

L119: Include also the glacier outline data and glacier mass balance data you used in this section. Also mention the future climate data (the one downscaled GCM).

- Reply: Thanks. We will add those information in Data section:
  "The basin boundary in the study is delineated based on HYDRO1k (USGS, 1996a), which is derived from the GTOPO30 30-arc-second global-elevation dataset (USGS, 1996b) and has a spatial resolution of 1 km. HYDRO1k is hydrographically corrected such that local depressions are removed and basin boundaries are consistent with topographic maps. Daily precipitation of 7 rain gauge stations, daily minimum and maximum air temperature of 4 meteorological stations and daily potential evapotranspiration of one station are obtained from Bhakra Beas Management Board (BBMB) in India were used for GSM-WASMOD modelling. The outlet discharge station of Thailout was used for GSM-WASMOD model calibration and evaluation, which was also obtained from the BBMB. Glacier outlines were taken from the recently published Randolph Glacier Inventory (RGI 6.0) (2017) (https://doi.org/10.7265/N5-RGI-60). The annual glacier mass balance data of Chhota Shigri Glacier that are used in the model calibration are taken from the previous studies of Berthier et al. (2007) and Vincent et al. (2013). The Beijing Climate Center Climate System Model (BCC_CSM1.1) developed at the Beijing Climate Center (BCC), China Meteorological Administration (CMA) (Wu et al., 2013) is chosen as the GCM model for use in regional statistical downscaling of future simulations. Furthermore, the daily precipitation from a horizontal 3 km WRF simulation by Li et al. (2017) is also used in the study for further experiment and discussion on the precipitation uncertainty."

L120: Add a citation for the Hydro1k dataset

- Reply: Yes. We have added it. Please see the answer above.

L123: Also show the locations of the stations where temperature and potential evapotranspiration is measured in Figure 1. Are you sure that potential evapotranspiration is measured there, or should this be actual evapotranspiration?

- Reply: Yes, we have added those 4 meteorlgical stations. Please see the new Figure 1 in the reply to L243-246. It is potential evaporation from Pan evaporation.

L133: Is there a specific reason you used the GLIMS dataset and not the more recent Randolph Glacier Inventory (Arendt and 87 others, 2015)? Did you do any quality control of the GLIMS data over your basin? Given that your basin is not that large, it may be worthwhile to do your own mapping of glacier outlines using remote sensing data, if the quality of GLIMS or RGI are insufficient over your basin.

- Reply: Thanks for your comment. We have downloaded the RGI 6.0 and compared it with the data from GLMS. The two glacier shape files have a slight difference but the glacier covered grids are identified the same as that from GLMS. So in this case, it didn't impact the simulation results and conclusions at the end. In the revised manuscript, we have updated the glacier outlines data to be Randolph Glacier Inventory 6.0 in section 3.1: "Those glacier grid cells were defined by ESRI ArcGIS system v. 9.0 (or higher) and set up before modeling based on the glacier outlines from the RGI (6.0) (2017) (https://doi.org/10.7265/N5-RGI-60) (Berthier, 2006; Raup et al., 2007)." Please also see the answer above to the comment of L119.

L137: What is the assumption of 20% based on?

- Reply: This is an empirical estimate. We will clarify it in the revision.

L142: How where these percentages for adjustments of the DDFs obtained? Has debris cover been considered in your modelling? If debris cover is present on glaciers in your basin, this will have very different melt properties (e.g. Vincent et al. 2016).

- Reply: This is an empirical estimate. No, the debris cover is not considered in the modeling right now. We will clarify it in the revision.

L130-144: This section requires more elaboration of the description of the GSM. Did you calculate for different elevation zones and use vertical temperature lapse rate? Or is the same elevation, temperature and precipitation used for all glaciers? If this module was used before, please provide a reference. Otherwise it will be better to write out the equations listed in Table 1 in the main text and complete describe the model.

- Reply: The GSM is calculated based on grids. In each grid cell of glacier, the input data (including elevation, temperature and precipitation) are the same. The temperature and precipitation are interpolated from stations by IDW method and the vertical temperature lapse rate is considered in the IDW method for temperature. This GSM has been used before. We added the citation here (Li et al. 2014; England et al. 2012) and have added more description in method section of GSM:

  "A conceptual glacier- and snow- melt module (GSM) (Li et al. 2014; England et al. 2012) was used to compute glacier mass balances and melt-water runoff from the glacier in the study basin, which was only applied to the grid cells of the glacier-covered area. Those glacier grid cells were defined by ESRI ArcGIS system v. 9.0 (or higher) and set up before modeling based on the glacier outlines from the RGI (6.0) (2017) (https://doi.org/10.7265/N5-RGI-60) (Berthier, 2006; Raup et al., 2007). The gridded temperature and precipitation are interpolated based on the station data by Inverse Distance Weighted (IDW) method, in which the vertical temperature lapse rate of −6 °C km−1 is used to downscale the temperature station to the elevations of the grid cells (Kattel et al., 2013). The daily gridded temperature and precipitation were input data for the GSM module, which calculated both snow accumulation and melt-water runoff."

L148: A spatial resolution of 10 x 10 km sounds very coarse to me considering the size of the basin. I don't think you can get a sufficient representation of the changes in meteorological variables, which vary strongly over short distances in the mountains. Besides, this resolution is probably problematic to do a proper routing.

  Reply: Thanks for the professional comments. Our response was presented to the General Comment #3, above. In which we have written, among others, "Following reviewer's suggestion, we are now test running the 3km resolution simulation and checking the results. We will replace with the simulation in higher resolution of 3*3 km in the revised manuscript, if it turns out that 3km is truly necessary for getting better result although with limited gauge data."

L150: Simply interpolation air temperature horizontally will not be sufficient for terrain

with strong vertical differences. I advise to use a vertical temperature lapse rate to downscale the temperature field to the elevations of your grid cells.

- Reply: Thanks for your comment. This is actually what we did in the IDW interpolation for gridded temperature, but we didn't explain it in detail and clear enough. So we have added the information in the methodology section of the revised manuscript: "The gridded temperature and precipitation are interpolated based on the station data by Inverse Distance Weighted (IDW) method, in which the vertical temperature lapse rate of −6 °C km−1 is used to downscale the temperature station to the elevations of the grid cells (Kattel et al., 2013)."

L164: Include reference to the paper that describes the glacier change parameterization (Lutz et al., 2013).

- Reply: We have added it: "The glacier changes are the result of a close interplay of projected changes in temperature and precipitation, which are calculated monthly in the parameterization approach (Lutz et al., 2013)."

L181: I do not understand the acronym MLR for linear regression. Where does the 'M' stand for?

- Reply: Thanks for the careful review. 'M' stands for 'multiple and MLR means the multiple linear regression. We will correct it in the revised manuscript.

L185: the variables 'u', 'w' and 'W' need to be described. Otherwise this part is completely unclear. A few lines of description in addition to the two equations would also be useful.

- Reply: Thank you for the comment. Now we have added the corresponding description in the 3.4.1 section: "$\hat{u}_t^i$ is the i th corresponding climatic predictor on the $t$ th day; $W_t^{sim}$ is the SDSM-simulated probability on the $t$ th day; $w_t^{sim}$ is the simulated precipitation state on the $t$ th day."

L188: I would not state 'superior ability of simulation' for a method that does well in transforming changes in the mean, but not the standard deviation and extremes, as stated in L181-182.

- Reply: Yes, agree. We will correct it to be "One of the advantages of SDSM is the visual, user-friendly interface that does not exist in most of the downscaling models."

L195 'l' and 'i' need to be explained.

- Reply: Thanks. 'l' is the upper limits of the numbers of the sets{X,Y}.'i' is an ordinal number for the vector X and Y. We will correct it in the revised manuscript.

L198: 'T' needs to be explained

- Reply: Thanks. 'T' means the transposition in the calculation. We have added it in the revised manuscript.

L199: What is meant by 'the parameters'. What kind of parameters?

- Reply: Thanks for the careful review. 'the parameters' means in this equation, both 'W' and 'b' can be adjusted to make the equation balanced. We have added the explanation in the revised manuscript: "where $W$ and $b$ are the parameters which determine the shape of hyperplanes $Y$"

L201: subscript 'j' needs to be explained

- Reply: Thanks. 'j' is also an ordinal number like, which makes $X_i$ and $X_j$ independent to each other. We have corrected equation (4) to make it clearer: " $K\left(X_i, X_j\right) = F\left(\phi(X_j), \phi(X_i)\right) \qquad (i, j \in Z, 1 \le i \le l, 1 \le j \le l)$ "

L213: Should 'station-scale hydrological data' be 'station-scale meteorological data'? I cannot imagine that the downscaling is done with hydrological data.

- Reply: Yes, thanks for the carefully review. We have corrected it.

L213-215: Which GCMs are used?

- Reply: We changed it to be: "The calibrated downscaling models are then utilized to predict the future climate change with the BCC-CSM1.1 variables from 2006 to 2099 in the RCP4.5 and RCP8.5 scenarios".

L221-222: Do I understand correctly that the observed glacier mass balance at one glacier was used for calibration? Can you elaborate on the assumption that this one glacier is representative for all glaciers in the catchment? You could also compare with remote sensing data (Brun et al., 2017; Gardelle et al., 2013) to see how large the spatial variation in glacier mass balance is in your catchment, to say something about the representativeness of Chhota Shigri.

- Reply: Thanks for the references and comment. In our study area, the glaciological mass balance series published in Spiti-Lahaul region (where Beas river basin locates) that is available for comparison are the Chhota Shigri glacier and Bara Shigri glacier (Berithier et al. 2007). In which, the only one is long enough to be comparable to our simulation period is the Chhota Shigri glacier (2002-2014), which has geodetic mass balance for validation (Azam et al. 2016). We have compared the mass balance data from previous studies for the Chhota Shigri glacier. In the study of Gardelle et al. (2013), a detailed map of elevation changes during 2000-2011 in Spiti-Lahsul region based on SPOT5 DEM is given, which showed that the changes of the glaciers in the Beas river basin are quite similar as the change in Chhota Shigri glacier in general, although there is variability both in individual glacier and over the region. So we used the mass balance data of Chhota Shigri glcier for representing all the glaciers in our small basin.

  We have added this explanation in the section of Model calibration: "There is an intra-regional variability of individual glacier mass balance in High Mountain Asia (HMA) and less negative mass balance than most other estimates according to the recent study of Brun et al. (2017). From the study, the annual glacier mass balance is $-0.49 +/- 0.2$ m w.e.yr-1 in Spiti-Lahaul region (where Chhota Shigri glacier locates) during 2000-2008 based on ASTER and $0.37 +/- 0.09$ m w.e.yr-1 in Western Himalaya region from RGI Inventory during 2000-2016 based on ASTER. Besides, a detailed map of elevation changes during 2000-2011 in Spiti-Lahsul region based on SPOT5 DEM is given in the study of Gardelle et al. (2013), which showed that the changes of the glaciers in the Beas river basin are quite similar as the changes in Chhota Shigri glacier during 2000-2011 in general, although there is variability both in independent glacier and over the region. Furthermore, in our study basin, the glaciological mass balance series published in Spiti-Lahaul region (of HMA) that is available for comparison are the Chhota Shigri glacier and Bara Shigri glacier (Berithier et al. 2007). In which, the only one is long enough to be comparable to our simulation period is the Chhota Shigri glacier (2002-2014), which has also geodetic mass balance for validation (Azam et al. 2016). So we used the mass balance data of Chhota Shigri glcier as a representant of glaciers in our study basin."

L233: The biases are seem to be large for June-August because of the common problem of underestimated high-altitude precipitation in gauge-based data. If you did not use the improved precipitation fields based on WRF which you discuss in section 5.1, I believe you should include a correction for that in your model. It could be an additional parameter that you calibrate in advance, to make sure that the precipitation input is at least higher than the observed discharge. Have a look at for example (Dahri et al., 2016; Immerzeel et al., 2015). I am not saying you should use an approach as in the cited studies, but at least do a correction on the precipitation input to make it more realistic.

- Reply: Thank you very much for the suggestion. We will use the combined precipitation (from both WRF and Gauge) for historical baseline simulation. We will update it in the revised manuscript of both methodology and results.

L236: Fig 2 and 3 are much repetition and can be combined in one figure. How do slow flow and fast flow relate to rain-runoff? I rain-runoff surface runoff and are slow flow and fast flow both groundwater flow and flow through the soil layer?

- Reply: Thanks for the comment. We will remove Fig.3 and add more information in the results part of the revised manuscript: "In Fig 2, the total discharge includes fast-flow and slow-flow from non-glacier area and discharge from glacier area, which includes rainfall discharge, snow-melt and ice-melt discharge. The fast-flow is generally considered to be surface runoff and the slow-flow refers to groundwater plus water through soil layer. "

L241-243: I think it is a bit misleading to show one of the years where the model has best performance in figure 5 and then conclude that the model 'worked fine' in the study basin. You clearly indicated that there are quite large biases, especially during the high flow season, which is understandable when simulating high mountain hydrology. I would remove figure 5.

- Reply: Thanks for the suggestion. We will remove Fig. 5.

L243-246: Similar comment as for L221-222. How representative is the glacier mass balance at Chhota Shigri for your entire catchment? Here you compare the simulated glacier mass balance for the entire catchment to the observed mass balance at one glacier.

- Reply: Thanks for the comment. Please see our reply above for L221-222. In our study area, the only glaciological mass balance series published are the Chhota Shigri glacier and Bara Shigri glacier (Berithier et al. 2007). In which, the only one is long enough to be comparable to our simulation period is the Chhota Shigri glacier, which has geodetic mass balance for validation (Azam et al. 2016). We have compared the mass balance data from previous studies for the Chhota Shigri glacier. In the study of Gardelle et al. (2013), a detailed map of elevation changes during 2000-2011 in Spiti-Lahsul region based on SPOT5 DEM is given, which showed that the changes of the glaciers in the Beas river basin are quite similar as the change in Chhota Shigri glacier in general, although there is variability both in independent glacier and over the region. So we used the mass balance data of Chhota Shigri glcier for representing all the glaciers in our small basin.
In order to make it clearer in the manuscript. We will rewrite L236-237 and add the location of Chhota Shigri glacier in Fig. 1 in the revised manuscript: "In our study area, the glaciological mass balance series published in Spiti-Lahaul region (where Beas river basin locates) that is available for comparison are the Chhota Shigri glacier and Bara Shigri glacier (Berithier et al. 2007). In which, the only one is long enough to be comparable to our simulation is the Chhota Shigri glacier, which has geodetic mass balance for validation (Azam et al. 2016). The Chhota Shigri Glacier intersect with the northeast boundary of Beas river basin, which is close to Manali and Bhuter."
We will update the Fig. 1 in the manuscript as below:

[Figure]

Fig 1. The topography, stream network and glacier cover of Beas river basin up to Pandoh dam with seven rain gauges and Thalout discharge station (The small figure on the upper right corner shows the location of Beas river basin up to Pandoh within Upper Indus Basin (UIB) region and India).

Fig.6: Move the legend outside the plot or draw a clear boundary around it. Now it seems that de symbols in the legend are actual plotted values.

- Reply: Thanks. We have corrected the figure (see the Fig. 2 below) and will update it in the revised manuscript:

[Figure]

Fig. 2 Simulated and observed (Observed 1 (Azam et al., 2016) and Observed2 (Vincent et al., 2013)) annual glacier mass balance of Beas River basin (mainly from Chhota Shigri) from 1990-2004.

Table 4: I do not understand the line below the headers (0, mean, mean, mean). I

also do not understand why the column indicating the statistical downscaling method is headed 'RCM'. I also do not understand the meaning of the header 'Glacier – GCM'. I also do not understand what the line at the bottom of the table 'CMIP5: Bcc-csm' should indicate.

- • Reply: Thanks for the comment. We have changed the Table 4 in order to make it clearer to read and understand as below:.

Table 1: Climate change scenarios for GSM-WASMOD at 21st century (2006-2099)

| Index | SDs | RCPs | GCMs | The scenarios of Glacier extent * |
|-------|------|------|-----------|-----------------------------------|
| 1 | SDSM | 4.5 | BCC-CSM1.1 | CamESM2_r4i1p1 |
| 2 | SDSM | 8.5 | BCC-CSM1.1 | CSIRO_Mk3_6_0 |
| 3 | SVM | 4.5 | BCC-CSM1.1 | CamESM2_r4i1p1 |
| 4 | SVM | 8.5 | BCC-CSM1.1 | CSIRO_Mk3_6_0 |
| 5 | SDSM | 4.5 | BCC-CSM1.1 | Inmcm4_r1ip1 |
| 6 | SDSM | 8.5 | BCC-CSM1.1 | IPSL_CM5A_LR |
| 7 | SVM | 4.5 | BCC-CSM1.1 | Inmcm4_r1ip1 |
| 8 | SVM | 8.5 | BCC-CSM1.1 | IPSL_CM5A_LR |
| 9 | SDSM | 4.5 | BCC-CSM1.1 | IPSL-CM5A-LR_r3ip1 |
| 10 | SDSM | 8.5 | BCC-CSM1.1 | MIROC5 |
| 11 | SVM | 4.5 | BCC-CSM1.1 | IPSL-CM5A-LR_r3ip1 |
| 12 | SVM | 8.5 | BCC-CSM1.1 | MIROC5 |
| 13 | SDSM | 4.5 | BCC-CSM1.1 | MRI_CGCM3 |
| 14 | SDSM | 8.5 | BCC-CSM1.1 | MRI_ESM_LR |
| 15 | SVM | 4.5 | BCC-CSM1.1 | MRI_CGCM3 |
| 16 | SVM | 8.5 | BCC-CSM1.1 | MRI_ESM_LR |

*: the scenarios of galcier extent are taken from Lutz et al. (2014)

Table 4: You used different GCMs to generate future meteorological forcing for the hydrological model than were used for the future glacier projections. Ideally these should be the same since they are part of the same system. However, I understand that you took glacier projections from another study and included them in yours. I agree that it is better to use some glacier projection is stead of none, but you need to describe the disadvantages of the mismatch in future meteorological forcing used for the glacier evolution model and the hydrological model. This should be more elaborated than in line 300-305.

- • Reply: Thank you for the comment. In order to keep consistence and have a more comprehensive future picture of water availability of Beas river basin, we will add the same two ensembles of four GCMs (lutz et al. 2016) as the future projection of glacier in the study of Lutz et al (2016) and we will add discussion for clarifying this mismatch uncertainty of the future scenarios results.

Fig 7: In the caption you mention 'RCMs'. I do not think you can do that because you did not use RCMs in your study, but statistically downscaled GCMs. I suggest to

replace 'RCMs' by 'downscaled GCMs'.

- Reply: Thanks. We have replaced "RCM" by "Statistical Downscaling methods" in the whole manuscript.

L250-254, Fig 7: The two different downscaling methods lead to very different changes. This needs explanation of the underlying reasons. I wonder why the two methods were used. I suggest to validate both downscaling methods for the historical period, select the method that performs best, then use that method for the future projections.

- Reply: Please see the reply at the beginning. We will do more comprehensive comparison of the two downscaling methods, if one is found to be significantly better than another, we will use the better one.

Fig9: For precipitation, there is a large mismatch between the two downscaling methods, even at the start of the future simulation. For SVM they start around 1000 mm/yr whereas for SDSM they start at around 1500 mm/yr. This means that at least one of them shows a large sudden jump going from the historical period to the future period. If the downscaling was done properly, the transition from the end of the historical period to the start of the future period should be smooth.This needs to be addressed.

- Reply: Thank you very much for the careful review. There are errors in the data loading and we will correct the Figure in the revised manuscript. We will address the transition from the history period to the future period of the statistical downscaling methods with more detailed contexts in the revision.

[Figure]

**Fig. 3  Ten-yrs moving average of annual precipitation and temperature of the Beas river basin (2003-2099).**

The difference in temperature projections for the SDSM and SVM method are enormous towards the end of the century. Describe the reasons for this large

difference in the manuscript. Since these methods where used to downscale 1 GCM, the quality of the downscaled forcing for at least one of the downscaling approaches is questionable to me.

- Reply: The temperature simulation algorithm for SDSM is multiple linear regression. It is different from SVM non-linear simulation method. Their response for the future tendency is also somehow different. In this study, we found the temperature goes higher in the future by SDSM, which may mean that multiple linear regression can reproduce higher climate change in the downscaling (Chen et al., 2011). We are now doing more comprehensive validation of the two statistical downscaling methods, and if one method is found to be significantly better than another, we will use the better one. Besides, another statistical downscaling method Advanced Delta Change (ADC) method (lutz et al. 2016) will be added into SD method comparison.
- Jie C, Brissette F P, Annie P, et al. Overall uncertainty study of the hydrological impacts of climate change for a Canadian watershed [J]. Water Resources Research, 2011, 47(12):1-16.

Table 5: The change in discharge is very negative, although you have positive changes in precipitation. It seems that it can partly be explained by the increase in evapotranspiration and by losing the additional water from the negative glacier mass balance in the future. Nevertheless, it feels to me that the decrease in total runoff cannot be that large when precipitation amounts are increasing. Please provide a check of the simulated water balance components (precipitation, evapotranspiration, ice melt, snow melt, rainfall discharge, fast flow, slow flow, and changes in storages) of the catchment for your reference and future runs in the revised manuscript.

- Reply: Thank you very much for the carefully review. We have checked the results and the water balance is correct. While there is a temperature 'jump' between historical period and future period from the statistical downscaling, which resulted in a higher impact of glacier retreating in the total runoff. The discharge significantly decreases because of the glacier retreating. We have made the correction and will update all the results. Furthermore, in order to limit the uncertainty of a single GCM, in the revised manuscript we will add another two ensembles of four GCMs (lutz et al. 2016) and have a more comprehensive investigation of uncertainty band and future change in hydrological cycle and water availability of Beas river basin.

Fig 12: The different lines of the ensemble member are indistinguishable. I suggest to show the ranges as a shaded area with a line for the mean. Since the figure shows all the members from Table 4, I do not understand why all precipitation projections here start around 1000 mm/yr, whereas in Figure 9 the precipitation projections show large difference for the two downscaling methods.

- Reply: Thank you very much for the suggestion and careful review. There are errors in the data loading and we will correct it in the revised manuscript (see

Fig 7 in this document). We also improved the Fig 12 according to the suggestion.

In the caption you mention that the plot shows glacier melt discharge for CanESM2. In Table 4 you indicate that 2 out of the 16 members use glacier projections for CanESM2. How come all the 16 members are shown in Figure 12? This is very unclear. It is also unclear how one could derive the in the caption mentioned tipping point years (2026 and 2052) from the plot.

- Reply: Thank you very much for the careful review. It was a typo and we have corrected it in the caption: "Fig. 1 Annual precipitation, total discharge and glacier-melt discharge under Climate change (see more information in Table 4) from 2005 to 2099". In the study, CanESM2 is used in 2 out of the 16 members in glacier projections. We have checked this error and will correct it in the revised manuscript.
  Besides, in the revised manuscript, we will add two ensembles of four GCMs (lutz et al. 2016) for a more comprehensive comparison and uncertainty analysis, in which we will explain more clearly in the methodology section.

Glacier melt contributions around 500 mm/yr seem rather high compared to the plot you showed in Figure 2. There the annual sum of the 'Glacier ablation' seems to be much lower (around 250 mm/yr as far as I can estimate). This would also imply a sudden 'jump' going from the historical period to the future period, which is unnatural. This makes the whole story somewhat questionable. To gain confidence about the projections please provide a check of water balance components as indicated in the comment to Table 5.

- Reply: Thank you very much for the careful review. We have carefully checked the results. It is because there is a temperature 'jump' between historical period and future period from the statistical downscaling. We have made the correction and will update all the results in the revised manuscript.

L306: See comment on the use of 'RCM' at Figure 7

- Reply: Yes. We have corrected it to be "statistical downscaling methods".

L308: I think you refer here to the 'jump' I point out in my comment about about the glacier melt in Figure 12. I think this is something that needs to be addressed before the projections have sufficient reliability to be published in HESS.

- Reply: Thank you for the comment. Yes, as we mentioned in the previous reply to the comment of Fig 12, there is a temperature 'jump' between

historical period and future period from the statistical downscaling. We have made the correction and will update all the results. Furthermore, in order to limit the uncertainty of a single GCM, in the revised manuscript we will add two ensembles of four GCMs (lutz et al. 2016) the same as in glacier projections (see Table 4) for a more comprehensive comparison and uncertainty investigation for the future water cycle and availability in this Himalaya headwater Beas river basin.

L313-337: This part comes out of the blue. It is unclear if you used the combined precipitation and WRF forcing in this study. If you did not, I suggest you redo the study with this precipitation dataset if it has a better representation of precipitation. If you did, integrate this part then in the manuscript (i.e. the methodology to the Methods section, and the results to the Results and Discussion section).

- Reply: As we mentioned in the earlier reply, the combined WRF and gauge precipitation is only evaluated in the experiment analysis part. In the revised manuscript, we will redo the modeling using the combined precipitation for the historical period as baseline

**Technical comments**

L11: Remove 'the' at the end of the sentence

L13: remove 'the'. 'Climate' shoud be with lower case 'c'

L18: remove 'impact'

L21: Better to reword to: 'This will result in a general decrease in river runoff for all the scenarios.'

L23: I guess you mean 'WRF precipitation projection'

L28: Remove the first 'The'. From here I will stop correcting the redundant use or absence of 'the'. Please have the manuscript checked by a native English speaker. It is advisable to do this before submission for any future manuscripts.

L29: Reword to: 'Hydrological models have been developed and used as a main assessment tool in the Himalayan region to estimate the impacts of climate change for future water resources.'

L35: 'the' should be 'an'. 'GCM' should be 'GCMs'.

L38: Change 'More' to 'An increase in'

L39: Change 'by' to 'according to'

L54: Introduce Regional Climate Models before using the acronym RCM

L83: 'simulations' should be 'simulation'

L95: Reword to 'The main questions we try to answer in this study are:'

L110: add m asl (metres above sea level)

L115: correct 'meteorological'

L137: No parentheses needed here

L141: No parentheses needed here

L167: 'totally' should be 'in total'

L178: Remove 'was' and 'which'

L243: Included 'simulated' between 'The' and 'annual'

General: There are many textual errors. Please have the whole text reviewed by a native English speaker before submitting the revised manuscript. Please do this for future submissions before the initial submission of the manuscript.

- Reply: Thank you so much for the careful review and correction! We will correct all theses in the manuscript. Besides, we will ask help from our native speaker colleagues and correct the further textual errors in the whole manuscript.

**References**

Arendt, A. and 87 others: Randolph Glacier Inventory [5.0]: A Dataset of Global Glacier Outlines, Version 5.0, Global Land Ice Measurements from Space (GLIMS), Boulder, Colorado, USA., 2015.

Brun, F., Berthier, E., Wagnon, P., Kääb, A. and Treichler, D.: A spatially resolved estimate of High Mountain Asia glacier mass balances, 2000-2016, Nat. Geosci., (August), doi:10.1038/ngeo2999, 2017.

Dahri, Z. H., Ludwig, F., Moors, E., Ahmad, B., Khan, A. and Kabat, P.: An appraisal of precipitation distribution in the high-altitude catchments of the Indus basin, Sci. Total Environ., 548–549, 289–306, doi:10.1016/j.scitotenv.2016.01.001, 2016.

Gardelle, J., Berthier, E., Arnaud, Y. and Kääb, a.: Region-wide glacier mass

balances over the Pamir-Karakoram-Himalaya during 1999–2011, Cryosph., 7(4), 1263–1286, doi:10.5194/tc-7-1263-2013, 2013.

Immerzeel, W. W., Wanders, N., Lutz, A. F., Shea, J. M. and Bierkens, M. F. P.: Reconciling high-altitude precipitation in the upper Indus basin with glacier mass balances and runoff, Hydrol. Earth Syst. Sci., 19(11), 4673–4687, doi:10.5194/hess-19-4673-2015, 2015.

Lutz, A. F., Immerzeel, W. W., Gobiet, A., Pellicciotti, F. and Bierkens, M. F. P.: Comparison of climate change signals in CMIP3 and CMIP5 multi-model ensembles and implications for Central Asian glaciers, Hydrol. Earth Syst. Sci., 17(9), 3661–3677, doi:10.5194/hess-17- 3661-2013, 2013.

Palazzi, E., Hardenberg, J. Von and Provenzale, A.: Precipitation in the Hindu-Kush

Karakoram Himalaya : Observations and future scenarios, J. Geophys. Res. Atmos., 118, 85–100, doi:10.1029/2012JD018697, 2013.

Vincent, C., Wagnon, P., Shea, J. M., Immerzeel, W. W., Kraaijenbrink, P. D. A., Shrestha, D., Soruco, A., Arnaud, Y., Brun, F., Berthier, E. and Sherpa, S.: Reduced melt on debris- covered glaciers: investigations from Changri Nup Glacier, Nepal, Cryosph. Discuss., 1–28, 2016.

- Reply: Thanks a lot for the recommended citations. We will add them in the revised manuscript.

---

## Author Comment (AC3) · 9 Mar 2018

The manuscript by Li et al. investigates the impact of climate change on glacier melt contribution to discharge in a medium-sized catchment in the Indus basin. To this end, a calibrated glacio-hydrological model was driven by statistically downscaled climate projections from one GCM under two GHG concentration scenarios. The simulations build on ensemble projections of glacier extent derived from a previous study by Lutz et al. (2016) who have already provided a more comprehensive assessment for the entire Indus basin. The manuscript mainly reports on model application in a particular basin and generally lacks novelty.

- Reply: Thanks for your comments. Although there have been studies (i.e. Lutz et al. 2016; Li et al., 2016), looked at the hydrological projections in Indus river basin under climate change, but there is no common conclusion from the studies about the water future of western Himalaya region. Both studies reveal that more studies need to be done in the region.
  Furthermore, in Lutz et al. (2016), they used a corrected precipitation dataset (which is based on 0.25 degree gridded dataset APHRODITE) for historical period, because it was found that there is underestimation of precipitation at high-altitude area in Himalaya region (Immerzeel et al., 2015). In our study, we used a high-resolution (which is 3 km) dynamical regional climate model (WRF) precipitation data for correcting the underestimated precipitation (Li et al., 2017). Moreover, we will add two more ensembles of four GCMs the same as in glacier projections of Lutz et al. (2016) for a more comprehensive comparison and uncertainty investigation for the future water cycle and availability in this Himalaya headwater Beas river basin. In the results, the uncertainty partition of hydrological projection from GCMs and statistical downscaling methods will be analyzed. We agree that the structure of our earlier version of the manuscript is not clear enough and made some confusion. We will sort out the whole structure of the manuscript. The combined WRF and gauge data will be moved to the first part of the method and results. We will re-calibrate the model based on the new corrected precipitation. Then we will do the other analysis and future scenarios based on that.

The glacio-hydrological modeling capitalizes on projections of future glacier extent from

Lutz et al. (2016). Data derived from the Lutz et al. study should be moved to Materials and Methods and should be separated more clearly from the GSM-WASMOD modeling results obtained in the current study. This concerns section 4.3 including figures 10 and 11.

- Reply: Thanks for your comments. Yes, we will correct re-structure the manuscript of Data, methods and Results according to the comment.

The manuscript has a poor structure and is more often than not hard to follow. For example, modeling results are presented and superficially discussed in "Results and discussion" which is however followed by a "Discussions" section that in fact introduces a completely new modeling experiment including data, methods, results and discussion. The additional material addresses the issue of uncertainty in precipitation data in high altitudes. This topic is without question relevant for hydrological modeling in the study region, however falls largely out of the scope of manuscript. In the remainder part (section 5.2) this topic is further discussed while a critical discussion of the main results presented in sections 4.2 - 4.4 is largely missing.

It is only mentioned by the end of the results section that only one GCM was down- scaled to drive the glacio-hydrological simulations while all previous sections give the impression that a GCM ensemble was used. A plethora of previous studies has shown that GCMs contribute a large share to total uncertainty in simulated hydrological impact and it is consequently common practice to drive (an ensemble of) impact models with a GCM ensemble. In this regard, the study clearly falls behind the state of the art and the material does not support significant conclusions.

The manuscript contains a large amount of figures and tables, 21 in total, of which some seem redundant and the authors should make an effort to streamline the mate- rial. For example, Table 4 listing all possible combinations of GCM, RCP and method of bias correction is largely identical in content to Table 2.

- Reply: Thanks for the comments. We agree and accept all of them. We feel sorry that failed to present our work well in the original version. As we mentioned above, we agree that the structure of the manuscript is not clear enough and made some confusion. We will re-structure the whole manuscript. Furthermore, we will add two ensembles of four GCMs the same as in glacier projections (lutz et al. 2016) for a more comprehensive comparison and uncertainty investigation for the future water cycle and availability in this Himalaya headwater Beas river basin.
  The rewritten manuscript will follow those three main questions: "(1) How much uncertainty is in the precipitation over the ungauged high-altitude in Beas river basin? (2) How will the future water availability change due to higher glacier melt under warmer future in Beas river basin over the Himalaya region? (3) What are the uncertainties of the future water from GCMs or Statistical downscaling methods? To answer these questions, a combined precipitation from a high-resolution regional climate model, i.e. WRF and gauge data is investigated and used for the hydrological simulation as the historical baseline.
  In the study, we use a glacio-hydrological model together with two ensembles of four GCMs, i.e. under two generation of scenarios of RCP 4.5 and RCP 8.5, and two statistical downscaling (SD) methods. We firstly focus on the simulation of the present day water cycle and validation of the simulated discharge by using the observed discharge. The uncertainties of the precipitation over high-altitude area and

hydrological simulation are discussed. Besides, the future climate change and glacier extent change and hydrological changes will be investigated. At last, the uncertainty from GCMs and statistical downscaling methods will be analyzed and discussed before presenting the main conclusions."

In this revised manuscript, we will remove Fig 3, Fig.5 and Table 2. We also will split section 5.1 and fill into three part: 1) section 2.2 Data, 2) section 3.1 of precipitation data correction by WRF and 3) section 4.1 results of model calibration and validation.

The standard of English needs to be improved throughout the manuscript. While the meaning is usually (but not always) clear, there are a lot of grammatical errors (far too

many to list) and diction is often poor.

- • Reply: Thanks for your comments. We will carefully check the typo and grammatical errors through the whole revised manuscript and a native speaker colleague will also correct it.

Specific comments

L. 11: Why would the glacier melt lead to extreme rainfall?

- • Reply: Thanks for the comment. That is obviously a mistake. We have corrected it: "The changes in glacier melt may lead to droughts as well as extreme floods in the Himalaya basins, which are vulnerable to the hydrological impacts."

L. 13: I strongly disagree with the use of the term RCM when referring to the two methods of GCM bias correction/downscaling applied in this study. The term RCM describes numerical prediction models.

- • Reply: Thanks for the comment. We changed them all in the revised manuscript to be "statistical downscaling methods".

L. 30-32: Colloquial, please rephrase.

L. 36: Please correct to "CMIP5"

L. 67: Correct to "Mishra 2015"

L. 88: Unclear, please rephrase.

L. 115-117 : This section describes the study basin/region; information on the model and data used should be moved to the corresponding sections.

L. 115: Please correct to "meteorological"

L. 130: Was the GSM module developed in the scope of this study? If not, please add the

reference to the original publication.

- Reply: Thanks for the careful review. All above corrections/comments from L.30 - L.130, will be corrected in the revised manuscript.

L. 148: What was the reason for choosing a modeling resolution of 10 km? Most of the input data sets do seem to support a higher modeling resolution; please clarify.

- Reply: Thanks for the comments. The reason for using 10*10 km resolution in our original manuscript was because in our study, we have only 7 rain gauge stations and 4 meteorological stations (for temperature and relative humidity). According to the previous studies and analysis of the influence of interpolation and station density on gridded daily data (Dirksa et al. 1998; Hofstra et al., 2010; Xu et al., 2013), the results showed that the network density could introduce biases in the mean and variance of the grid values (i.e. precipitation and temperature) compared to those expected for the true area-averages. Especially, it was found by Hiofstra et al. (2010) that both the mean and variance of daily precipitation and temperature are reduced through interpolation unless the network density is extremely high. In this case, we chose 10 km in the study in order to balance the low gauge density of measurements and the resolution of hydrological modeling, which can be fairly well for the simulation.

  Following reviewer's suggestion, we are now running the model with 3km resolution and ing the results. If the higher resolution of 3*3 km model run will yield better results although with limited gauge data, we will replace the old results.
- Dirks, K.N., Hay, J.E., Stow, C.D. and Harris, D., 1998. High-resolution studies of rainfall on Norfolk Island: Part II: Interpolation of rainfall data. *Journal of Hydrology*, *208*(3-4), pp.187-193.
- Hofstra, N., New, M. & McSweeney, C. Clim Dyn (2010) 35: 841. https://doi.org/10.1007/s00382-009-0698-1
- Xu, H., Xu, C.Y., Chen, H., Zhang, Z. and Li, L., 2013. Assessing the influence of rain gauge density and distribution on hydrological model performance in a humid region of China. *Journal of Hydrology*, *505*, pp.1-12.

L. 149: It was mentioned earlier that potential evaporation was only available from one station. Were these station values used for the entire basin? Please clarify.

- Reply: The data of pan-evaporation is only available at one station from 1996-2011. In this case, we used combined potential evaporation, which is calculated by the relative humidity (1990-2004) from four meteorological stations and one-year data (2005) from this one pan-evaporation station (which has bias corrected and uniform with previous potential evaporation data). We have added this clarification in the Data section of the revised manuscript.

L. 155-156: Unclear, please rephrase.

- We will add more explanation here in the revised manuscript: "The routing method in GSM-WASMOD is called NFR routing algorithm (Gong et al. 2009, 2010), which was developed to adapt to the coarse resolution hydrological modeling. This is a scale-independent routing method for network-response function using high-resolution aggregated hydrographgy HYDRO1k. The algorithm preserves the

spatially distributed time-delay information in the form of simple network-response functions for any low-resolution grid cell in a large-scale hydrological model."

Section 3.4: 1) The authors miss to describe and reference the 21st century GCM ensemble data used in the study. Please add a section or paragraph.

- Thanks for the comment. We used only one GCM data here and we will add the reference in the section 2.2 of Data: "The version 1.1 of the Beijing Climate Center Climate System Model (BCC_CSM1.1) developed at the Beijing Climate Center (BCC), China Meteorological Administration (CMA) (Wu et al., 2013) is chosen as the GCM model for use in regional statistical downscaling future simulations." Furthermore, we will add two ensembles of four GCMs (lutz et al. 2016) for comparison in the revised manuscript and more clarification will be added in the Data section.

2) Lutz et al. (2016) applied the same GCM ensemble but a different downscaling approach to simulate the future glacier extent used in this study. Why did the authors choose a different downscaling technique? Given that the downscaling technique is found to have a profound effect on projected precipitation and temperature (which drive both the simulated glacier extent and melt), how does this inconsistency affect the results for the Beas river basin and the conclusions drawn?

- Reply: Thanks for the comment. We agree that the different SD methods may result in inconsistency in the simulations. In the revision, we will add the statistical downscaling method: Advanced Delta Change (ADC) in the study of lutz et al. (2016) for keeping the consistency. It is proved that different downscaling methods can cause uncertainty in future precipitation and temperature simulation. In this case, we chose two different statistical downscaling methods in the original study, which are popular and widely used in the world to simulate the different future climate changes in our study. Both of the downscaling methods are regression-based downscaling technique, which belongs to the perfect prognosis (or change factor). These methods construct the regression relationship between the historical large-scale atmospheric factors and site-specific climatic variables (i.e. precipitation and temperature) and then use this relationship to generate future variables. As we mentioned in the earlier reply, we are now doing more comprehensive validation of those two statistical downscaling methods, and if one method is found to be significantly better than another, we will use the better one. We will also analysis the uncertainty from different SD methods. In this case, we can better answer the question here that you concerned about.

4) Sections 3.4.1 and 3.4.2 need to be rewritten to enhance comprehensibility. In the current version, it is impossible to understand how both downscaling approaches work.

- Thanks for the comment. We will rewrite this part and make it more understandable in the revised manuscript.

L. 209-215: "SSVM is directly used to construct the relationship between hydrological data

and atmospheric variables" and "The calibration of downscaling models used the station-scale hydrological data and GCM historical atmospheric variables to construct the relationship": I understood from the earlier text that both techniques were used the downscale GCM simulated atmospheric variables to station-scale meteorological data which subsequently were used to drive the glacio-hydrological simulation. Did the authors establish a direct statistical relationship between atmospheric variables and hydrological fluxes? Please clarify.

- Reply: Thanks for the comment. There is no direct statistical relationship between atmosphere and hydrological fluxes in the method of the study. A direct statistical regression relationship is just established between large-scale atmospheric factors and site-specific climatic variables. The introduction of downscaling method can be found in the previous answer to 'Section 3.4: 2) Lutz et al. (2016) applied…'. Then the generated future climatic variables (i.e. precipitation and temperature) are used as the input of GSM-WASMOD to simulate the future hydrological fluxes. We will add more detailed explanation in the revised manuscript.

Section 3.5: 1) In L. 220, Li et al. 2013a or Li et al. 2013b?

- Thanks for the comment. We have corrected the reference to be Li et al. 2013a.

2) Glacier mass balance data were apparently used for calibration, but this data-set has not been described or mentioned yet. Please add a description to the data section.

- We will add more information about the glacier mass balance data that we used in the section 2.2 of Data: "The annual glacier mass balance data of Chhota Shigri Glacier used in the model calibration are taken from the previous studies of Berthier et al. (2007), Vincent et al. (2013), Azam et al. (2016) and Enghardt et al. (2017)." Besides, we will also add more explain in the section 4.1: "There is an intra-regional variability of individual glacier mass balance in High Mountain Asia (HMA) and less negative mass balance than most other estimates according to the recent study of Brun et al. (2017). From the study, the annual glacier mass balance is -0.49+/-0.2 m w.e.yr-1 in Spiti-Lahaul region (where Chhota Shigri glacier locates) during 2000-2008 based on ASTER and 0.37+/-0.09 m w.e.yr-1 in Western Himalaya region from RGI Inventory during 2000-2016 based on ASTER. Besides, a detailed map of elevation changes during 2000-2011 in Spiti-Lahsul region based on SPOT5 DEM is given in the study of Gardelle et al. (2013), which showed that the changes of the glaciers in the Beas river basin are quite similar as the changes in Chhota Shigri glacier during 2000-2011 in general, although there is variability both in independent glacier and over the region. Furthermore, in our study basin, the glaciological mass balance series published in Spiti-Lahaul region (of HMA) that is available for comparison are the Chhota Shigri glacier and Bara Shigri glacier (Berithier et al. 2007). In which, the only one is long enough to be comparable to our simulation period is the Chhota Shigri glacier (2002-2014), which has also geodetic mass

balance for validation (Azam et al. 2016). So we used the mass balance data of Chhota Shigri glacier as a representation for the glaciers in our small basin."

3) The efficiency criteria listed seem to refer to simulated discharge only. How was model efficiency evaluated with respect to glacier mass balance?

- We manually adjusted the parameters of glacier module according to the annual glacier mass balance. The bias is used for evaluation with respect to glacier mass balance. We will add more explanation in the revised manuscript.

4) Were discharge and glacier mass balance calibrated simultaneously?

- We first 'pre-calibrate' all parameters according to the total discharge. Then we manually adjusted the parameters of glacier module according to the glacier mass balance. At last, we set the glacier module parameters and re-calibrate the other parameters according to discharge data one more time. We will add more clearly explanation in the revised manuscript.

L. 242 "worked fine": Colloquial, please rephrase. Further, I cannot see how Fig. 5 adds important new information. If its only purpose was to show that the model "worked fine", the figure can be removed.

- Thanks for the comment. We have removed Figure5.

L. 245: It was mentioned earlier that glacier mass balance data were used to calibrate GSM-WASMOD; are those the same data as used here for validation?

- No, the mass balance data are only used for calibration. We didn't use it for validation of the model performance.

L. 250: Table 4 formally belongs to the methods section and should be referenced

there.

- Thanks for the comment. We will change it.

L. 255-265: The two downscaling methods seem to introduce a large uncertainty with respect to future climate in the region. How does this uncertainty compare to the spread between the different GCMs?

- We will add two ensembles of four GCMs (Lutz et al., 2016) in the revised manuscript and we will be able answer this question. We will add this analysis in the revised manuscript.

L. 294 "It shows that the summer peak of runoff sifts to the other seasons in Beas river

basin": Cannot be inferred from the figure.

- Thanks for the careful comment. We have removed it.

L. 300 and following: It is mentioned here for the first time that only GCM was down-scaled to drive the glacio-hydrological model. This should have been made clear in the methods section.

- Thanks for the comment. We will correct it and add clearer contexts in the method part.

Tab. 2: Please rephrase the caption and correct to "glacier evolution"; "Selected model" in the table heading is rather ambiguous and could be replaced by "GCMs"

Table 3: Please correct to "validation", "Nash-Sutcliffe coefficient" and "NS_d" (row 6); typographical error in the last row; missing space before table number.

Table 5: Please provide a more informative caption. I assume ensemble median and range are show. "Change" should be spelled lower case. Does the table show changes over the glacierized area or for the entire river basin?

Fig. 2: In the legend, please correct to "Simulated dis"

Fig. 3: Please add the observed discharge for reference

Fig. 4: Please correct to "Monthly hydrographs". The quality of the Figure should be improved.

Fig. 6: The observed data shown seem to be mean values over certain time periods rather than estimates for a single year (e.g. 1999–2004 in Vincent et al. 2013), but are depicted as points in the figure which is misleading. Please correct. Further, please add a table listing all external glacier MB data including reference period and estimation method.

Fig. 7+8+9: I strongly disagree with the use of the term "RCM" when the authors actually refer to bias correction methods, please correct. Please revise the captions. Do the figures show the ensemble mean? If yes, please add the ensemble range.

Fig. 10: Y-axis label should read "Glacier"

Fig. 11: Is this the ensemble mean?

Fig. 12: The figure needs profound revision. 1) I can only guess that the numbers in the legend refer to the index given in Table 4. Listing all ensemble members in the legend is

somewhat obsolete since they are not distinguishable in the plot. 2) The caption claims that results for only one GCM are shown (CANESM2) while the figure apparently shows the whole ensemble. 3) Are both RCPs shown? If yes, please color- code accordingly. 4) In all simulations glacier melt discharge approaches 0 by the end of the century while according to Table 5 glacier cover remains larger than 0. Please explain. 4) Why is glacier-melt discharge given in negative numbers?

Fig. 14: The two subfigures seem to show exactly the same data with respect to the single ensemble members. Please double-check.

- Thank you so much for the careful review and corrections! We will correct them according to the above comments about the Tables and Figures in the revised manuscript. Besides, all the figures and tables will be updated by the new results in the revised manuscript with respect to the adding new GCMs.

---

## Author Response (AR1)

Dear Prof Pechlivanidis,

I, on behalf of my co-authors, would like to thank you, reviewers and Abhishek Sharma for the efforts on the improvement of our manuscript entitled "Twenty-first century glacio-hydrological changes in the Himalayan headwater Beas river basin" (hess-2017-525). These comments are all valuable and very helpful not only for improving this paper but also beneficial for our research in general.

In response to reviewers' comments and suggestions, two ensembles of four GCMs and two new bias correction methods were used for providing the forcing for a glacio-hydrological model at new 3 * 3 km resolution (it was 10*10 in the earlier version). The glacier extent module was also added in revision and re-run with the same meteorological forcing data. All the simulations have been re-run and the results have been all updated. There were certainly a lot more simulations in the revision work and new co-authors were invited in the revision regarding the new work load: Mingxi Shen worked for the new bias correction methods and provided the meteorological forcing data, drafted sections 3.4 and 4.3 and helped with plotting Fig 6 and Fig 7; Arthur F. Lutz worked with glacier extent modelling and provided the glacier extent data, also helped with the manuscript of section 4.5; Jie Chen helped with guiding bias correction methods and manuscript of sections 3.4 and 4.3; Jingjing Li helped with interpolation of forcing data under future scenarios and also helped with plotting Figs. 8, 9 and 10. So the co-authors ranking was adjusted according to their contributions to the new version of the manuscript.

We have carefully followed these comments in making revisions. Changed parts are marked in blue text in the paper, except for language corrections, which are not marked. The point-by-point response to the comments is presented.

It did take a long time and intensive work for the revised manuscript, since more GCMs data were included, new bias correction methods were used, spatial resolution was changed from 10*10 to 3*3 km, all the results have been reproduced, etc. We would like to thank Editor's encouragement, support and patience to our work and extends submission deadline during the revision process. We hope the revised manuscript is to your satisfaction, and of course we are happy to improve it according to further comments if needed.

Looking forward to hearing from you.

With all best wishes

Yours
Lu Li
On behalf of my co-authors

**View Letter**

Dear Editor and reviewers:
Many thanks for the review comments that we received with respect to our paper. They have contributed to considerably improving the quality of the manuscript. We have carefully addressed the reviewers' comments and suggestions. In the revised version, **blue colored** text represents text that has been revised or relocated, including methodology of two statistical downscaling methods, two ensembles of four GCMs and glacier extent projection and all the results (Tables and Figures). Typos have been corrected. Below are our point-by-point responses to each of the reviewer's comments in blue.

**COMMENTS FROM EDITORS AND REVIEWERS**

**Review of "Projection of future glacier and runoff change in Himalayan headwater Beas basin by using a coupled glacier and hydrological model", by Li et al., HESS**

**General comments**

Comment #1:

The manuscript addresses a relevant topic: the impact of climate change for hydrology in Asian high mountain catchments, which supply water to the irrigated areas downstream. Although the assessment is relevant and provides new insights on climate change impacts for water resources in Himalayan catchment, I believe that some major revisions are required to make the work publishable. These are specified in the comments below.

- Reply: We thank to the reviewer for your positive evaluation on our work in general and for your professional and constructive comments detailed below.

Comment #2:

Section 3.4 on statistical downscaling requires more elaboration. See also the specific comments below.

It is unclear which precipitation input has been used for the historical period. The paper mention gauge based precipitation, which I assume has been used. In the latter part of the paper the improvement of precipitation forcing using a combination of WRF modelling and gauge data is introduced. It is however unclear if this is used in the study. To me it seems that is was not used although the authors indicate that this method yields better precipitation data. If it was not used, I suggest to redo the modelling using this improved precipitation fields.

- Reply: thank you for the comment and sorry that we failed to describe this part clear enough in the original version. Yes, the gauge precipitation was used as input for the historical period. The combined WRF and gauge precipitation was only evaluated in the experiment analysis part. In this revised manuscript, we have redone the modeling using the corrected precipitation for the historical period as baseline. In this case, we have split section 5.1 and fill it into three parts: 1) section 2.2 Data, 2) section 3.5 Precipitation correction and 3) section 4.1 Corrected

precipitation and section 4.2 GSM-WASMOD model calibration and validation.

Comment #3:

The model resolution (10x10 km) seems to coarse to me for hydrological modelling of mountainous areas with large variability over short horizontal distances. I think this coarse resolution is problematic for proper simulation of melt process, which are very much dictated by elevation and lapsing of temperature fields. Besides I believe that routing will be problematic at this resolution. Since the authors mention that they have higher resolution precipitation forcing data (3x3 km), my suggestion would be to setup the whole model at that resolution.

- Reply: Thanks for the comments. We have now re-run all the simulations at 3*3 km resolution and found that the results of calibration and validation were not improved comparing with the results from 10*10 km resolution simulations. It was not a surprise because of limited gauge data that we have in the study area. According to the previous studies and analysis of the influence of interpolation and station density on gridded daily data  (i.e. Dirksa et al., 1998; Hofstra et al., 2010; Xu et al., 2013), the results showed that the network density could introduce biases in the mean and variance of the grid values (i.e. precipitation and temperature) compared to those expected for the true area-averages.
  However, concerning the precision of routing, glacier revolution and smooth of discharge graph and 'step change' because of the coarse resolution, we finally decided to use the 3km simulation in the revised manuscript. All the Tables and figures are updated by the new simulation results.
  The routing method in GSM-WASMOD is called NFR routing algorithm (Gong et al., 2009, 2010). We have added those clarification and citations in the methodology part related to the model routing method.

- Dirks, K.N., Hay, J.E., Stow, C.D. and Harris, D., 1998. High-resolution studies of rainfall on Norfolk Island: Part II: Interpolation of rainfall data. *Journal of Hydrology*, *208*(3-4), pp.187-193.
- Hofstra, N., New, M. & McSweeney, C. Clim Dyn (2010) 35: 841. https://doi.org/10.1007/s00382-009-0698-1
- Xu, H., Xu, C.Y., Chen, H., Zhang, Z. and Li, L., 2013. Assessing the influence of rain gauge density and distribution on hydrological model performance in a humid region of China. *Journal of Hydrology*, *505*, pp.1-12.
- Gong L., E. Widén-Nilsson, S. Halldin, C.-Y. Xu, 2009. Large-scale runoff routing with an aggregated network-response function. Journal of Hydrology, Volume 368, Issues 1-4, Pages 237-250, doi: 10.1016/j.jhydrol.2009.02.007. Copyright 2009 by Elsevier, reprinted with permission.

Comment #4:

The two used statistical downscaling techniques yield very different climate projections although they were used to downscale the same GCM. This implies that the quality of the downscaling for at least one of the methods is questionable. Also the sudden jumps in forcing when going from the historical period to the future period are unnatural and would be smooth if the downscaling performed well. My advice is to validate both statistical downscaling methods for a historical period, then use the one that performs best for the remainder of the study, if the performance is sufficient. If the performance of the downscaling method is insufficient, another method should be used. See also the specific comments below

- Reply: Thank you for your advice. After a careful consideration of the disadvantages of perfect prognosis (PP) methods and the advantages of bias correction methods, the downscaling methods of SDSM and SVM used in the original manuscript were replaced by two bias correction methods of DBC and LOCI in the revised manuscript.
  Both SDSM and SVM are regression-based downscaling methods, which involve estimating the

statistical relationship (e.g. linear relationship for SDSM and nonlinear relationship for SVM) between large scale predictors (e.g. vorticity, mean sea-level pressure, geopotential height and relative humidity) and local or site-specific predictands (e.g. precipitation and temperature) using observed climate data. The reliability of a regression-based method relies on relationships between observed daily climate predictors and predictands. However, these relationships are usually weak, especially for daily precipitation. In addition, the regression-based method is usually incapable of downscaling precipitation occurrence and generating proper temporal structure of daily precipitation, which is critical for hydrological simulations. Moreover, the PP downscaling method establishes relationship between predictors and predictands for the historical period and then applies it to future periods. However, this relationship may not hold for the future in a changed climate. This may partly explain why there was a jump between downscaled historical and future precipitation and temperature simulation in our previous manuscript. In particular, the relationship between predictors and predictands established using reanalysis predictors are applied to GCM predictors based on an assumption that reanalysis predictors and GCM predictors are both "perfectly" simulated at the grid scale (Wilby et al., 2002; Dibike and Coulibaly, 2005; Chen et al., 2011a). While reanalysis and GCM data do share some similarities, they are completely independent. Reanalysis data aims at representing the real world, whereas GCMs operate in their own virtual world. This may further result in the jump of precipitation and temperature between historical and future period.

In our revised manuscript, the bias correction methods involve estimating a statistical relationship between a climate model variable (e.g. precipitation) and the same variable of the observations to correct the climate model outputs. The use of bias correction methods is usually considered as reasonable way to achieve physically plausible results for impact studies. Compared to PP methods, bias correction methods are relatively simple to use and negate the prerequisite of a strong relationship between local-scale variables and large scale climate model variables. Previous work indicates that statistical downscaling using GCM precipitation or temperature directly as a predictor performed much better than using other predictors.

We are now using two new statistical downscaling methods (DBC and LOCI) and have added more comprehensive validation in the results section 4.3. All the relevant parts including introduction, methodology and results have been updated in the revised manuscript.

Comment #5:

The paper now has a 'Results and Discussion' section and a 'Discussion' section. This is double and should be restructured as a 'Result and Discussion' section or a separate 'Results' and 'Discussion' section.

- Reply: Thanks for the careful review. We have corrected it and there are two parts, i.e. "Results" and "Discussion".

Comment #6:

There are many textual errors. Please have the whole text reviewed by a native English speaker before submitting the revised manuscript.

- Reply: Thank you for the comments. We have done that.

**Specific comments**

L14: Be more specific about 'future water change'. Do you mean changes in water availability, water resources, hydrological regimes, or a combination?

- Reply: We meant both water availability and hydrological regimes. We have clarified it in the revision.

L45-46: What is the message of this sentence? I do not understand the relation here between the importance of glacier melt and the increase in precipitation.

- Reply: Thanks. We have corrected the sentence to be "The impact of glacier melt on river flow is noteworthy in the future in the Himalaya region."

L50: You mention 'few studies' but you cite only one. If there are a few, please cite them all.

- Reply: Thank you. Yes, we have added two more citations here (Li et al. 2016; Hasson et al. 2016).
- Li H, Xu C-Y, Beldring S, Tallaksen TM, Jain SK, 2016. Water Resources under Climate Change in Himalayan basins. Water Resources Management 30:843–859. DOI:10.1007/s11269-015-1194-5.
- Hasson, S.U., 2016. Future Water Availability from Hindukush-Karakoram-Himalaya upper Indus Basin under Conflicting Climate Change Scenarios. Climate 2016, Vol. 4, Page 40 4, 40. doi:10.3390/cli4030040

L79: You could add citation to (Palazzi et al., 2013), providing an overview of the variation in precipitation estimates in gridded products.

- Reply: Thanks. We have added the citation here as: "An overview of the variation in precipitation estimates of gridded products was provided by Palazzi et al. (2013), in which six gridded products are compared with simulation results from a global climate model EC-Earth despite having different resolutions."

L88-90: Is this referring to the current study or to the cited Li et al., 2017 study?

- Reply: This is referring to the current study. We have corrected it like this:
  "… This high-resolution WRF model from Li et al. (2017) provides a first estimation of liquid and solid precipitation in high altitude areas, where satellite and rain gauge networks are not reliable.

L95-98: I would expect that you would answer question 3 first, because it also affects the answers to questions 1 and 2.

- Reply: We agree with reviewer's suggestion and have changed the order of the questions as suggested: "(1) How much uncertainty is in the precipitation over the ungauged high-altitude in Beas river basin? (2) How will the future water availability change due to higher glacier melt under warmer future in Beas river basin over the Himalaya region? (3) What are the uncertainties of the future water from GCMs or Statistical downscaling methods? "
  In this revised manuscript, we have re-run the modeling using the corrected precipitation for the historical period as baseline.

L99-105: I would not sum the sections but just describe your approach in 2 or 3 sentences: To answer these questions we use ... and ... etc.

- Reply: We have re-written this part as this: "To answer these questions, precipitations from a high-resolution WRF simulation and Gauge are investigated and a corrected precipitation is

used for the hydrological simulation as the historical baseline. In the study, we use a glacio-hydrological model together with eight GCMs under two generation of scenarios, i.e. RCP 4.5 and RCP 8.5 and two statistical downscaling (SD) methods. We firstly focus on the simulation of the present day water cycle and validation of the simulated discharge by using the observed discharge. The uncertainties of the precipitation over high-altitude area and hydrological simulation are further discussed. Besides, the future climate change and glacier extent change and hydrological changes have been investigated. At last, the uncertainty from GCMs and statistical downscaling methods is analyzed and discussed before presenting the main conclusions."

L107: Mention the percentage of the basin area covered by glaciers.

- Reply: Yes, we have added it: "The study area is Beas river basin, upstream of the Pandoh Dam with a drainage area of 5406 km$^2$, out of which 780 km$^2$ (14.4%) is under permanent snow and ice".

L119: Include also the glacier outline data and glacier mass balance data you used in this section. Also mention the future climate data (the one downscaled GCM).

- Reply: Thanks. We have added those information in Data section:
  "The basin boundary in the study is delineated based on HYDRO1k (USGS, 1996a), which is derived from the GTOPO30 30-arc-second global-elevation dataset (USGS, 1996b) and has a spatial resolution of 1 km. HYDRO1k is hydrographically corrected such that local depressions are removed and basin boundaries are consistent with topographic maps. Daily precipitation of 7 rain gauge stations, daily minimum and maximum air temperature of 4 meteorological stations and daily potential evapotranspiration of one station obtained from Bhakra Beas Management Board (BBMB) in India were used for GSM-WASMOD modelling. The outlet discharge station of Thailout was used for GSM-WASMOD model calibration and evaluation, which was also obtained from the BBMB. Glacier outlines were taken from the recently published Randolph Glacier Inventory (RGI 6.0) (2017) (https://doi.org/10.7265/N5-RGI-60). The annual glacier mass balance data of Chhota Shigri Glacier that are used in the model calibration are taken from the previous studies of Berthier et al. (2007) and Vincent et al. (2013). The Beijing Climate Center Climate System Model (BCC_CSM1.1) developed at the Beijing Climate Center (BCC), China Meteorological Administration (CMA) (Wu et al., 2013) is chosen as the GCM model for use in regional statistical downscaling of future simulations. Furthermore, the daily precipitation from a horizontal 3 km WRF simulation by Li et al. (2017) is also used in the study for further experiment and discussion on the precipitation uncertainty."

L120: Add a citation for the Hydro1k dataset

- Reply: Yes. We have added it. Please see the answer above.

L123: Also show the locations of the stations where temperature and potential evapotranspiration is measured in Figure 1. Are you sure that potential evapotranspiration is measured there, or should this be actual evapotranspiration?

- Reply: Yes, we have added those 4 meteorological stations. Please see the new Figure 1 in the reply to L243-246. It is potential evaporation from Pan evaporation.

L133: Is there a specific reason you used the GLIMS dataset and not the more recent Randolph

Glacier Inventory (Arendt and 87 others, 2015)? Did you do any quality control of the GLIMS data over your basin? Given that your basin is not that large, it may be worthwhile to do your own mapping of glacier outlines using remote sensing data, if the quality of GLIMS or RGI are insufficient over your basin.

- Reply: Thanks for your comment. We have downloaded the RGI 6.0 and compared it with the data from GLIMS. The two glacier shape files have a slight difference but the glacier covered grids are identified the same as that from GLIMS. So in this case, it didn't impact the simulation results and conclusions at the end. In the revised manuscript, we have updated the glacier outlines data to be Randolph Glacier Inventory 6.0 in section 3.1: "Those glacier grid cells were defined by ESRI ArcGIS system v. 9.0 (or higher) and set up before modeling based on the glacier outlines from the RGI (6.0) (2017) (https://doi.org/10.7265/N5-RGI-60) (Berthier, 2006; Raup et al., 2007)." Please also see the answer above to the comment of L119.

L137: What is the assumption of 20% based on?

- Reply: We have clarified that this is an empirical estimate.

L142: How where these percentages for adjustments of the DDFs obtained? Has debris cover been considered in your modelling? If debris cover is present on glaciers in your basin, this will have very different melt properties (e.g. Vincent et al. 2016).

- Reply: This is an empirical estimate and we have added citations in the revised manuscript. No, the debris cover is not considered in the modeling right now. We have clarified it in the revision.

L130-144: This section requires more elaboration of the description of the GSM. Did you calculate for different elevation zones and use vertical temperature lapse rate? Or is the same elevation, temperature and precipitation used for all glaciers? If this module was used before, please provide a reference. Otherwise it will be better to write out the equations listed in Table 1 in the main text and complete describe the model.

- Reply: The GSM is calculated based on grids. In each grid cell of glacier, the input data (including elevation, temperature and precipitation) are the same. The temperature and precipitation are interpolated from stations by IDW method and the vertical temperature lapse rate is considered in the IDW method for temperature. This GSM has been used before. We added the citation here (Li et al. 2014; Engelhardt et al. 2012) and have added more description in method section of GSM:

  "A conceptual glacier- and snow- melt module (GSM) (Li et al. 2014; England et al. 2012) was used to compute glacier mass balances and melt-water runoff from the glacier in the study basin, which was only applied to the grid cells of the glacier-covered area. Those glacier grid cells were defined by ESRI ArcGIS system v. 9.0 (or higher) and set up before modeling based on the glacier outlines from the RGI (6.0) (2017) (https://doi.org/10.7265/N5-RGI-60) (Berthier, 2006; Raup et al., 2007). The gridded temperature and precipitation are interpolated based on the station data by Inverse Distance Weighted (IDW) method, in which the vertical temperature lapse rate of $-6$ °C km$^{-1}$ is used to downscale the temperature station to the elevations of the

grid cells (Kattel et al., 2013). The daily gridded temperature and precipitation were input data for the GSM module, which calculated both snow accumulation and melt-water runoff."

L148: A spatial resolution of 10 x 10 km sounds very coarse to me considering the size of the basin. I don't think you can get a sufficient representation of the changes in meteorological variables, which vary strongly over short distances in the mountains. Besides, this resolution is probably problematic to do a proper routing.

Reply: Thanks for the professional comments. Our response was presented to the General Comment #3, above. We have now re-run all the simulations on 3*3 km resolution and updated table and figures in the revised manuscript.

L150: Simply interpolation air temperature horizontally will not be sufficient for terrain with strong vertical differences. I advise to use a vertical temperature lapse rate to downscale the temperature field to the elevations of your grid cells.

- Reply: Thanks for your comment. This is actually what we did in the IDW interpolation for gridded temperature, but we didn't explain it in detail and clear enough. So we have added the information in the methodology section of the revised manuscript: "The gridded temperature and precipitation are interpolated based on the station data by Inverse Distance Weighted (IDW) method, in which the vertical temperature lapse rate of −6 °C km−1 is used to downscale the temperature station to the elevations of the grid cells (Kattel et al., 2013)."

L164: Include reference to the paper that describes the glacier change parameterization (Lutz et al., 2013).

- Reply: We have added it: "The glacier changes are the result of a close interplay of projected changes in temperature and precipitation, which are calculated monthly in the parameterization approach (Lutz et al., 2013)."

L181: I do not understand the acronym MLR for linear regression. Where does the 'M' stand for?

- Reply: Thanks for the careful review. 'M' stands for 'multiple' and MLR means the multiple linear regression. We have corrected it in the revised manuscript.

L185: the variables 'u', 'w' and 'W' need to be described. Otherwise this part is completely unclear. A few lines of description in addition to the two equations would also be useful.

- Reply: Thank you for the comment. $\hat{u}_t^i$ is the $i$ th corresponding climatic predictor on the $t$ th day; $W_t^{sim}$ is the SDSM-simulated probability on the $t$ th day; $w_t^{sim}$ is the simulated precipitation state on the $t$ th day. Now we have updated the corresponding description in the 3.4 section.

L188: I would not state 'superior ability of simulation' for a method that does well in transforming changes in the mean, but not the standard deviation and extremes, as stated in L181-182.

- Reply: Yes, agree. We have corrected it and updated the corresponding description in the 3.4 section.

L195 'l' and 'i' need to be explained.

- Reply: Thanks. 'l' is the upper limits of the numbers of the sets{X,Y}.'i' is an ordinal number for the vector X and Y. We have corrected it in the revised manuscript.

L198: 'T' needs to be explained

- Reply: Thanks. 'T' means the transposition in the calculation. We have corrected it in the revised manuscript.

L199: What is meant by 'the parameters'. What kind of parameters?

- Reply: Thanks for the careful review. 'the parameters' means in this equation, both 'W' and 'b' can be adjusted to make the equation balanced: "where $W$ and $b$ are the parameters which determine the shape of hyperplanes $Y$ ". We have updated it in the revised manuscript.

L201: subscript 'j' needs to be explained

- Reply: Thanks. 'j' is also an ordinal number like, which makes $X_i$ and $X_j$ independent to each other. We have corrected it and made it clearer.
-

L213: Should 'station-scale hydrological data' be 'station-scale meteorological data'? I cannot imagine that the downscaling is done with hydrological data.

- Reply: Yes, thanks for the carefully review. We have corrected it. Please see section 3.4.2

L213-215: Which GCMs are used?

- Reply: We have added the information of GCMs in Data section. Besides, we also re-run all the simulations with eight GCMs including both Rcp4.5 and Rcp8.5. Please see them in table 2.

L221-222: Do I understand correctly that the observed glacier mass balance at one glacier was used for calibration? Can you elaborate on the assumption that this one glacier is representative for all glaciers in the catchment? You could also compare with remote sensing data (Brun et al., 2017; Gardelle et al., 2013) to see how large the spatial variation in glacier mass balance is in your catchment, to say something about the representativeness of Chhota Shigri.

- Reply: Thanks for the references and comment. In our study area, the glaciological mass balance series published in Spiti-Lahaul region (where Beas river basin locates) that is available for comparison are the Chhota Shigri glacier and Bara Shigri glacier (Berithier et al. 2007). In which, the only one is long enough to be comparable to our simulation period is the Chhota Shigri glacier (2002-2014), which has geodetic mass balance for validation (Azam et al. 2016). We have compared the mass balance data from previous studies for the Chhota Shigri glacier. In the study of Gardelle et al. (2013), a detailed map of elevation changes during 2000-

2011 in Spiti-Lahsul region based on SPOT5 DEM is given, which showed that the changes of the glaciers in the Beas river basin are quite similar as the change in Chhota Shigri glacier in general, although there is variability both in individual glacier and over the region. So we used the mass balance data of Chhota Shigri glcier for representing all the glaciers in our small basin.
We have added this explanation in the section of Model calibration: "There is an intra-regional variability of individual glacier mass balance in High Mountain Asia (HMA) and less negative mass balance than most other estimates according to the recent study of Brun et al. (2017). From the study, the annual glacier mass balance is -0.49+/-0.2 m w.e.yr-1 in Spiti-Lahaul region (where Chhota Shigri glacier locates) during 2000-2008 based on ASTER and 0.37+/-0.09 m w.e.yr-1 in Western Himalaya region from RGI Inventory during 2000-2016 based on ASTER. Besides, a detailed map of elevation changes during 2000-2011 in Spiti-Lahsul region based on SPOT5 DEM is given in the study of Gardelle et al. (2013), which showed that the changes of the glaciers in the Beas river basin are quite similar as the changes in Chhota Shigri glacier during 2000-2011 in general, although there is variability both in independent glacier and over the region.

L233: The biases are seem to be large for June-August because of the common problem of underestimated high-altitude precipitation in gauge-based data. If you did not use the improved precipitation fields based on WRF which you discuss in section 5.1, I believe you should include a correction for that in your model. It could be an additional parameter that you calibrate in advance, to make sure that the precipitation input is at least higher than the observed discharge. Have a look at for example (Dahri et al., 2016; Immerzeel et al., 2015). I am not saying you should use an approach as in the cited studies, but at least do a correction on the precipitation input to make it more realistic.

- Reply: Thank you very much for the suggestion. We have used the corrected precipitation based on WRF and gauge for historical baseline simulation. And the same correction has also been done for the precipitation in all the future scenarios. We have updated it in the revised manuscript of both methodology and results.

L236: Fig 2 and 3 are much repetition and can be combined in one figure. How do slow flow and fast flow relate to rain-runoff? I rain-runoff surface runoff and are slow flow and fast flow both groundwater flow and flow through the soil layer?

- Reply: Thanks for the comment. We have removed Fig.3 and added more information in the results part of the revised manuscript: "In Fig 2, the total discharge includes fast-flow and slow-flow from non-glacier area and discharge from glacier area, which includes rainfall discharge, snow-melt and ice-melt discharge. The fast-flow is generally considered to be surface runoff and the slow-flow refers to base flow. "

L241-243: I think it is a bit misleading to show one of the years where the model has best performance in figure 5 and then conclude that the model 'worked fine' in the study basin. You clearly indicated that there are quite large biases, especially during the high flow season, which is understandable when simulating high mountain hydrology. I would remove figure 5.

- Reply: Thanks for the suggestion. We have removed Fig. 5.

L243-246: Similar comment as for L221-222. How representative is the glacier mass balance at

Chhota Shigri for your entire catchment? Here you compare the simulated glacier mass balance for the entire catchment to the observed mass balance at one glacier.

- Reply: Thanks for the comment. Please see our reply above for L221-222.
  In our study area, the only glaciological mass balance series published are the Chhota Shigri glacier and Bara Shigri glacier (Berithier et al. 2007). In which, the only one is long enough to be comparable to our simulation period is the Chhota Shigri glacier, which has geodetic mass balance for validation (Azam et al. 2016). We have compared the mass balance data from previous studies for the Chhota Shigri glacier. In the study of Gardelle et al. (2013), a detailed map of elevation changes during 2000-2011 in Spiti-Lahsul region based on SPOT5 DEM is given, which showed that the changes of the glaciers in the Beas river basin are quite similar as the change in Chhota Shigri glacier in general, although there is variability both in independent glacier and over the region. So we used the mass balance data of Chhota Shigri glcier for representing all the glaciers in our small basin.
  In order to make it clearer in the manuscript. We have rewritten L236-237 and add the location of Chhota Shigri glacier in Fig. 1 in the revised manuscript: "In our study area, the glaciological mass balance series published in Spiti-Lahaul region (where Beas river basin locates) that is available for comparison are the Chhota Shigri glacier and Bara Shigri glacier (Berithier et al. 2007). In which, the only one is long enough to be comparable to our simulation is the Chhota Shigri glacier, which has geodetic mass balance for validation (Azam et al. 2016). The Chhota Shigri Glacier intersects with the northeast boundary of Beas river basin, which is close to Manali and Bhuter." We have updated the Fig. 1 in the revised manuscript.

Fig.6: Move the legend outside the plot or draw a clear boundary around it. Now it seems that de symbols in the legend are actual plotted values.

- Reply: Thanks. We have corrected the figure and have updated it in the revised manuscript.

Table 4: I do not understand the line below the headers (0, mean, mean, mean). I also do not understand why the column indicating the statistical downscaling method is headed 'RCM'. I also do not understand the meaning of the header 'Glacier – GCM'. I also do not understand what the line at the bottom of the table 'CMIP5: Bcc-csm' should indicate.

- Reply: Thanks for the comment. We have changed this table in order to make it clearer to read and understand. Please see Table 3 in the revised manuscript.

Table 4: You used different GCMs to generate future meteorological forcing for the hydrological model than were used for the future glacier projections. Ideally these should be the same since they are part of the same system. However, I understand that you took glacier projections from another study and included them in yours. I agree that it is better to use some glacier projection is stead of none, but you need to describe the disadvantages of the mismatch in future meteorological forcing used for the glacier evolution model and the hydrological model. This should be more elaborated than in line 300-305.

- Reply: Thank you for the comment. In order to keep consistence and have a more comprehensive future picture of water availability of Beas river basin, we have added the same two ensembles of four GCMs as the future projection of glacier in the study of Lutz et al (2016). We invited Arthur Lutz as a co-author for the revised manuscript. The same meteorological forcing was taken for re-running glacier evolution model by DR Lutz. So there is no longer mismatch in the meteorological forcing for hydrological model and for glacier evolution model in the revised manuscript.

Fig 7: In the caption you mention 'RCMs'. I do not think you can do that because you did not use

RCMs in your study, but statistically downscaled GCMs. I suggest to replace 'RCMs' by 'downscaled GCMs'.

- Reply: Thanks. We have replaced "RCM" by "Statistical Downscaling methods" in the whole manuscript.

L250-254, Fig 7: The two different downscaling methods lead to very different changes. This needs explanation of the underlying reasons. I wonder why the two methods were used. I suggest to validate both downscaling methods for the historical period, select the method that performs best, then use that method for the future projections.

Reply: Please see the reply to general comment #4. We have changed the statistical downscaling methods of SDSM and SVM to BDC and LOCI, and done more comprehensive comparison of the two new bias correction methods of the DBC and LOCI. In the revised manuscript, the uncertainty related to the choice of bias correction methods has been considered by using two bias correction methods (LOCI and DBC) with different levels of complexity. LOCI and DBC representing two typical bias correction categories are both commonly used in literatures. LOCI is a mean-based bias correction, which applies a mean monthly correction factor to GCM-projected simulations for each calendar month over the future period. DBC is a distribution-based method, which corrects the empirical distribution of GCM-projected simulations for each calendar month. Both methods correct the frequency of precipitation occurrence.
We have added section 4.3 Evaluation of DBC and LOCI in the revised manuscript.

Fig9: For precipitation, there is a large mismatch between the two downscaling methods, even at the start of the future simulation. For SVM they start around 1000 mm/yr whereas for SDSM they start at around 1500 mm/yr. This means that at least one of them shows a large sudden jump going from the historical period to the future period. If the downscaling was done properly, the transition from the end of the historical period to the start of the future period should be smooth.This needs to be addressed.

- Reply: Thank you very much for the careful review. We agree that the different SD methods may result in inconsistency in the simulations. After a careful consideration of the disadvantages of perfect prognosis (PP) methods and the advantages of bias correction methods, the downscaling methods of SDSM and SVM used in the original manuscript were replaced by two bias correction methods of DBC and LOCI in the revised manuscript.

In our revised manuscript, the bias correction methods involve estimating a statistical relationship between a climate model variable (e.g. precipitation) and the same variable of the observations to correct the climate model outputs. The use of bias correction methods is usually considered as reasonable way to achieve physically plausible results for impact studies. Compared to PP methods, bias correction methods are relatively simple to use and negate the prerequisite of a strong relationship between local-scale variables and large scale climate model variables.

We have now done a more comprehensive validation of those two statistical downscaling methods, and have added the results section 4.3. All the relevant parts including introduction, methodology and results in the revised manuscript have been updated.

The difference in temperature projections for the SDSM and SVM method are enormous towards the end of the century. Describe the reasons for this large difference in the manuscript. Since these methods where used to downscale 1 GCM, the quality of the downscaled forcing for at least one of the downscaling approaches is questionable to me.

- Reply: Please see the reply above. We have chosen two new bias correction methods, i.e. DBC and LOCI in the revised manuscript. We have done more comprehensive validation of those two methods and added section 4.3 Evaluation of DBC and LOCI in the revised manuscript.

Table 5: The change in discharge is very negative, although you have positive changes in precipitation. It seems that it can partly be explained by the increase in evapotranspiration and by losing the additional water from the negative glacier mass balance in the future. Nevertheless, it feels to me that the decrease in total runoff cannot be that large when precipitation amounts are increasing. Please provide a check of the simulated water balance components (precipitation, evapotranspiration, ice melt, snow melt, rainfall discharge, fast flow, slow flow, and changes in storages) of the catchment for your reference and future runs in the revised manuscript.

- Reply: Thank you very much for the carefully review. We have now chosen two new statistical downscaling methods, i.e. DBC and LOCI and added two ensembles of four GCMs of Rcp4.5 and Rcp8.5 in the revised manuscript.
There are no more issues of 'jump' between historical period and future period from the statistical downscaling. The discharge significantly decreases because of the glacier retreating. We have made the correction and have updated all the results.

Fig 12: The different lines of the ensemble member are indistinguishable. I suggest to show the ranges as a shaded area with a line for the mean. Since the figure shows all the members from Table 4, I do not understand why all precipitation projections here start around 1000 mm/yr, whereas in Figure 9 the precipitation projections show large difference for the two downscaling methods.

- Reply: Thank you very much for the suggestion and careful review. We have now chosen two new statistical downscaling methods, i.e. DBC and LOCI and added two ensembles of four GCMs of Rcp4.5 and Rcp8.5 in the revised manuscript. We have made the correction and have updated all the results. We also improved the Fig 12 according to the suggestion.

In the caption you mention that the plot shows glacier melt discharge for CanESM2. In Table 4 you indicate that 2 out of the 16 members use glacier projections for CanESM2. How come all the 16 members are shown in Figure 12? This is very unclear. It is also unclear how one could derive the in the caption mentioned tipping point years (2026 and 2052) from the plot.

- Reply: Thank you very much for the careful review. It was a typo. But we have now chosen two new statistical downscaling methods, i.e. DBC and LOCI and added two ensembles of four GCMs of Rcp4.5 and Rcp8.5 for a more comprehensive comparison and uncertainty analysis in the revised manuscript. We have updated all the results.

Glacier melt contributions around 500 mm/yr seem rather high compared to the plot you showed in Figure 2. There the annual sum of the 'Glacier ablation' seems to be much lower (around 250 mm/yr as far as I can estimate). This would also imply a sudden 'jump' going from the historical period to the future period, which is unnatural. This makes the whole story somewhat questionable. To gain confidence about the projections please provide a check of water balance components as indicated in the comment to Table 5.

- Reply: Thank you very much for the careful review. We have now chosen two new statistical downscaling methods, i.e. DBC and LOCI and added two ensembles of four GCMs of Rcp4.5 and Rcp8.5 in the revised manuscript.
  There are no more issues of 'jump' between historical period and future period from the statistical downscaling. We have updated all the results.

L306: See comment on the use of 'RCM' at Figure 7

- Reply: Yes. We have corrected it to be "statistical downscaling methods".

L308: I think you refer here to the 'jump' I point out in my comment about about the glacier melt in Figure 12. I think this is something that needs to be addressed before the projections have sufficient reliability to be published in HESS.

- Reply: Thank you for the comment. Yes, We have now chosen two new statistical downscaling methods, i.e. DBC and LOCI and added two ensembles of four GCMs of Rcp4.5 and Rcp8.5 for a more comprehensive comparison and uncertainty investigation for the future water cycle and availability in this Himalaya headwater Beas river basin.

L313-337: This part comes out of the blue. It is unclear if you used the combined precipitation and WRF forcing in this study. If you did not, I suggest you redo the study with this precipitation dataset if it has a better representation of precipitation. If you did, integrate this part then in the manuscript (i.e. the methodology to the Methods section, and the results to the Results and Discussion section).

- Reply: As we mentioned in the earlier reply, in the revised manuscript, we have redone the modeling using the corrected precipitation for both the historical period as baseline and future scenarios.

**Technical comments**

L11: Remove 'the' at the end of the sentence

- Done.

L13: remove 'the'. 'Climate' shoud be with lower case 'c'

- Done.

L18: remove 'impact'

- Done.

L21: Better to reword to: 'This will result in a general decrease in river runoff for all the scenarios.'

- Done.

L23: I guess you mean 'WRF precipitation projection'

- Yes, clarified.

L28: Remove the first 'The'. From here I will stop correcting the redundant use or absence of 'the'. Please have the manuscript checked by a native English speaker. It is advisable to do this before submission for any future manuscripts.

- Reply: thank you very much. We have carefully done proofreading.

L29: Reword to: 'Hydrological models have been developed and used as a main assessment tool in the Himalayan region to estimate the impacts of climate change for future water resources.'

- Done.

L35: 'the' should be 'an'. 'GCM' should be 'GCMs'.

- Done.

L38: Change 'More' to 'An increase in'

- Done.

L39: Change 'by' to 'according to'

- Done.

L54: Introduce Regional Climate Models before using the acronym RCM

- Done.

L83: 'simulations' should be 'simulation'

- Done.

L95: Reword to 'The main questions we try to answer in this study are:'

- Corrected.

L110: add m asl (metres above sea level)

- Done.

L115: correct 'meteorological'

- Done.

L137: No parentheses needed here

- Done.

L141: No parentheses needed here

- Done.

L167: 'totally' should be 'in total'

- Done.

L178: Remove 'was' and 'which'

- Done.

L243: Included 'simulated' between 'The' and 'annual'

- Done.

General: There are many textual errors. Please have the whole text reviewed by a native English speaker before submitting the revised manuscript. Please do this for future submissions before the initial submission of the manuscript.

- Reply: Thank you so much for the careful review and correction! Besides, we have asked help from our native English speaker colleagues to correct the further textual errors in the whole manuscript.

- Reply: Thanks a lot for the recommended citations. We have added most of them in the revised manuscript.
The manuscript by Li et al. investigates the impact of climate change on glacier melt contribution to discharge in a medium-sized catchment in the Indus basin. To this end, a calibrated glacio-hydrological model was driven by statistically downscaled climate projections from one GCM under two GHG concentration scenarios. The simulations build on ensemble projections of glacier extent derived from a previous study by Lutz et al. (2016) who have already provided a more comprehensive assessment for the entire Indus basin. The manuscript mainly reports on model application in a particular basin and generally lacks novelty.

- Reply: Thanks for your comments. We have stated better the novelty of the study. Although there have been studies (i.e. Lutz et al. 2016; Li et al., 2016), looked at the hydrological projections in Indus river basin under climate change, but there is no common conclusion from the studies about the water future of western Himalayan region. Both studies suggest that more studies need to be done in the region. Furthermore, in Lutz et al. (2016), they used a corrected precipitation dataset (which is based on 0.25 degree gridded dataset APHRODITE) for historical period, because it was found that there is underestimation of precipitation at high-altitude area in Himalayan region (Immerzeel et al., 2015). In our study, we used a high-resolution (which is 3 km) dynamical regional climate model (WRF) precipitation data for correcting the underestimated precipitation (Li et al., 2017). Moreover, we have added two more ensembles of four GCMs according to the study of Lutz et al. (2016) for a more comprehensive comparison and uncertainty investigation for the future water cycle and availability in this Himalaya headwater Beas river basin. In the results, the uncertainty partition of hydrological projection from GCMs and statistical downscaling methods has been analyzed.

  We agree that the structure of our earlier version of the manuscript is not clear enough and made some confusion. We have sorted out the whole structure in the revised manuscript. The corrected precipitation based on Gauge and WRF has been moved to the first part of the method and results. We have also re-calibrated the model based on the new corrected precipitation. Furthermore, we have done the same correction for future precipitation. We have updated all the results based on the new simulations.

The glacio-hydrological modeling capitalizes on projections of future glacier extent from Lutz et al. (2016). Data derived from the Lutz et al. study should be moved to Materials and Methods and should be separated more clearly from the GSM-WASMOD modeling results obtained in the current study. This concerns section 4.3 including figures 10 and 11.

- Reply: Thanks for your comments. In our revised manuscript, we invited Arthur Lutz as co-author for re-running glacier evolution model by the same meteorological forcing from two new statistical downscaling methods under future scenarios of two ensembles of four GCMs (Rcp4.5 and Rcp8.5). All results were updated by the output from new simulations. We have also re-structured the manuscript including description of Data, methods and Results accordingly.

The manuscript has a poor structure and is more often than not hard to follow. For example, modeling results are presented and superficially discussed in "Results and discussion" which is however followed by a "Discussions" section that in fact introduces a completely new modeling experiment including data, methods, results and discussion. The additional material addresses the issue of uncertainty in precipitation data in high altitudes. This topic is without question relevant for hydrological modeling in the study region, however falls largely out of the scope of manuscript. In the remainder part (section 5.2) this topic is further discussed while a critical discussion

of the main results presented in sections 4.2 - 4.4 is largely missing.

It is only mentioned by the end of the results section that only one GCM was down- scaled to drive the glacio-hydrological simulations while all previous sections give the impression that a GCM ensemble was used. A plethora of previous studies has shown that GCMs contribute a large share to total uncertainty in simulated hydrological impact and it is consequently common practice to drive (an ensemble of) impact models with a GCM ensemble. In this regard, the study clearly falls behind the state of the art and the material does not support significant conclusions.

The manuscript contains a large amount of figures and tables, 21 in total, of which some seem redundant and the authors should make an effort to streamline the mate- rial. For example, Table 4 listing all possible combinations of GCM, RCP and method of bias correction is largely identical in content to Table 2.

- Reply: Thanks for the comments. We agree and accept all of them. We feel sorry that we failed to present our work well in the original version. As we mentioned above, we agree that the structure of the previous manuscript was not clear enough and caused some confusion. We have now re-structured the whole manuscript. Furthermore, we have added two ensembles of four GCMs similar as in glacier projections (Lutz et al. 2016) for a more comprehensive comparison and uncertainty investigation for the future water cycle and availability in this Himalaya headwater Beas river basin.
  The rewritten manuscript follows those three main questions: "(1) How much uncertainty is in the precipitation over the ungauged high-altitude in Beas river basin? (2) How will the future water availability change due to higher glacier melt under warmer future in Beas river basin over the Himalaya region? (3) What are the uncertainties of the future water from GCMs or Statistical downscaling methods?  To answer these questions, precipitations from a high-resolution WRF simulation and Gauge are investigated and a corrected precipitation is used for the hydrological simulation as the historical baseline.
  In the study, we use a glacio-hydrological model together with eight GCMs under two generation of scenarios, i.e. RCP 4.5 and RCP 8.5 and two statistical downscaling (SD) methods. We firstly focus on the simulation of the present day water cycle and validation of the simulated discharge by using the observed discharge. The uncertainties of the precipitation over high-altitude area and hydrological simulation are further discussed. Besides, the future climate change and glacier extent change and hydrological changes are investigated. At last, the uncertainty from GCMs and statistical downscaling methods is analyzed and discussed before presenting the main conclusions."
  In this revised manuscript, we have removed Fig 3, Fig.5 and Table 2. We also split section 5.1 and fill into three parts: 1) section 2.2 Data, 2) section 3.1 of precipitation correction and 3) section 4.1 corrected precipitation and section 4.2 GSM-WASMOD model calibration and validation.

The standard of English needs to be improved throughout the manuscript. While the meaning is usually (but not always) clear, there are a lot of grammatical errors (far too

many to list) and diction is often poor.

- Reply: Thanks for your comments. We have carefully checked the typos and grammatical errors through the whole revised manuscript and a native English speaker colleague has also corrected it.

Specific comments

L. 11: Why would the glacier melt lead to extreme rainfall?

- Reply: Thanks for the comment. That is obviously a mistake. We have corrected it: "The changes in glacier melt may lead to droughts as well as extreme floods in the Himalaya basins, which are vulnerable to the hydrological impacts."

L. 13: I strongly disagree with the use of the term RCM when referring to the two methods of GCM bias correction/downscaling applied in this study. The term RCM describes numerical prediction models.

- Reply: Thanks for the comment. We changed them all in the revised manuscript to be "statistical downscaling methods".

L. 30-32: Colloquial, please rephrase.

- Done.

L. 36: Please correct to "CMIP5"

- Corrected.

L. 67: Correct to "Mishra 2015"

- Corrected.

L. 88: Unclear, please rephrase.

- Done.

L. 115-117 : This section describes the study basin/region; information on the model and data used should be moved to the corresponding sections.

- Done.

L. 115: Please correct to "meteorological"

- Corrected.

L. 130: Was the GSM module developed in the scope of this study? If not, please add the reference to the original publication.

- Reply: Thanks for the comment. The GSM module is not original developed in this study. We have already added the original references in the revised manuscript.

L. 148: What was the reason for choosing a modeling resolution of 10 km? Most of the input data sets do seem to support a higher modeling resolution; please clarify.

- Reply: Thanks for the comments. We have now re-run all the simulations at 3*3 km resolution and found that the results of calibration and validation were not improved comparing with the results from 10*10 km resolution simulations. It was not a surprising because of limited gauge data that we have in the study area. According to the previous studies and analysis of the influence of interpolation and station density on gridded daily data  (i.e. Dirksa et al. 1998; Hofstra et al., 2010; Xu et al., 2013), the results showed that the network density could introduce biases in the mean and variance of the grid values (i.e. precipitation and temperature) compared to those expected for the true area-averages.
  However, concerning the precision of routing, glacier revolution and smooth of discharge graph and 'step change' because of the coarse resolution, we finally decided to use the 3km simulation in the revised manuscript. All the Tables and figures are updated by new simulation results.

- Dirks, K.N., Hay, J.E., Stow, C.D. and Harris, D., 1998. High-resolution studies of rainfall on Norfolk

Island: Part II: Interpolation of rainfall data. *Journal of Hydrology*, *208*(3-4), pp.187-193.
- Hofstra, N., New, M. & McSweeney, C. Clim Dyn (2010) 35: 841. https://doi.org/10.1007/s00382-009-0698-1
- Xu, H., Xu, C.Y., Chen, H., Zhang, Z. and Li, L., 2013. Assessing the influence of rain gauge density and distribution on hydrological model performance in a humid region of China. *Journal of Hydrology*, *505*, pp.1-12.

L. 149: It was mentioned earlier that potential evaporation was only available from one station. Were these station values used for the entire basin? Please clarify.

- Reply: There is only one pan-evaporation station during 1996-2011 in the study region. It didn't show improvement in the simulation with observed Epan and it results in inconsistency to combine it. After a few testing run, we decided not to use it in the simulation. We have updated this in the Data section of the revised manuscript.

L. 155-156: Unclear, please rephrase.

- We have added more explanation here in the revised manuscript: "The routing method in GSM-WASMOD is called NFR routing algorithm (Gong et al. 2009, 2010), which was developed to adapt to the coarse resolution hydrological modeling. This is a scale-independent routing method for network-response function using high-resolution aggregated hydrographgy HYDRO1k. The algorithm preserves the spatially distributed time-delay information in the form of simple network-response functions for any low-resolution grid cell in a large-scale hydrological model."

Section 3.4: 1) The authors miss to describe and reference the 21st century GCM ensemble data used in the study. Please add a section or paragraph.

- Thanks for the comment. We have added two ensembles of four GCMs (Lutz et al. 2016) for comparison in the revised manuscript, and more clarification has been added in the Data section and in Table 2.

2) Lutz et al. (2016) applied the same GCM ensemble but a different downscaling approach to simulate the future glacier extent used in this study. Why did the authors choose a different downscaling technique? Given that the downscaling technique is found to have a profound effect on projected precipitation and temperature (which drive both the simulated glacier extent and melt), how does this inconsistency affect the results for the Beas river basin and the conclusions drawn?

- Reply: Thanks for the comment. We agree that the different SD methods may result in inconsistency in the simulations. After a careful consideration of the disadvantages of perfect prognosis (PP) methods and the advantages of bias correction methods, the downscaling methods of SDSM and SVM used in the original manuscript were replaced by two bias correction methods of DBC and LOCI in the revised manuscript.

  Both SDSM and SVM are regression-based downscaling methods, which involve estimating the statistical relationship (e.g. linear relationship for SDSM and nonlinear relationship for SVM) between large scale predictors (e.g. vorticity, mean sea-level pressure, geopotential height and relative humidity) and local or site-specific predictands (e.g. precipitation and temperature) using observed climate data. The reliability of a regression-based method relies on relationships between observed daily climate predictors and predictands. However, these relationships are usually weak, especially for daily precipitation. In addition, the regression-based method is usually incapable of downscaling precipitation

occurrence and generating proper temporal structure of daily precipitation, which is critical for hydrological simulations. Moreover, the PP downscaling method establishes relationship between predictors and predictands for the historical period and then applies it to future periods. However, this relationship may not hold for the future in a changed climate. This may partly explain why there was a jump between downscaled historical and future precipitation and temperature simulation in our previous manuscript. In particular, the relationship between predictors and predictands established using reanalysis predictors are applied to GCM predictors based on an assumption that reanalysis predictors and GCM predictors are both "perfectly" simulated at the grid scale (Wilby et al., 2002; Dibike and Coulibaly, 2005; Chen et al., 2011a). While reanalysis and GCM data do share some similarities, they are completely independent. Reanalysis data aim at representing the real world, whereas GCMs operate in their own virtual world. This may further result in the jump of precipitation and temperature between historical and future period.

In our revised manuscript, the bias correction methods involve estimating a statistical relationship between a climate model variable (e.g. precipitation) and the same variable of the observations to correct the climate model outputs. The use of bias correction methods is usually considered as reasonable way to achieve physically plausible results for impact studies. Compared to PP methods, bias correction methods are relatively simple to use and negate the prerequisite of a strong relationship between local-scale variables and large scale climate model variables. Previous work indicates that statistical downscaling using GCM precipitation or temperature directly as a predictor performed much better than using other predictors.

We are now using two new bias correction methods (DBC and LOCI) and have added more comprehensive validation in the results section 4.3. All the relevant parts including introduction, methodology and results in the revised manuscript have been updated.

4) Sections 3.4.1 and 3.4.2 need to be rewritten to enhance comprehensibility. In the current version, it is impossible to understand how both downscaling approaches work.

- Thanks for the comment. We have rewritten this part and made it more understandable in the revised manuscript.

L. 209-215: "SSVM is directly used to construct the relationship between hydrological data and atmospheric variables" and "The calibration of downscaling models used the station-scale hydrological data and GCM historical atmospheric variables to construct the relationship": I understood from the earlier text that both techniques were used the downscale GCM simulated atmospheric variables to station-scale meteorological data which subsequently were used to drive the glacio-hydrological simulation. Did the authors establish a direct statistical relationship between atmospheric variables and hydrological fluxes? Please clarify.

- Reply: We have re-run all the simulations based on two new bias correction methods. The methodology and results have been updated in the revised manuscript accordingly. Please see the reply on the early comment of section 3.4, 2).

Section 3.5: 1) In L. 220, Li et al. 2013a or Li et al. 2013b?

- We have corrected the reference to be Li et al. 2013a.

2) Glacier mass balance data were apparently used for calibration, but this data-set has not been described or mentioned yet. Please add a description to the data section.

- We have added more information about the glacier mass balance data that we used in the section 2.2 of Data: "The annual glacier mass balance data of Chhota Shigri Glacier used in the model calibration are taken from the previous studies of Berthier et al. (2007), Vincent et al. (2013), Azam et al. (2016)." Besides, we have also added more explanation in the section 3.6: "There is an intra-regional variability of individual glacier mass balance in High Mountain Asia (HMA) and less negative mass balance than most other estimates according to the recent study of Brun et al. (2017). From the study, the annual glacier mass balance is -0.49+/-0.2 m w.e.yr-1 in Spiti-Lahaul region (where Chhota Shigri glacier locates) during 2000-2008 based on ASTER and 0.37+/-0.09 m w.e.yr-1 in Western Himalaya region from RGI Inventory during 2000-2016 based on ASTER. Besides, a detailed map of elevation changes during 2000-2011 in Spiti-Lahsul region based on SPOT5 DEM is given in the study of Gardelle et al. (2013), which showed that the changes of the glaciers in the Beas river basin is quite similar as the changes in Chhota Shigri glacier during 2000-2011 in general, although there is variability both in independent glacier and over the region. Furthermore, in our study basin, the glaciological mass balance series published in Spiti-Lahaul region (of HMA) that is available for comparison are the Chhota Shigri glacier and Bara Shigri glacier (Berithier et al. 2007). In which, the only one is long enough to be comparable to our simulation period is the Chhota Shigri glacier (2002-2014), which has also geodetic mass balance for validation (Azam et al. 2016). So we used the mass balance data of Chhota Shigri glacier as a representation for the glaciers in our small basin."

3) The efficiency criteria listed seem to refer to simulated discharge only. How was model efficiency evaluated with respect to glacier mass balance?

- We manually adjusted the parameters of glacier module according to the annual glacier mass balance data from previous studies. The bias is used for evaluation with respect to glacier mass balance. We have added the explanation in the revised manuscript. Please see section 3.6.

4) Were discharge and glacier mass balance calibrated simultaneously?

- We firstly 'pre-calibrate' all parameters according to the total discharge. Then we manually adjusted the parameters of glacier module according to the glacier mass balance. At last, we set the glacier module parameters and re-calibrate the other parameters according to discharge data one more time. We have added more clearly explanation in the revised manuscript.

L. 242 "worked fine": Colloquial, please rephrase. Further, I cannot see how Fig. 5 adds important new information. If its only purpose was to show that the model "worked fine", the figure can be removed.

- Thanks for the comment. We have removed Figure5.

L. 245: It was mentioned earlier that glacier mass balance data were used to calibrate GSM-WASMOD; are those the same data as used here for validation?

- Thanks for the comment. We used the mass balance data for calibration and also compared the glacier mass balance for validation in the section of 4.2 in the revised manuscript.

L. 250: Table 4 formally belongs to the methods section and should be referenced

there.

- Thanks for the comment. We have changed it.

L. 255-265: The two downscaling methods seem to introduce a large uncertainty with respect to future climate in the region. How does this uncertainty compare to the spread between the different GCMs?

- We have added two ensembles of four GCMs (Lutz et al., 2016) in the revised manuscript and compared the uncertainty from statistical downscaling methods and from GCMs in the revised manuscript. Please see section 5.2.

L. 294 "It shows that the summer peak of runoff sifts to the other seasons in Beas river basin": Cannot be inferred from the figure.

- Thanks for the careful comment. We have modified it.

L. 300 and following: It is mentioned here for the first time that only GCM was down- scaled to drive the glacio-hydrological model. This should have been made clear in the methods section.

- Thanks for the comment. We have corrected it and updated the methodology and results in the revised manuscript.

Tab. 2: Please rephrase the caption and correct to "glacier evolution"; "Selected model" in the table heading is rather ambiguous and could be replaced by "GCMs"

- Done.

Table 3: Please correct to "validation", "Nash-Sutcliffe coefficient" and "NS_d" (row 6); typographical error in the last row; missing space before table number.

- Done.

Table 5: Please provide a more informative caption. I assume ensemble median and range are show. "Change" should be spelled lower case. Does the table show changes over the glacierized area or for the entire river basin?

- We have added a new informative caption for Table 5. Yes, the values are mean with range of minimum and maximum values. This is the result from the whole river basin. We have clarified it in the revised manuscript.

Fig. 2: In the legend, please correct to "Simulated dis"

- Done.

Fig. 3: Please add the observed discharge for reference

- Done.

Fig. 4: Please correct to "Monthly hydrographs". The quality of the Figure should be improved.

- Done.

Fig. 6: The observed data shown seem to be mean values over certain time periods rather than estimates for a single year (e.g. 1999–2004 in Vincent et al. 2013), but are depicted as points in the figure which is misleading. Please correct. Further, please add a table listing all external glacier MB data including reference period and estimation method.

- Thank you for the comment. We have corrected the data and its reference in the Fig. 6. We have also added Table 4 for listing all of the observed annual mass balance data from previous studies, which was used in the glacier module calibration and validation in the revised manuscript.

Fig. 7+8+9: I strongly disagree with the use of the term "RCM" when the authors actually refer to bias correction methods, please correct. Please revise the captions. Do the figures show the ensemble mean? If yes, please add the ensemble range.

- We have corrected the use of term "RCM" in the whole revised manuscript. Besides, we have added the ensemble mean and range in Fig 7, Fig 8, Fig 15 and Fig 16.

Fig. 10: Y-axis label should read "Glacier"

- Corrected.

Fig. 11: Is this the ensemble mean?

- Yes, it is the ensemble mean. We have removed this figure in the revised manuscript regarding in order to reduce the number of figures that we have in the paper.

Fig. 12: The figure needs profound revision. 1) I can only guess that the numbers in the legend refer to the index given in Table 4. Listing all ensemble members in the legend is somewhat obsolete since they are not distinguishable in the plot. 2) The caption claims that results for only one GCM are shown (CANESM2) while the figure apparently shows the whole ensemble. 3) Are both RCPs shown? If yes, please color- code accordingly. 4) In all simulations glacier melt discharge approaches 0 by the end of the century while according to Table 5 glacier cover remains larger than 0. Please explain. 4) Why is glacier-melt discharge given in negative numbers?

- Thanks for the comment. We have re-plot the figure into two new figures of Fig. 13 and Fig.14 for a clearer clarification in the revised manuscript. In the new figures, the different bias correction methods are represented by different colors (blue for LOCI and red for DBC), while the different GCMs are shown in different line styles with (for RCP45) or without (for RCP85) marker.
  We have corrected the glacier-melt discharge to be positive values. And the glacier discharge in Fig 14 is the total discharge including rainfall discharge, snowmelt discharge and ice-melt discharge from glacier-covered grids. We have clarified clearer in the revised manuscript. The result in Fig 14 is now consistent with the results in Fig. 12 of glacier extent evolution in Beas river basin.

Fig. 14: The two subfigures seem to show exactly the same data with respect to the single ensemble members. Please double-check.

- Thank you so much for the careful review and corrections! We have corrected them according to the above comments about the Tables and Figures in the revised manuscript. Besides, all the figures and tables have been updated by the new results in the revised manuscript with respect to the added new GCMs and statistical downscaling methods.
This is a very useful study that has been conducted for the data-scarce Himalayan Basin. I have gone meticulously through the paper and I have the following queries:

1) Line 24. The study helps to understand the hydrological impacts of climate change in North India and make a contribution to stakeholders and policymakers with re- spect to the future of water resources in North India. - However, since only one GCM (BCC_CSM 1.1) is used for the study, how accurate would be the predictions to be able to be referred by the policymakers? -How is the use of this particular GCM, 'Bei- jing Climate Center Climate System Model' (BCC_CSM 1.1), justified for use over the Himalayan basin? Please elaborate on this issue.

- Thanks for your positive evaluation in general and for your professional comment. We agree with it and we have added two ensembles of four GCMs (lutz et al. 2016) and invited Arthur Lutz as co-author for re-run glacier extent projections by the same meterological forcing as hydrological model in the revised manuscript for a more comprehensive comparison and uncertainty investigation for the future water cycle and availability in this Himalaya headwater Beas river basin.

2) Line 237. Authors should present a figure showing the location of Chhota Shigri glacier in the Beas Basin. Because according to SERB report (Ramanathan, 2011), Chhota Shigri glacier is a part of the Chandra Basin. Chandra basin is a sub-basin of the Chenab river basin according to IndiaWRIS basin maps and the SERB report by Ramanathan (2011).

- Thanks for the comment and reference. We have corrected it in the manuscript: "The Beas river basin is located in Spiti-Lahaul region, where the available glaciological mass balance series published for comparison are the Chhota Shigri glacier and Bara Shigri glacier (Berithier et al. 2007). The Chhota Shigri glacier is the only one which has been well studied and has detailed and longer period of glacier mass balance data, which also has geodetic mass balance data for validation (Azam et al. 2016). The Chhota Shigri Glacier is a part of the Chandra Basin, which is a sub-basin of the Chenab river basin (Ramanathan, 2011), but it is attached to northeast boundary of Beas river basin, which is close to Manali and Bhunter stations (Fig 1.). In this case, it is used for glacier module calibration in the study, which is to be comparable to the simulation."

  We have also added the location of Chhota Shigri glacier in the Fig. 1 of the revised manuscript. Please see the Fig 1 in the "reply to the RC1".

3) Line 150. Chhota Shigri glacier Area is about 16 Km2 (Ramanathan, 2011), the resolution of the hydrological model GSM-WASMOD is 10*10 Km2.The limitation mea- sured on line 306 also mentions the same thing. However, I feel that the model in the study is too coarse to be able to accurately represent the outflow from the glacier melt. How is such a coarse model justified to be used for representing glacier melt from such small area glaciers and the glacier evolution?

- Reply: Thanks for the comment. We understand the concern from you. We used mass balance data of Chhota Shigri glacier for comparison with the simulation of the study, because it is the well monitored

and studied glacier whose data are available for using. From the revised new Figure 1, we can see that the Chhota Shigri glacier is very small glacier compared with the whole glacier cover in the Beas river basin. In the study, we are looking at the whole glacier extent of Beas river basin and its impact to the total basin runoff, instead of a single Chhota Shigri glacier, which has been done by several previous papers (i.e. Berithier et al. 2007, Azam et al.2016).

Furthermore, we have now re-run the model on 3*3 km resolution and updated all the results based on the new simulations.

4) Line115. Since the outlet station is Thalout station used for calibration of discharge, I would like to know what is the area of the Beas basin upto Thalout?

- Thanks for the comment and reference. We added the mark of watershed area up to Thalout in the new Fig 1. (please see Fig. 1 in the "reply to the RC1").

[revised manuscript text omitted]

---

## Referee Report (RR1)

[revised manuscript text omitted]

LOCI RCP45 IN4
LOCI RCP45 IPR
LOCI RCP45 MR3
LOCI RCP45 CA2
LOCI RCP85 CS0
LOCI RCP85 IPR
LOCI RCP85 MI5
LOCI RCP85 MR1
DBC RCP45 IN4
DBC RCP45 IPR
DBC RCP45 MR3
DBC RCP45 CA2
DBC RCP85 CS0
DBC RCP85 IPR
DBC RCP85 MI5
DBC RCP85 MR1
BASELINE

860     **Fig. 8 Average monthly mean observed (Baseline of 1986-2005) and simulated precipitation based on two bias correction methods under climate change scenarios from two ensembles of four GCMs over Beas river basin during (a) 2046-2065, (b) 2080-2099.**

[Figure]

**Fig. 9** Average monthly mean observed (Baseline of 1986-2005) and simulated temperature based on two bias correction methods under climate change scenarios from two ensembles of four GCMs over Beas river basin during (a) 2046-2065, (b) 2080-2099.

[Figure]

**Fig. 10 The average annual temperature and precipitation based on two bias correction methods under climate change scenarios from two ensembles of four GCMs, including RCP45 and RCP84, over the Beas river basin during 1986-2005, 2046-2065 and 2080-2099.**

[Figure]

**Fig. 11 Projected changes of glacier extent for Beas river basin during 21st century.**

[Figure]

**Fig. 12 Glacier discharge from LOCI and DBC under scenarios from two ensembles of four GCMs including RCP4.5 and RCP8.5 over Beas river basin during (a) 1986-2005, (b) 2046-2065 and (c) 2080-2099. The back line in sub-figure (a) represents the baseline glacier discharge in historical period with the corrected precipitation (please note the scales change of Y-axis in three sub-figures).**

[Figure]

**Fig. 13 Total discharge from LOCI and DBC under scenarios from two ensembles of four GCMs including RCP4.5 and RCP8.5 over Beas river basin during (a) 1986-2005, (b) 2046-2065 and (c) 2080-2099. The back line in sub-figure (a) represents the baseline discharge in historical period with the corrected precipitation.**

[Figure]

**Fig. 14 Changes of mean monthly evaporation (upper panel) and discharge (down panel) over Beas river basin for the middle of the century (2045-2055) ( left panel) and the end of the century (2080-2099) (right panel) comparing with the baseline period (1986-2005). The envelope represents the results of multiple model ensemble.**

895

900

[Figure]

**Fig. 15 The mean monthly evaporation (upper panel) and discharge (down panel) over Beas river basin for the middle of the century (2045-2055) (left panel) and the end of the century (2080-2099) (right panel) comparing with the baseline (1986-2005). The envelope represents the results of multiple model ensemble.**

905

---

## Author Response (AR2)

**Cover Letter for hess-2017-525**

Dear Prof Pechlivanidis,

I, on behalf of my co-authors, would like to thank you and the reviewers for the efforts on the improvement of our manuscript entitled "Twenty-first century glacio-hydrological changes in the Himalayan headwater Beas river basin" (hess-2017-525).

We have carefully followed all suggestions in making our revisions. Changed parts are marked in blue text in the paper. The point-by-point response to the comments is presented separately.

Again, we would like to thank the Editor's encouragement, support and patience to our work. We hope the revised manuscript is to your satisfaction, and of course we are happy to improve it according to further comments if needed.

Looking forward to hearing from you.

With all best wishes

Yours

Lu Li

On behalf of my co-authors

**View Letter**

Dear Editor and reviewers:

Many thanks for the review comments that we received with respect to our paper. We have carefully addressed both the editor and reviewer comments and suggestions. In the revised version, blue colored text represents text that has been revised or relocated. The whole manuscript is reviewed and corrected by a native English speaker. Below are our point-by-point responses to each of the reviewer's comments in blue.

**Editor Decision: Publish subject to minor revisions (review by editor)** (21 Dec 2018) by Ilias Pechlivanidis
Comments to the Author:
This article was reviewed by two new referees. This was required, discussed and agreed with the authors, given the major work that was conducted and the changes in the list of authors. However, I am pleased to read by the referees that the authors managed successfully to address major comments. The referees and I would like to highlight the significant effort to improve the analyses resulting to a much improved manuscript.

- Reply: We thank the editor for your positive evaluation on our work in general.

However, I would like to stress three points, which should be fully addressed together with the minor comments from the referees, in order to consider the manuscript for publication:

1) The manuscript is very long, and could be shortened. Particularly the Introduction is very extended (3 pages) which is apparently not necessary, since the reader is interested on key messages from existing literature. Although the Discussion is adequate, I still believe that there are parts which are repetitive. Please go through the manuscript and focus on conveying simple but key and concrete messages.

- Reply: Thank you for the suggestion. We have cut the introduction and have gone through the whole manuscript to make the messages clearer and simpler.

  2) The results and text is much focused on the climatological variables. This is fine since climatological variables drive the hydrological response, however the title directs towards a more hydrological investigation. I would therefore suggest reducing some repetitive text from the climatological analysis.
- Reply: Thanks for the advice. We agree and therefore we have removed Fig .10, because we feel the information that Fig 10 shows is repetitive with Fig. 8 and Fig. 9. Besides, we also added more discussion on the evaporation and streamflow change regarding different scenarios in section 4.6 Future Hydrological changes.

  3) It is crucial that a native English speaker revise the manuscript prior to resubmission. This is highlighted by the referees also in order to improve readability.
- Reply: Sorry that we fail to fulfil the improvement on language. A native English speaker has done a proofreading of the revised manuscript.

  I would like once again to thank the authors for this effort.

Thank you very much!

Referee 1:

**Suggestions for revision or reasons for rejection (will be published if the paper is accepted for final publication)**

1. Add more discussions on river flow response to the various scenarios.
- Reply: Thanks for the comment, following which, we have added more detailed discussion on the streamflow change regarding to various scenarios in the revised manuscript.

2. Some of the figures are not easy to visualise and need to be re-drawn.

- Reply: We have gone through all the figures and have re-drawn Fig. 1, Fig. 7, Fig. 8, Fig. 9, Fig 11 and Fig. 12 in order to make them clearer. These are included in the revised manuscript.

3. Pdf copy with more comments is attached

- Reply: Thanks for the detailed comments. The point-by-point responses to the reviewer's comments including those in the attached pdf.

- Thanks for the suggestion. We want to keep the title as it is although the uncertainty assessment of glacio-hydrological change is one of the main objectives. Because we didn't include a sophisticated method to estimate the uncertainty, although a range of models from both GCMs and bias corrections are used for comparison and quantifying uncertainty somehow.
- We have cut the abstract and made it more condense.
- We have added those two bias correction methods here: "… This study used an integrated glacio-hydrological model: Glacier and Snow Melt - WASMOD model (GSM-WASMOD) for hydrological projections under 21st century climate change by eight Global Climate Models (GCMs), two bias correction methods (i.e., the daily bias correction (DBC) and the local intensity scaling (LOCI)) under two Representative Concentration Pathways (RCP4.5 and RCP 8.5) in order to assess the future hydrological change at the Himalayan Beas basin up to Pandoh dam (upper Beas basin)."
- We have removed those two bias correction methods from here and move it in the earlier place (please see reply above).
- We have capitalized them and corrected them to be "Support Vector Machine (SVM)".
- Thanks for the comment. We have re-framed the aims of our study and make it more clearly: "The following research questions are examined in this paper: (1) How will the river streamflow change due to higher glacier melt under a warmer future in the upper Beas basin? (2) How large will the variability be in future key hydrological terms regarding different climate scenarios (i.e., RCPs, GCMs and statistical downscaling methods) in the upper Beas river basin?"
- We have rewritten this section for a clearer interpretation: "To answer the questions, we used a glacio-hydrological model to assess future glacio-hydrological changes in the Himalayan Beas river basin forced with two ensembles of four GCMs under two scenarios (RCP 4.5 and RCP 8.5), and two bias correction methods. Our paper is structured as follows: after the introduction, a description of the study area and data is presented, followed by the methods utilized, including the GSM-WASMOD model, glacier evolution parameterization, bias correction methods, precipitation correction and model calibration. Next, we focus on the simulation of the present-day water cycle, and calibration and validation of the model to observed data. Then, the results of future climate change and its impact on glacier extent and hydrological projections are presented. Finally, a more detailed discussion on uncertainties of precipitation over high-altitude region and future hydrological projections in the upper Beas basin are addressed before presenting the main conclusion."
- Thanks for the comment. The glacier extent has certainly been changing in the past and a conceptual model does not mean the glacier extent is not changing. The assumption here in the study is that the number of glacier-covered grid cells is consistent in the historical simulation of 1986-2005. We have rewritten this part to be: "GSM-WASMOD is a conceptual glacio-hydrological model and we assume that the number of glacier-covered grid cells does not change in the historical simulation. For the future simulations, we used a basin-scale regionalized glacier mass balance model with parameterization of

glacier area changes and subsequent aggregation of regional glacier characteristics (Lutz et al., 2013), to estimate future changes in glacier extent."

120    Referee 2

Overall comments

I am satisfied the authors addressed the issues raised by the previous reviewers, and did so in a professional and courteous manner. The one exception is with respect to improving the readability of the document, more work needs to be done in this space (please see below). In responding to the reviewers comments the authors have clearly undertaken considerable additional analysis to overcome the shortcomings in the original manuscript, including re-running the model at a finer resolution, and using different and more appropriate statistical downscaling methods. As a result the manuscript would appear to be greatly improved upon the original. However, as mentioned below the main short coming with the manuscript is that the language is still poor and quite loose in places.

Reply:  Thanks for your positive evaluation in general, and the constructive comments, which are responded below.

Structure

I found the introduction to be long and I was unsure of its direction. Given its length I think it would be helpful to the reader if there were some subheadings to provide some additional structure this section. The introduction and consequently the entire manuscript would also benefit from a paragraph at the end of the introduction clearly describing the structure of the rest of the paper. This would greatly assist the reader follow the manuscript. There appears to be some semblance of this at the end of the introduction but I could not understand it.

- Reply:  Thanks for the comment. We have rewritten and shortened the introduction section. We have also rewritten the last paragraph of the introduction in order to give a better interpretation about the structure of the whole paper.

Language

Before this manuscript is published it needs to be professionally edited by a native English speaker, and the language needs to be tightened considerably to be considered for publication. The authors state in their response to review that the language has been corrected by a native English speaker, however, I found the manuscript still very difficult and slow to read. In most (though not all) cases the meaning of a sentence is apparent but only after reading the sentence multiple times. I provide a few miscellaneous examples below. There are far too many poorly worded sentences to list here.

Reply: We have paid much effort on this comment including asking a native English speaker to proofread the whole manuscript.

e.g. Line 269 – "….from WRF at mountainous over 4000 m and 4800 m amsl are almost triple times as that from Gauge".

- Reply: We corrected it to be: "The winter precipitation from gauge and WRF over different altitudes are listed in Table 3, from which we can see that the winter precipitation from WRF at altitudes over 4000 m and 4800 m a.m.s.l. are almost three times higher than the gauged data."

Line 306 – "We used the data of 1986 for three preceding spin-up years."

- Reply: We corrected it to be: " We repeated 1986 three times as spin-up for the model."

Line 322 – "The results confirmed that there is much heavier precipitation at high altitude in Himalya regions than what we knew from the gauge data and other gridded data set".

- Reply: We corrected it to be: "This suggests that the high-altitude precipitation in the Himalayan upper Beas basin is underestimated in the gauge data, which was also found for other commonly used gridded data sets in previous studies (Immerzeel et al., 2015; Li et al., 2017)."

Line 384 – "The annual mean temperature of Beas river basin is approximately warm up to 1.8C (RCP4.5) and 2.8 (RCP8.5)…."

- Reply: We have corrected it to be: "The annual temperature of the upper Beas basin may warm up to ~1.8°C (RCP4.5) and ~2.8 °C (RCP8.5) in the middle of the century (2046-2065) and up to ~2.3 °C (RCP4.5) and ~5.4 °C (RCP8.5) at the end of the century (2080-2099) compared with the historical period (1986-2005). "

Line 408 – "There is a consistent trend of projected hydrological changes over all the scenarios, although there are large uncertainities".

- Reply: We corrected it to be: "There is a consistent trend of projected hydrological changes from all the scenarios, although there are large variabilities regarding seasonality and magnitude."

Line 432 – "There are many uncertainties and challenges for the future hydrological projection under climate change in the Beas river basin".

- Reply: We have corrected it to be: " There are large uncertainties in the future hydrological projections under climate change for the upper Beas basin."

Line 438 – "Most common knowledge of one o fhte challenges in high mountain areas is the data issue".

- Reply: We have corrected it to be: "One of the well-known challenges in high-altitude area is the data issue."

In a number of cases language is too colloquial. E.g. Line 207 – "right now".

- Reply: We have removed the "right now" in the sentence.

The word "uncertainty" is used very loosely throughout the entire manuscript. Finding a range in the output of selection of models (e.g. 6 GCM) is not a measure of "uncertainty". In most instances it would be more appropriate to say the models displayed a large range or a large variability in results. I appreciate that this loose use of the word uncertainty is somewhat common in this field of science, nevertheless I ask the authors to be more considered in using the word "uncertainty" in this manuscript, preferably only using the term if the uncertainty is being explicitly quantified.
For example Line 109 – "Chen e al. (2011) investigated the uncertainty of six dynamical and statistical downscaling methods in quantifying the hydrological impacts under climate change in a Canadian river basin. A significant uncertainty was found to be associated with the choice of downscaling methods, which is comparable to uncertainty from GCM." I am not familiar with this study but I suspect the authors didn't really investigate the uncertainty but simply compared the range in output of the methods/models.

- Reply: Thanks for the suggestion. We agree that the uncertainty should be used more carefully. We have gone through the whole manuscript and have changed some of them: "Chen et al. (2011) investigated the variability of six dynamical and statistical downscaling methods in quantifying hydrological impacts under climate change in a Canadian river basin. A large range in results was found to be associated with the choice of downscaling method, which is comparable to the range stemming from different GCMs." and "For example, Samaniego et al. (2017) set up six hydrological models in seven large river basins over the world, which were forced by bias-corrected outputs from five GCMs under RCP2.6 and RCP8.5 for the period 1971-2099. They found that the selection of the GCM mostly dominated the variability of hydrological results for the projections of runoff drought characteristics in general and emphasized the need for multi-model ensembles for the assessment of future drought projections." Besides, we have added more discussions of the streamflow change regarding different future scenarios in revised manuscript.

Please avoid using the phrase "water availability" unless you define it upfront. Water availability is ambiguous and means different things to different people. For example Line 63 – "reduction in water availability" – what do you mean? I honestly don't know. Do you mean reduction in streamflow or discharge? If so why not just say a reduction in streamflow or discharge.

- Reply: Thanks for the comment. Yes, it mainly means streamflow. Sorry for the confusion in the previous manuscript. We have changed the "water availability" to be "streamflow" in the revised manuscript.

In a number of places the language needs to be tightened. Again too many to list every instance but for example:
Line 427 – (6) the largest increase of evaporation will be in April, with also the largest spread…" Aside from the fact the meaning of the sentence is unclear, the use of "will" is inappropriate and should be replaced with "…is projected to…".

- Reply: Thanks for the comment. We have corrected it to be " the largest increase of evaporation is projected to be in April and the largest spread of evaporation increase is also found in April, i.e., around 5 ~ 26 mm and 1 ~ 26 mm by LOCI and DBC, respectively.".

Additional material
There are already a lot of figures and the authors have already undertaken considerable additional analysis in response to the reviewer's comments. Hence I am hesitant to suggest additional material. However, given the authors comment that in re-running all simulations at 3*3 km resolution and that they found that the results of calibration and validation were no better than the results using 10*10 km resolution, this may be an additional finding that may be worth documenting in the manuscript. I do not recall seeing any mention of this in the manuscript. Assuming the 3 and 10 km output data constitute an apples with apples comparison the authors may like to consider structuring an additional question/aim around providing insights into the additional value provided by using a finer model resolution?

- Reply: Thanks for the suggestion. Given the limited gauge data that we have in the study area, it is not a surprise that the results from 3*3 km are not improved from that from 10*10 km. According to the previous studies and analysis of the influence of interpolation and station density on gridded daily data (i.e. Dirksa et al., 1998; Hofstra et al., 2010; Xu et al., 2013), the results showed that the network density could introduce biases in the mean and variance of the grid values (i.e. precipitation and temperature) compared to those expected for the true area-averages. But this is certainly an interesting research goal. Regarding the length of our manuscript, we think it is better to further work on it in another paper. We have commented this finding in the Discussion of 5.2 section: "Furthermore, we found that the modelling results from 3 * 3 km resolution are not improved much from that of 10 * 10 km resolution, which is probably due to the limited gauge data in the study area. This limitation of data availability, e.g., sparse rainfall stations and absence of snowfall measurements, in such high-mountain drainage

basins also leads to considerable uncertainty in the hydrological simulation, and this is a common challenge for modeling studies in this region."

Specific comments
50 – "proper" – who is to say this is a "proper" representative glacier module. What makes this one proper and all others improper? They are all models after all.

- Reply: We have corrected it to be: "However, most hydrological models either do not have a representation of glaciers (Ali et al., 2015; Horton et al., 2006; Stahl et al., 2008) or have a simple glacier representation.".

69 – I suggest changing introductory sentence to paragraph to something like "…GMCs are USUALLY downscaled by an appropriate regional climate model (RCM) OR STATISTICAL DOWNSCALED for use …." Caps are new words. Then go onto talk about RCM and their limitations before talking about statistical downscaling. Note I included usually because often many studies just use pattern scaling or simple empirical scaling methods.

- Reply: Thanks for the suggestion. We have rewritten it to be: " To investigate the climate change impact on the future hydrological cycle, the variables produced by GCMs are usually dynamically downscaled by using a Regional Climate Model (RCM) or downscaled using empirical-statistical methods for use as inputs in hydrological models. These approaches are adopted because the outputs of GCMs are too coarse to directly drive hydrological models at regional or basin scale, in particular over mountainous terrain (Akhtar et al., 2008)."

79 – Are statistical downscaling methods the most popular and widely used approach? What about pattern scaling and other simple empirical scaling approaches (which are not statistical downscaling methods).

- Reply: We have removed "most" and reformulated the related sentence to be: "Considering the large biases in GCMs and RCMs, empirical-statistical downscaling is a popular and widely used approach to generate inputs for hydrological models to analyses the impact of climate change on hydrology".

153 – I think the aims need to be rewritten. Do the authors really quantify the uncertainty in the precipitation data? What do they mean by (3) How are the uncertainties of the future water from GCMs or statistical downscaling methods? It was difficult to compare the conclusions with the aims. Furthermore the structure of the introduction doesn't entirely sit comfortably with the order of the aims. For example it seems a little odd that the first question the paper seeks to address is the uncertainty in precipitation, yet this was the last thing discussed in the introduction.

- Reply: Thanks for the comment. We realized that to quantify the uncertainty in precipitation data is not the aim in our paper. We have now rewritten the aims of the paper and reconstructed this part in the revised manuscript to be: "The following research questions are examined in this paper: (1) How will the river streamflow change due to higher glacier melt under a warmer future in the upper Beas basin? (2) How large will the variability be in future key hydrological terms regarding different climate scenarios (i.e., RCPs, GCMs and statistical downscaling methods) in the upper Beas river basin? To answer the questions, we used a glacio-hydrological model to assess future glacio-hydrological changes in the Himalayan Beas river basin forced with two ensembles of four GCMs under two scenarios (RCP 4.5 and RCP 8.5), and two bias correction methods. Our paper is structured as follows: after the introduction, a description of the study area and data is presented, followed by the methods utilized, including the GSM-WASMOD model, glacier evolution parameterization, bias correction methods, precipitation correction and model calibration. Next, we focus on the simulation of the present-day water cycle, and calibration and validation of the model to observed data. Then, the results of future

climate change and its impact on glacier extent and hydrological projections are presented. Finally, a more detailed discussion on uncertainties of precipitation over high-altitude region and future hydrological projections in the upper Beas basin are addressed before presenting the main conclusions."

Besides, we have shortened the introduction section. We have also rewritten the last paragraph of the introduction in order to give a better interpretation about the structure of the whole paper.

170 – A bit unclear. Elsewhere you talk of max and min temperature. Is this mean daily temperature or mean daily maximum temperature and is it the mean daily temperature that falls below 2 C in January or the mean daily minimum temperature.

- Reply: We corrected it to be "daily temperature" which is also the input data for modelling.

186 – two ensembles of four GCM, can you please be a little more clear, the brevity of this description is to ambiguous for such an important point.

- Reply: We have rewritten it in the revised manuscript as "Two ensembles of four statistically downscaled GCMs under RCP4.5 (i.e., CanESM2, Inmcm4, IPSL_CM5A_LR and MRI_CGCM3) and RCP 8.5 (i.e., CSIRO_Mk3_6_0, MRI-ESM1, IPSL_CM5A_LR and MIROC5) (Taylor et al., 2012) are chosen to force the future simulations (see in Table 2)."

190 – methodology – sorry but this is one of my pet hates, and you may choose to ignore me as others do ☺. Technically and originally methodology is the study of methods (e.g. a study of different farming systems is a methodology) or the science or organisation. What you describe is a method or methods not a methodology.

- Reply: We have changed it to: "Methods".

195 – Did ESRI ARcGIS system v9.0 really define the glacier grid cells?

- Reply: The glacier grid cell was defined by the glacier outlines which were taken from the recently published Randolph Glacier Inventory (RGI 6.0). Sorry that we didn't make it clear enough in the previous paper. We have corrected it in the revised manuscript as "Those glacier grid cells were defined by glacier outlines from the RGI (6.0) (2017) (https://doi.org/10.7265/N5-RGI-60) ".

222- Does a conceptual glacio-hydrological model really mean that the glacial extent does not change in the historical simulation? To me a conceptual model simply means that one or more parameters cannot be physically assigned, and requires calibration. Maybe this sentence needs rephrasing?

- Reply: Thanks for the comment. We have rewritten it as "GSM-WASMOD is a conceptual glacio-hydrological model and we assume that the number of glacier-covered grid cells does not change in the historical simulation. For the future simulations, we used a basin-scale regionalized glacier mass balance model with parameterization of glacier area changes and subsequent aggregation of regional glacier characteristics (Lutz et al., 2013), to estimate future changes in glacier extent."

306 – "We used the data of 1986…" This does not make sense.

- Reply: We have corrected it as "We repeated 1986 three times as spin-up for the mode."

311-314 –Note Nash-Sutcliffe efficiency (NSE) coefficient is the original reference not NSC. Also how did you select the best parameter set based on three indices? Did you equally weight them? Did you give more preference to one than the other?

314- This needs to be rephrased. It is very loose definition.

- Reply: Sorry we failed to make it clear enough in the previous paper. We have now corrected it to be "Nash-Sutcliffe efficiency (NSE) coefficient" and changed all "NSC" to be "NSE" in the revised manuscript.
  Regarding the best parameter set, we used only NSE for chosen best parameter set and the other two indices are used for evaluation. We now have corrected this part of context in the revised manuscript: "The best parameter set was then chosen based on the Nash-Sutcliffe efficiency (NSE) coefficient, and two more indices, including relative volume error (VE) and root-mean-square error (RMSE) are also used for evaluation. For perfect model performance, the NSE value is 1 and the values of VE and RMSE are 0."

316 – annual already specified hence do not need to say mm/yr.

- Reply: We have gone through the whole manuscript and have corrected all the "mm/yr" to be "mm" if there is "annual" already specified.

356 – "w.e. a-1" a-1 is not necessary given you already state this is the measured annual glacier mass balance.

- Reply: We have gone through the whole manuscript and have corrected all the " w.e. a-1" to be "w.e." if there is "annual" already specified.

435 – How is 5% during 1986-2004 (i.e. 18 years) comparable to 5% during 2003-2008 (i.e. 5 years)?? One is three times the time period of the other?

- Reply: Sorry for the confusion. We have corrected it to be: "There are large uncertainties in the future hydrological projections under climate change for the upper Beas basin. The contribution of snow and glacier melt is significant for the total runoff, which varies from 27.5 % ~ 40% in previous studies (e.g., Kumar et al., 2007; Li et al., 2013a, 2015a). Besides, in the study of Kääb et al. (2015), the researchers used ICESat satellite altimetry data and estimated that 5% of the runoff originated from glacier retreat in the upper Beas river basin during 2003-2008. In our study, the total snow and glacier melt from the glacier-covered area is estimated to contribute around 19% of the total runoff, and the glacier retreat is accounting for around 5% during 1986-2004."

431 – This heading is misleading. No uncertainty assessment is provided.

- Reply: We agree, and we have changed it to be "Uncertain high-altitude precipitation".

Figures and tables
Sorry to be vague but all figure and table captions need tightening and editing.

- Reply: We have revised all the captions and rewritten all of them. Please see them in the revised manuscript.

Figure 1 – can you use a different colour ramp for elevation? The blue (low values) is too similar to the blue colour of the glaciers. Also legend should be Elevation (m) not DEM (m).

- Reply: Thanks for the comment. We have now changed both the color ramp for the elevation and colors of the glaciers in order to make it clearer in the Figure 1. We have also changed "DEM (m)" to be "Elevation (m)". Please see the updated Figure 1.

Figure 2 – Need to define JAS and DJFM so figures can be read as stand alone.

- Reply: We have rewritten the caption of fig2 as "Fig. 2 Seasonal precipitation of July - September (JAS) and December - March (DJFM) during 1998-2005 from 3km WRF (from Li et al., 2017) and Gauge (dot) in the upper Beas basin.".

Figure 7 – This figure is hard to interpret because some of the data appear to be missing or exactly overlay on one another. Is it possible to show these more clearly. For example LOCI would appear to be missing from three of the four plots.

- Reply: Thanks for your comments. We have found DBC was missing in one of the sub-figures, so we have now corrected it and re-drawn the fig. 7. However, the data of LOCI are not missed but is overlaid by other lines. For downscaling precipitation, the mean-based LOCI method can correct the mean wet-day precipitation to be exactly as the observation, so in the upper left-side sub-figure, the shading of LOCI is overlaid by the line of observation. For downscaling temperature, the LOCI method corrects both the mean and standard deviation of temperature to be exactly as the observation. So that the shading of LOCI is overlaid by the line of observation in the two sub-figures at the bottom panel. Furthermore, the DBC method corrects the distribution of temperature, indirectly correcting the mean and standard deviation of temperature to be equal to the observation. So, in the two sub-figures at bottom panel, the shading of DBC is overlaid by the shading of LOCI and the line of observation.

[revised manuscript text omitted]